# Generalized FitzHugh-Nagumo equations with Caputo gH-differentiability: A novel fuzzy fractional approach to digital memristor networks

Muhammad Yousuf [1], Uzma Ahmad[1], Ghulam Muhammad[2]*, Hamed Alsulami[3]

1 Institute of Mathematics, University of the Punjab, Lahore, Pakistan, 2 Department of Mathematics, Lahore Garrison University, Lahore, Pakistan, 3 Department of Mathematics, King Abdulaziz University, Jeddah, Saudi Arabia

* ghulammuhammad@lgu.edu.pk

## Abstract

The fuzzy fractional generalized FitzHugh-Nagumo differential equations (FFGFH-NDEs) is a well-known and generalized model that plays a significant role in biological systems, including complex synchronization in brain networks, cardiac dynamics, propagation of signals through nerve impulses, and digital circuit theory. The analytical study of the FFGFH-NDEs is more complex and difficult to deal with. An effective and efficient technique is required to solve FFGFH-NDEs analytically. This article introduces and investigates the analytical fuzzy solutions of FFGFH-NDEs using fuzzy fractional Caputo generalized Hukuhara (*FFCgH*)-differentiability. The closed-form solutions of FFGFH-NDEs for various cases and types of *FFCgH*-differentiability are extracted for the homogeneous and nonhomogeneous case of the concerned model. The potential solutions are determined using fuzzy Laplace transform (*FLT*) and are presented in terms of multivariate Mittag-Leffler functions (MLFs). To highlight the innovation of this work, the digital memristor networks problem is designed and solved as an application of the proposed study including the graphical analysis to understand the uncertain behavior of the proposed model.

## 1 Introduction

Over the last few decades, fractional differential equations (FDEs) have established themselves as one of the most active and potential domains of discussion among mathematicians because they are able to capture the non-integer behaviour of systems more accurately than the standard integer-order equations. The versatility of FDEs allow them to take account of memory effects as well as heredity characteristics, which means they are useful in a whole range of scientific and engineering fields. All of these applications lie in the fields of physics, biology, chemistry, applied mathematics and engineering. FDEs generate rigorous problems in computing their analytical solutions, which calls for efficient and effective methods in their resolution. The field of fractional calculus has thus registered some important developments,

**Data availability statement:** All relevant data supporting the findings of this study are contained within the paper.

**Funding:** The author(s) received no specific funding for this work.

**Competing interests:** The authors have declared that no competing interests exist.

and there are many research articles which can be considered critical in improving the overall concept of the field. Some articles that made significant contributions to the analysis include those of [1,2] and [3], which provided a good account of the entire process.

### Fuzzy differential equations with an integer and fractional order derivatives

All real-world physical problems are inherently uncertain and vague. However, due to the presence of uncertainty, modelling and obtaining meaningful results from this sort of problem can often be quite challenging. The science and engineering domains, in most of their cases, depend on differential equations, which are constrained by very complex environmental factors. Newly formulated fuzzy differential equations ($\mathbb{F}$DEs) have been formulated to address this complexity more effectively. These $\mathbb{F}$DEs are, however, inherently difficult to manage. Realistically, there is very little data available about the variables and parameters of these systems, and most of what is available is vague and lacking. This imprecision gives rise to measurement, experiment or observation errors. To deal with this ambiguity and uncertainty, researchers either use stochastic and statistical or interval and fuzzy methods. Uncertainty in stochastic and probabilistic processes stems from natural randomness and is conventionally expressed using a probability density function. Nevertheless, this approach demands sufficient information on the variables and parameters for an accurate estimate of the distribution. On the other hand, uncertainty arising from incomplete or imprecise knowledge of variables and parameters is treated by means of interval and fuzzy set theories. The objective of the development of these methods is to lighten the interval uncertainty present in computational and mathematical models. The research articles [4,5] investigate the basic advancements in the field of interval and fuzzy analysis. An innovative methodology for solving interval-valued differential equations was proposed by Wang et al. [6] using the gH-derivative. The $\mathbb{F}$DEs are of major significance for modelling dynamic systems in the case of uncertainty and vagueness when the boundaries are well known. The notion of $\mathbb{F}$DEs was first presented in 1978, but by 1982, the idea had matured considerably. This concept is based on the fuzzy derivative, called the Dubois-Prade derivative [7]. Many fuzzy derivatives have been defined such as Puri-Ralescu derivatives (H-derivative) [8]; Goetschel-Voxman derivative [9]; Seikala derivative (S-derivative) [10] and Friedman-Ming-Kandel derivative [11]. However, the H-derivative and S-derivative are most widely used in academic literature when dealing with continuous fuzzy valued functions (FvFs). The Hukuhara difference (H-difference) [12] defines bounds of FvFs through ꞩ-cut, while the S-derivative uses ꞩ-cut to define the lower and upper bounds of FvFs. The study of the $\mathbb{F}$DEs mainly proves the existence and uniqueness of solutions, develops new solution methods, proposes diverse fuzzy derivatives, and studies the featured equations from an understanding perspective. In [13], Kaleva establishes the existence and uniqueness of solutions to $\mathbb{F}$DEs. The non-decreasing diameter of FvFs limits the H-derivative and S-derivative novelty. Therefore, such derivatives have some limitations when used in real world problems related to $\mathbb{F}$DEs. To overcome these limitations, Zadeh [14] proposed the extension principle method

as a solution for $\mathbb{F}$DEs. An approach to study $\mathbb{F}$DEs by mining popular homologous sequences has received a lot of attention, but does not define the fuzzy derivative. This area is novelized by the introduction of the same and reverse order derivative. On the other hand, this derivative overcomes the shortcomings of the Hukuhara and Seikkala derivatives, while keeping their strong relationship to the generalized forms of each. The strongly generalized Hukuhara derivative ($\mathbb{SGH}$-derivative) was introduced by Bede [15] for two cases of differentiability. The first form is very similar to the H-derivative, while the second one takes care of the non-decreasing diameter of FvFs. This derivative is unique along with the switching points at which first form of differentiability switches to the second form in certain intervals. The $\mathbb{SGH}$-derivative is a major one that has been used in many important studies. This offers utility but the $\mathbb{SGH}$-derivative suffers from the same limitations as the *H*-difference, principally relying on it. As a result, it does not exists when the *H*-difference is undefined.

The limitations of the $\mathbb{SGH}$-derivative are tackled by the $\mathbb{H}$-derivative, introduced by Bede and Stefanini [16]. Esmi et al. [17] used strong linear independence to establish connections between $\mathbb{H}$-derivative and the fuzzy derivative. The idea of solving fuzzy fractional differential equations ($\mathbb{F}$FDEs) has been introduced by Agarwal et al. [18] in 2010. This problem has hence drawn the attention of many researchers [19–21], in which many methods for solving $\mathbb{F}$FDEs have been developed. Different forms of derivatives such as Riemann-Liouville ($\mathbb{RL}$), Caputo, Caputo-Fabrizio and conformable for the FVFs have been proposed. The classical derivative is extended by these derivatives. Jeong [22] derived the existence and uniqueness of $\mathbb{F}$FDEs basing on $\mathbb{RL}$-derivative. A solution method of the $\mathbb{F}$FDEs is introduced by Salahshour et al. [23] using the $\mathbb{SGH}$-derivative and the fractional Laplace transform (*FLT*). Afterward, Akram et al. [24] provided analytical solutions to the Langevin $\mathbb{F}$FDEs using *FLT*. Many researchers [25–27] contributed a lot of work on the stochastic systems of time-variying differential equations. In a Pythagorean fuzzy environment, Akram et al. [28,29] extended the concept of $\mathbb{F}$FDEs and derive the solutions in terms of $\mathbb{SGH}$-differentiability. In $\mathbb{F}$FDEs, $\mathbb{RL}$ and Caputo derivatives are used and they differ from one another in certain important aspects such as the zero result of the Caputo derivative of a constant function does not exist but for the case of $\mathbb{RL}$-derivative. There is also the fact that $\mathbb{RL}$ $\mathbb{F}$FDEs have fractional order initial conditions while Caputo derivative needs integer order initial conditions. Based on integrating Caputo-derivative and $\mathbb{H}$-derivative, Allahviranloo et al. [30] introduced the concept of the $\mathbb{F}$FDEs under the Caputo $\mathbb{H}$-derivative.The existence and uniqueness of $\mathbb{F}$FDEs were examined by Arshad and Lupulescu [31]. The application of Schauder fixed point theorem established results for $\mathbb{F}$FDEs by Agarwal et al. [32]. A special function plays an important role in fractional calculus theory such as MLF is of particular significance [33]. The concept of MLF is widely used to determine the analytical solutions of the complex fuzzy fractional models. With the help of single variable, the concept of bivariate MLF ($\mathbb{BV}$MLF) and trivariate MLF ($\mathbb{TV}$MLF) is presented in [34] to determine the solution of $\mathbb{F}$DE*s*. Recently, MLF is further extended in more than three variables by numerous authors [34,35] that leads to many applications in diverse mathematical domains such as control theory and fractional calculus. Akram et al. [36] extended this concept and derived explicit analytic solutions for systems of $\mathbb{F}$FDEs with incommensurate orders. They further generalized this approach to obtain solutions for coupled systems of $\mathbb{F}$FDEs [37]. Allahviranloo et al. [38] broadened the concept of generalized differentiability and introduced level-wise *gH*-differentiability for one-dimensional FvFs. They also applied artificial neural networks to solve fractional integro-differential equations [39]. Akın et al. [40] developed a novel algorithm for solving second-order $\mathbb{F}$FDEs. Ghulam et al. [41] established the solution of fuzzy fractional generalized Bagley–Torvik equation using fuzzy Caputo gH-differentiability. Ghulam et al. [42] introduced the solution of fuzzy Langevin fractional delay differential equations using granular derivative. Ghulam et al. [43] proposed the bounded and symmetric solutions in dual form of fully bipolar fuzzy linear systems. Akram et al. [44] developed the incommensurate non-homogeneous system of fuzzy linear fractional differential equations. Solutions to $\mathbb{F}$FDEs using the natural transform method and interactive derivatives are discussed in [45,46]. An and Hoa [47] investigated the stability of controlled $\mathbb{F}$FDEs using the Caputo derivative of random order. Alinezhad and Allahviranloo [48] proposed a solution procedure for optimal control problems using the Caputo derivative in a fuzzy setting. The FitzHugh-Nagumo differential equations (FH-NDEs) forms the key mathematical model in the class of excitable systems, which is of widespread use in neuroscience and cardiac dynamics. FH-NDE is the simpler version of

the more complex Hodgkin Huxley model for action potentials in a neurons. The FH-NDE explains how electrical signals propagate through neurons, signal transmission, and synchronization in brain networks. The remarkable work has been done by the researchers [49–52] on the neuromorphic networks based on memristor platform. Originally the FH-NDE was developed in the 1960 by Richard FitzHugh and John Nagumo. However, the FH-NDE has emerged in one a numerous domains such as applied mathematics, physics and neuroscience. The FH-NDEs is used to explain how electrical signals are transmitted along the neurons and the signals are transmitted and synchronized in the neural networks. Moreover, FH-NDE was analyzed by Devendra Kumar et al. [53], where a numerical scheme based on the combination of the q-homotopy analysis method and the Laplace transform was applied. The conformable Sumudu decomposition method was applied analytically to the FH-NDE by Suliman Alfaqeih and Emine Misirli [54]. Uma et al. [55] recently developed a numerical method of solving stochastic partial FH-NDE model which occur in models of biological sciences. Yousif et al. [56] presented a finite difference $\beta$-fractional method of solving the time-fractional FH-NDE.

## Motivation and contribution

The fractional FH-NDE (FFH-NDE) has also attracted considerable attention due to its ability to represent real world physical phenomena like, signal transmission, synchronization in brain networks and cardiac dynamics. The solution of FFH-NDE neurons model was studied by Shaher Momani et al. [57] using the multi-step generalized differential transform method. Using homotopy perturbation transform technique, Amit Prakash and Hardish Kaur [58] studied FFH-NDE. The fractional reduced differential transform method was applied by Ramani et al. [59] to solve the FFH-NDE. Akinnukawe et al. [60] described a hybrid second derivative two step algorithm in numerical integration for the solution of non linear FH-NDE. The FFH-NDEs have attained the attention because of their capability to model real life physical phenomena such as signal transmission, synchronization within brain networks and cardiac dynamics. All above techniques only provide the solutions of FFH-NDEs for exact initial conditions and data without uncertainty. Fan et al. [61] established the semi-analytical solution of time FFH-NDE using semi-analytical techniques. The previous study motivated us to develop the FFGFH-NDEs, a generalized model that plays a significant role in biological systems such as complex synchronization in brain networks, cardiac dynamics, propagation of signals through neurons and to digital circuit theory. Mathematically, the FFGFH-NDEs are formulated as a reaction-diffusion equation, expressed in the form:

$$
\begin{cases}
\left[{}^{\mathbb{CF}}_{gH}\mathfrak{D}^{\varsigma}_{+}\mathbb{S}\right](\mathfrak{r},\mathfrak{f}) \ominus \left[{}^{\mathbb{CF}}_{gH}\mathfrak{D}^{\gamma}_{+}\mathbb{S}\right](\mathfrak{r},\mathfrak{f}) \oplus \mathbb{S}(\mathfrak{r},\mathfrak{f})\left[(1-\mathbb{S}(\mathfrak{r},\mathfrak{f}))(\mathbb{S}(\mathfrak{r},\mathfrak{f})-\psi)\right] = 0, \\[2ex]
\mathbb{S}(\mathfrak{r},0) = (\mathbb{S}_1,\mathbb{S}_2,\mathbb{S}_3) \odot \tilde{\ell}(\mathfrak{r}), \\[2ex]
\mathbb{S}'(\mathfrak{r},0) = (\mathbb{S}_1,\mathbb{S}_2,\mathbb{S}_3) \odot \tilde{j}(\mathfrak{r}),
\end{cases}
\tag{1}
$$

where $\left[{}^{\mathbb{CF}}_{gH}\mathfrak{D}^{\varsigma}_{+}\mathbb{S}\right](\mathfrak{r},\mathfrak{f})$ and $\left[{}^{\mathbb{CF}}_{gH}\mathfrak{D}^{\gamma}_{+}\mathbb{S}\right](\mathfrak{r},\mathfrak{f})$ represents the $\mathbb{CFF}gH$-derivatives of FVF $\mathbb{S}(\mathfrak{r},\mathfrak{f})$ having orders $0 < \gamma \leq 1$ and $0 \leq \varsigma \leq 2$ respectively. The initial conditions $\mathbb{S}(\mathfrak{r},0)$ and $\mathbb{S}'(\mathfrak{r},0)$ represent triangular FVFs with respect to spatial variable $\mathfrak{r}$. The FFGFH-NDEs act as an extension of classical FFH-NDEs in the fuzzy environments, where the $\mathbb{CFF}gH$-derivatives of $\mathbb{S}(\mathfrak{r},\mathfrak{f})$ deals with the memory effects in time, the orders $\varsigma$ and $\gamma$ of the fractional derivatives control the memory strength, fuzzy threshold function $\psi$ acts as a control function with the graded values ranging with in [0,1] and the initial conditions as an triangular FVFs helps to investigates how uncertainty propagates through fractional derivatives and non-linear dynamics. The FFGFH-NDE, a prominent and generalized reaction-diffusion model which is widely employed in describing nerve impulse transmission and digital memristors netwoks. It serves as a system for bifurcations, stability and chaotic behavior in dynamical systems and to understand how complex behavior arises from simple rules. The fuzzy fractional FitzHughNagumo equations are extensions of neurodynamical models that allow an expression of uncertainty with initial

states that are fuzzy, memory effect formulation using a fractional operator, and the maintenance of nonlinear excitability provided by cubic FHN structure, providing a strong and biologically natural of a system that can be used when creating a framework of neuronal behavior under variability and long-term memory interactions. The novelty of FFGFH-NDEs is presented as follows:

(i). The novel and generalized model of FFGFH-NDEs is established with initial conditions as the triangular fuzzy valued functions.

(ii). The generalized results in the form of theorems are presented in order to determine the fuzzy solutions of the FFGFH-NDEs under different types of *FFCgH* differentiability.

(iii). An effective and efficient schematic technique is developed for determining the analytical fuzzy solutions of FFGFH-NDEs.

(iv). The fuzzy solutions of FFGFH-NDEs are constructed using the *FFCgH* differentiability along with *FLT* in the form of multivariate MLFs. Furthermore, the fuzzy solutions of FFGFH-NDEs are discussed for different values of $\psi$ and fractional orders $\gamma$ and $\varsigma$.

(v). The graphical illustration of analytical fuzzy solutions as the multivariate Mittag-Leffler functions is presented to understand the complexity and novelty of our work.

(vi). The comparative analysis of the fuzzy solutions with the existing techniques of crisp solutions of the proposed model is presented in order to validate the innovation and better understanding of the fuzzy solutions of FFGFH-NDEs.

(vii). To highlight the innovation of this work, the real-world application of FFGFH-NDEs in digital memristor networks is designed through proper correspondence and analyzed by various parameters.

The rest of the article is designed: Sect 2 summarizes some important concepts and results of special functions and fuzzy fractional calculus. With the aid of MLF, the analytical solution scheme to the FFGFH-NDEs along with some important results are developed in Sect 3. To validate the main results of Sect 3, illustrative examples are presented in Sect 4. Also, for particular values of $\psi$ such as $\psi = 1, 0$ and fractional orders $\gamma; \varsigma$, some examples are presented in this Section. Sect 5 addresses real-world applications of FFGFH-NDEs in digital memristor networks to demonstrate the originality of the proposed approach along with the graphical representation. Finally, Sect 6 concludes the paper and sketches avenues for future research.

## 2 Preliminaries

This section covers the very basic concepts and definitions of fuzzy fractional calculus, FLT, and MLF, which is an integral part of fuzzy fractional calculus. The notations used throughout this article are given in Table 1.

**Definition 1.** [62] Suppose that $\mathbb{R}$ denotes the real line. A fuzzy set $q$ on the real line $\mathbb{R}$ is characterized by rule of membership $q : [\mathfrak{s}, \mathfrak{d}] \subset \mathbb{R} \longrightarrow [0, 1]$ with the conditions that $q$ is bounded support, upper semi-continuous and convex. Through $\natural_{\mathbb{R}}$, we define the collection of fuzzy numbers on $\mathbb{R}$. The $\mathfrak{n}$-cut of $q$ is symbolized as $[q]^{\mathfrak{n}}$ and is defined in two cases: If $\mathfrak{n} \in (0, 1]$, then $[q]^{\mathfrak{n}} = \{x \in \mathbb{R} \ : \ q(x) \geq \mathfrak{n}\}$. For specific case, if $\mathfrak{n} = 0$, then $[q]^{\mathfrak{n}} = cl(supp \ q)$ . It can be defined in parametric form as: $[q]^{\mathfrak{n}} = [\underline{q}(\mathfrak{n}), \overline{q}(\mathfrak{n})]$.

**Definition 2.** [8] Suppose that $q_1, q_2 \in \natural_{\mathbb{R}}$. The H-difference of FNs $q_1$ and $q_2$ denoted by $q_1 \ominus q_2$ is defined as follows:

$$q_1 = q_2 \oplus q_3$$

with the condition that there exists $q_3 \in \natural_{\mathbb{R}}$.

**Table 1**. Table of notations.

| Notations | Representation |
|---|---|
| *FLT* | Fuzzy Laplace Transform |
| $\mathbb{FFI}$ | Fuzzy Fractional Integral |
| $\mathbb{F}$*DEs* | Fuzzy Differential Equations |
| $\natural_{\mathbb{R}}$ | The class of fuzzy numbers on $\mathbb{R}$ |
| FDEs | Fractional Differential Equations |
| $\mathbb{UVMLF}$ | Univariate Mittag-Leffler functions |
| $\mathbb{BVMLF}$ | Bivariate Mittag-Leffler functions |
| $\mathbb{TVMLF}$ | Trivariate Mittag-Leffler functions |
| $\mathcal{C}^{\natural_{\mathbb{R}}}(0,\mathfrak{u})$ | The family of continuous FVFs on $(0,\mathfrak{u})$ |
| $\mathbb{F}$*FDEs* | Fuzzy Fractional Differential Equations |
| $\mathbb{IFFLT}$ | Inverse Fuzzy Fractional Laplace Transform |
| $\mathscr{L}^{\natural_{\mathbb{R}}}(0,\mathfrak{u})$ | The class of Lebesgue integrable FVFs on $(0,\mathfrak{u})$ |
| FFH-NDEs | Fractional FitzHugh-Nagumo differential equations |
| $\mathbb{C}^1((0,\mathfrak{u}),\natural_{\mathbb{R}})$ | The class of *gH*-differentiable and continuous FVFs on $(0,\mathfrak{u})$ |
| FFGFH-NDEs | Fuzzy Fractional Generalized FitzHugh-Nagumo differential equations |

**Definition 3.** [63] Let $q_1, q_2 \in \natural_{\mathbb{R}}$, then the generalized H-difference of FNs denoted by $q_1 \ominus_{gH} q_2$ for $q_3 \in \natural_{\mathbb{R}}$ is defined by

$$q_1 \ominus_{gH} q_2 = q_3 \Longleftrightarrow \begin{cases} \quad (1) \quad q_1 = q_2 \oplus q_3, \\ \text{or} \quad (2) \quad q_1 = q_2 \ominus (-1)q_3, \end{cases}$$

where the Minkovski addition of $q_1$ and $q_2$ is denoted by $\oplus$.

Suppose that $\mathbb{S} : (0,\mathfrak{u}) \longrightarrow \natural_{\mathbb{R}}$ is a FVF. The reader is referred [16] to understand the fundamental concepts of continuity, limit and *gH*-differentiability of first form ($\mathbb{F}^* - \mathbb{F}^*$) and second form ($\mathbb{S}^{\oplus} - \mathbb{F}^*$) of FVF. Through $\mathbb{C}^1((0,\mathfrak{u}),\natural_{\mathbb{R}})$, we denote the class of all *gH*-differentiable and continuous FVFs on $(0,\mathfrak{u})$. For $\mathbb{S} \in \mathbb{C}^1((0,\mathfrak{u}),\natural_{\mathbb{R}})$, the reader is also referred [64] to investigate the $\mathbb{RL}$ integral and $\mathbb{CF}$- derivative. In this paper, we denote the class of all Lebesgue integrable and continuous FVFs on $(0,\mathfrak{u})$ by $\mathscr{L}^{\natural_{\mathbb{R}}}(0,\mathfrak{u})$ and $\mathcal{C}^{\natural_{\mathbb{R}}}(0,\mathfrak{u})$ respectively.

**Definition 4.** [30] Let $\mathbb{S} : (0,\mathfrak{u}) \longrightarrow \natural_{\mathbb{R}}$, where $\mathbb{S} \in \mathscr{L}^{\natural_{\mathbb{R}}}(0,\mathfrak{u}) \cap \mathcal{C}^{\natural_{\mathbb{R}}}(0,\mathfrak{u})$ be a FVF, then the $\mathbb{CFF}$- derivative of $\mathbb{S}(\mathfrak{r})$ of fractional order $\mu \in (0,1)$ is given by:

$$^{\mathbb{CF}}_{gH}\mathfrak{D}^{\mu}_{+}\mathbb{S}(\mathfrak{r}) = \frac{1}{\Gamma(1-\mu)} \int_0^{\mathfrak{r}} (\mathfrak{r}-\mathfrak{k})^{-\mu} \odot \mathbb{S}'_{gH}(\mathfrak{k}) d\mathfrak{k}. \tag{2}$$

For $\mu \in (1,2)$, the $\mathbb{CFF}$- derivative of $\mathbb{S}(\mathfrak{r})$ is related as:

$$^{\mathbb{CF}}_{gH}\mathfrak{D}^{\mu}_{+}\mathbb{S}(\mathfrak{r}) = \frac{1}{\Gamma(2-\mu)} \int_0^{\mathfrak{r}} (\mathfrak{r}-\mathfrak{k})^{1-\mu} \odot \mathbb{S}'_{gH}(\mathfrak{k}) d\mathfrak{k}. \tag{3}$$

**Definition 5.** [34] Let $\mathbb{S} \in \mathbb{C}^{\natural_{\mathbb{R}}}(0,\mathfrak{u}) \cap L^{\natural_{\mathbb{R}}}(0,\mathfrak{u})$, then $\mathbb{BVMLF}$ with $\mathfrak{d}_1, \mathfrak{d}_2, \mathfrak{d}_3, \omega \in \mathbb{C}$ as its parameters and $\mathrm{Re}(\mathfrak{d}_1), \mathrm{Re}(\mathfrak{d}_2), \mathrm{Re}(\mathfrak{d}_3) > 0$ is defined as

$$\mathbb{E}^{\omega}_{\mathfrak{d}_1,\mathfrak{d}_2,\mathfrak{d}_3}(\mathfrak{r}_1,\mathfrak{r}_2) = \sum_{\tilde{\imath}=0}^{\infty} \sum_{\tilde{\jmath}=0}^{\infty} \frac{(\omega)_{\tilde{\imath}+\tilde{\jmath}} \mathfrak{r}_1^{\tilde{\imath}} \mathfrak{r}_2^{\tilde{\jmath}}}{\Gamma(\tilde{\imath}\mathfrak{d}_1 + \tilde{\jmath}\mathfrak{d}_2 + \mathfrak{d}_3)\tilde{\imath}!\tilde{\jmath}!}. \tag{4}$$

Using $\omega = 1$ and $\mathfrak{r}_1, \mathfrak{r}_2$ as power functions, then $\mathbb{UVMLF}$ is related as

$$\mathfrak{r}^{\mathfrak{d}_3-1}\mathbb{E}^1_{\mathfrak{d}_1,\mathfrak{d}_2,\mathfrak{d}_3}(\xi\mathfrak{r}^{\mathfrak{d}_1},\varrho\mathfrak{r}^{\mathfrak{d}_2}) = \sum_{\tilde{i}=0}^{\infty}\sum_{\tilde{j}=0}^{\infty}\frac{(\tilde{i}+\tilde{j})!}{\tilde{j}!\tilde{i}!}\frac{\xi^{\tilde{j}}\varrho^{\tilde{j}}}{\Gamma(\tilde{i}\mathfrak{d}_1+\tilde{j}\mathfrak{d}_2+\mathfrak{d}_3)\tilde{i}!\tilde{j}!}\mathfrak{r}^{\tilde{i}\mathfrak{d}_1+\tilde{j}\mathfrak{d}_2+\mathfrak{d}_3-1}. \tag{5}$$

For $\mathbb{UVMLF}$, the $\mathbb{FFI}$ is defined as follows:

$$\left({}_a\mathbb{I}^{\omega;\xi,\varrho}_{\mathfrak{d}_1,\mathfrak{d}_2,\mathfrak{d}_3}\mathbb{S}\right)(\mathfrak{r}) = \int_0^{\mathfrak{r}}(\mathfrak{r}-\tau)^{\mathfrak{d}_3-1}\mathbb{E}^{\omega}_{\mathfrak{d}_1,\mathfrak{d}_2,\mathfrak{d}_3}(\xi(\mathfrak{r}-\tau)^{\mathfrak{d}_1},\varrho(\mathfrak{r}-\tau)^{\mathfrak{d}_2})\odot\mathbb{S}(\tau)d\tau, \quad \text{for } \mathfrak{r} > 0. \tag{6}$$

**Definition 6.** [65] Let $\mathbb{S} \in \mathbb{C}^{\natural_{\mathbb{R}}}(0,\mathfrak{u}) \cap L^{\natural_{\mathbb{R}}}(0,\mathfrak{u})$, then a $\mathbb{TVMLF}$ with $\mathfrak{d}_1,\mathfrak{d}_2,\mathfrak{d}_3,\mathfrak{d}_4,\omega \in \mathbb{C}$ as its five parameters and $\mathrm{Re}(\mathfrak{d}_1),\ \mathrm{Re}(\mathfrak{d}_2),\ \mathrm{Re}(\mathfrak{d}_3) > 0$ is given by

$$\mathbb{E}^{\omega}_{\mathfrak{d}_1,\mathfrak{d}_2,\mathfrak{d}_3,\mathfrak{d}_4}(\mathfrak{r}_1,\mathfrak{r}_2,\mathfrak{r}_3) = \sum_{\tilde{i}=0}^{\infty}\sum_{\tilde{j}=0}^{\infty}\sum_{\tilde{k}=0}^{\infty}\frac{(\omega)_{\tilde{i}+\tilde{j}+\tilde{k}}\mathfrak{r}_1^{\tilde{j}}\mathfrak{r}_2^{\tilde{j}}\mathfrak{r}_3^{\tilde{k}}}{\Gamma(\tilde{i}\mathfrak{d}_1+\tilde{j}\mathfrak{d}_2+\tilde{k}\mathfrak{d}_3+\mathfrak{d}_4)\tilde{i}!\tilde{j}!\tilde{k}!}. \tag{7}$$

Suppose that $\mathfrak{r}_1 = \xi\mathfrak{r}^{\mathfrak{d}_1}, \mathfrak{r}_2 = \varrho\mathfrak{r}^{\mathfrak{d}_2}, \mathfrak{r}_3 = \sigma\mathfrak{r}^{\mathfrak{d}_3}$, then the Eq (7) reduces in $\mathbb{UVMLF}$ as follows

$$\mathfrak{r}^{\mathfrak{d}_4-1}\mathbb{E}^{\omega}_{\mathfrak{d}_1,\mathfrak{d}_2,\mathfrak{d}_3,\mathfrak{d}_4}(\xi\mathfrak{r}^{\mathfrak{d}_1},\varrho\mathfrak{r}^{\mathfrak{d}_2},\sigma\mathfrak{r}^{\mathfrak{d}_3}) = \sum_{\tilde{i}=0}^{\infty}\sum_{\tilde{j}=0}^{\infty}\sum_{\tilde{k}=0}^{\infty}\frac{(\omega)_{\tilde{i}+\tilde{j}+\tilde{k}}\xi^{\tilde{j}}\varrho^{\tilde{j}}\sigma^{\tilde{k}}}{\Gamma(\tilde{i}\mathfrak{d}_1+\tilde{j}\mathfrak{d}_2+\tilde{k}\mathfrak{d}_3+\mathfrak{d}_4)\tilde{i}!\tilde{j}!\tilde{k}!}\mathfrak{r}^{\tilde{i}\mathfrak{d}_1+\tilde{j}\mathfrak{d}_2+\tilde{k}\mathfrak{d}_3+\mathfrak{d}_4-1}. \tag{8}$$

Substituting $\omega = 1$ in Eq (8), one gets $\mathbb{UVMLF}$ as a special case of $\mathbb{TVMLF}$

$$\mathfrak{r}^{\mathfrak{d}_4-1}\mathbb{E}^1_{\mathfrak{d}_1,\mathfrak{d}_2,\mathfrak{d}_3,\mathfrak{d}_4}(\xi\mathfrak{r}^{\mathfrak{d}_1},\varrho\mathfrak{r}^{\mathfrak{d}_2},\sigma\mathfrak{r}^{\mathfrak{d}_3}) = \sum_{\tilde{i}=0}^{\infty}\sum_{\tilde{j}=0}^{\infty}\sum_{\tilde{k}=0}^{\infty}\frac{(\tilde{i}+\tilde{j}+\tilde{k})!}{\tilde{i}!\tilde{j}!\tilde{k}!}\frac{\xi^{\tilde{j}}\varrho^{\tilde{j}}\sigma^{\tilde{k}}}{\Gamma(\tilde{i}\mathfrak{d}_1+\tilde{j}\mathfrak{d}_2+\tilde{k}\mathfrak{d}_3+\mathfrak{d}_4)}\mathfrak{r}^{\tilde{i}\mathfrak{d}_1+\tilde{j}\mathfrak{d}_2+\tilde{k}\mathfrak{d}_3+\mathfrak{d}_4-1}. \tag{9}$$

For $\mathbb{TVMLF}$, the $\mathbb{FFI}$ in univariate form is defined as follows:

$$\left({}_a\mathbb{I}^{\omega;\xi,\varrho,\sigma}_{\mathfrak{d}_1,\mathfrak{d}_2,\mathfrak{d}_3,\mathfrak{d}_4}\mathbb{S}\right)(\mathfrak{r}) = \int_0^{\mathfrak{r}}(\mathfrak{r}-\tau)^{\mathfrak{d}_3-1}\mathbb{E}^{\omega}_{\mathfrak{d}_1,\mathfrak{d}_2,\mathfrak{d}_3,\mathfrak{d}_4}(\xi(\mathfrak{r}-\tau)^{\mathfrak{d}_1},\varrho(\mathfrak{r}-\tau)^{\mathfrak{d}_2},\sigma(\mathfrak{r}-\tau)^{\mathfrak{d}_3})\odot\mathbb{S}(\tau)d\tau, \text{for } \mathfrak{r} > 0, \tag{10}$$

where $\mathfrak{d}_1,\mathfrak{d}_2,\mathfrak{d}_3,\mathfrak{d}_4,\omega,\xi,\varrho,\sigma \in \mathbb{C}$ provided that $\mathrm{Re}(\mathfrak{d}_1),\ \mathrm{Re}(\mathfrak{d}_2),\ \mathrm{Re}(\mathfrak{d}_3),\ \mathrm{Re}(\mathfrak{d}_4) > 0$. Setting $\omega = 0$, the Eq (10) reduces to $\mathbb{RL}$ integral having order $\mathfrak{d}_4$ as

$$\left({}_a\mathbb{I}^{\omega;\xi,\varrho,\sigma}_{\mathfrak{d}_1,\mathfrak{d}_2,\mathfrak{d}_3,\mathfrak{d}_4}\mathbb{S}\right)(\mathfrak{r}) = \sum_{\tilde{i}=0}^{\infty}\sum\sum_{k=0}^{\infty}\frac{(\omega)_{\tilde{i}+\tilde{j}+k}\xi^{\tilde{j}}\varrho^{\tilde{j}}\sigma^k}{\tilde{i}!\tilde{j}!k!}\left(\mathbb{I}^{\tilde{i}\mathfrak{d}_1+\tilde{j}\mathfrak{d}_2+k\mathfrak{d}_3+\mathfrak{d}_4}_{a_+}\mathbb{S}\right)(\mathfrak{r}). \tag{11}$$

**Definition 7.** [66] Let $\mathbb{S} \in \mathbb{C}^{\natural_{\mathbb{R}}}(0,\mathfrak{u}) \cap L^{\natural_{\mathbb{R}}}(0,\mathfrak{u})$ provided that $\mathbb{S}(\mathfrak{r}) \odot e^{-\varphi\mathfrak{r}}$ is improper integrable on the interval $[0,\infty)$, then the *FLT* denoted by $\mathsf{L}[\mathbb{S}(\mathfrak{r})]$ is defined by

$$\mathsf{L}[\mathbb{S}(\mathfrak{r})] = \int_0^{\infty}\mathbb{S}(\mathfrak{r})\odot e^{-\varphi\mathfrak{r}}d\mathfrak{r}. \tag{12}$$

In $\mathfrak{s}$-cut form, Eq (12) can be expressed as follows

$$\int_0^\infty \mathbb{S}(\mathfrak{r}) \odot e^{-\mathfrak{k}\mathfrak{r}} d\mathfrak{r} = \left[ \int_0^\infty \mathbb{S}^-(\mathfrak{r}; \mathfrak{s}) e^{-\mathfrak{k}\mathfrak{r}} d\mathfrak{r}, \int_0^\infty \mathbb{S}^+(\mathfrak{r}; \mathfrak{s}) e^{-\mathfrak{k}\mathfrak{r}} d\mathfrak{r} \right], \tag{13}$$

where

$$\mathsf{L}[\mathbb{S}^-(\mathfrak{r}; \mathfrak{s})] = \int_0^\infty \mathbb{S}^-(\mathfrak{r}; \mathfrak{s}) e^{-\mathfrak{k}\mathfrak{r}} d\mathfrak{r} \quad ; \quad \mathsf{L}[\mathbb{S}^+(\mathfrak{r}; \mathfrak{s})] = \int_0^\infty \mathbb{S}^+(\mathfrak{r}; \mathfrak{s}) e^{-\mathfrak{k}\mathfrak{r}} d\mathfrak{r}. \tag{14}$$

The linearity property of *FLT* $\mathsf{L}$ on the FVFs $\mathbb{S}(\mathfrak{r})$ and $\mathfrak{d}(\mathfrak{r})$ is defined as follows:

**Lemma 1.** [66] Let $\mathbb{S}(\mathfrak{r}), \mathfrak{d}(\mathfrak{r}) \in \mathbb{C}^{\natural_\mathbb{R}}(0, \mathfrak{u}) \cap L^{\natural_\mathbb{R}}(0, \mathfrak{u})$ and $x, w \in \mathbb{R}$. Then

$$\mathsf{L}[x \odot \mathbb{S}(\mathfrak{r}) \oplus w \odot \mathfrak{d}(\mathfrak{r})] = x \odot \mathsf{L}[\mathbb{S}(\mathfrak{r})] \oplus w \odot \mathsf{L}[\mathfrak{d}(\mathfrak{r})]. \tag{15}$$

**Theorem 8.** [66] Let $\mathbb{S}(\mathfrak{r}) \in \mathbb{C}^{\natural_\mathbb{R}}(0, \mathfrak{u}) \cap L^{\natural_\mathbb{R}}(0, \mathfrak{u})$ be a FVF and $\natural \in \mathbb{R}$, then

$$(\mathbb{S} \star \natural)(\mathfrak{r}) = \int_0^{\mathfrak{r}} \mathbb{S}(\mathfrak{k}) \odot \natural(\mathfrak{r} - \mathfrak{k}) d\mathfrak{k}. \tag{16}$$

The *FLT* of $(\mathbb{S} \star \natural)(\mathfrak{r})$ is given by

$$\mathsf{L}[(\mathbb{S} \star \natural)(\mathfrak{r})] = \mathsf{L}[\mathbb{S}(\mathfrak{r})] \odot \mathsf{L}[\natural(\mathfrak{r})]. \tag{17}$$

**Theorem 9.** [35] Suppose that $\mu_2 < \mu_1$, $\mu_3 < \mu_1$, where $\mu, \mathfrak{t} \in \mathbb{R}$ and $0 < \text{Re}(\mathfrak{m})$, then the $\mathbb{IFFLT}$ and $\mathbb{MLF}$ are related by the following result

$$\mathsf{L}^{-1}\left[ \frac{\mathfrak{m}^{\mu_3}}{\mathfrak{m}^{\mu_1} - \mu\mathfrak{m}^{\mu_2} - \mathfrak{t}} \right](\mathfrak{k}) = \mathfrak{k}^{\mu_1 - \mu_3 - 1} \mathbb{E}_{\mu_1, \mu_1 - \mu_2, \mu_1 - \mu_3}(\mathfrak{t}\mathfrak{k}^{\mu_1}, \mu\mathfrak{k}^{\mu_1 - \mu_2}). \tag{18}$$

**Corollary 1.** [35] Given any $\mathfrak{r} \in \mathbb{R}$ and $\mu_1, \mu_2, \mu_3, \mu, \mathfrak{t} \in \mathbb{R}$ provided that $0 < \mu_1, \mu_2, \mu_3 - 1 > \lfloor \mu_1 \rfloor$, we have

$$\begin{array}{c} {}^{\mathrm{CF}}_{gH}\mathfrak{D}^{\mu_1}_+ \left[ \mathfrak{r}^{\mu_3 - 1} \mathbb{E}_{\mu_1, \mu_2, \mu_3}(\mathfrak{t}\mathfrak{r}^{\mu_1}, \mu\mathfrak{r}^{\mu_2}) \right] = \mathfrak{r}^{\mu_3 - \mu_1 - 1} \mathbb{E}_{\mu_1, \mu_2, \mu_3 - \mu_1}(\mathfrak{t}\mathfrak{r}^{\mu_1}, \mu\mathfrak{r}^{\mu_2}). \end{array} \tag{19}$$

**Theorem 10.** [35] Given any $\mu_1, \mu_2, \mu_3 \in \mathbb{R}$ and $0 < \mu_1, \mu_2$. The $\mathbb{FFI}$ corresponding to $\mathbb{UVMLF}$ is defined by the following expression:

$$(\mathbb{I}^{1;\mathfrak{t},\mu}_{\mu_1,\mu_2,\mu_3} \mathfrak{d})(\mathfrak{r}) = \int_0^{\mathfrak{r}} (\mathfrak{r} - \mathfrak{s})^{\mu_1 - 1} \mathbb{E}^1_{\mu_1,\mu_2,\mu_3}(\mathfrak{t}(\mathfrak{r} - \mathfrak{s})^{\mu_1}, \mu(\mathfrak{r} - \mathfrak{s})^{\mu_1 - \mu_2}) \odot \mathfrak{d}(\mathfrak{s}) d\mathfrak{s}.$$

## 3 Fuzzy fractional generalized FitzHugh-Nagumo differential equations

Let $\mathbb{S}(\mathfrak{r}, \mathfrak{k})$ be a Lebesgue integrable and continuous FVF on $(0, \mathfrak{u})$ such that $\mathbb{S}(\mathfrak{r}, \mathfrak{k}) \in \mathbb{C}^{\natural_\mathbb{R}}(0, \mathfrak{u}) \cap L^{\natural_\mathbb{R}}(0, \mathfrak{u})$. Suppose that $\mathbb{S}(\mathfrak{r}, \mathfrak{k})$ represents a fuzzy valued transmembrane function of space and time variables $\mathfrak{r}$ and $\mathfrak{k}$ respectively. For the fuzzy threshold function $\psi : \mathbb{R} \longrightarrow [0, 1]$ and the fuzzy initial conditions $\mathbb{S}_0$ and $\mathbb{S}'_0$ with respect to spatial variable $\mathfrak{r}$ and time

variable $\mathfrak{k} = 0$, the FFGFH-NDEs are developed as:

$$
\begin{cases}
\left[{}_{gH}^{\mathbb{CF}}\mathfrak{D}_+^{\varsigma}\mathbb{S}\right](\mathfrak{r},\mathfrak{k}) \ominus \left[{}_{gH}^{\mathbb{CF}}\mathfrak{D}_+^{\gamma}\mathbb{S}\right](\mathfrak{r},\mathfrak{k}) \oplus \mathbb{S}(\mathfrak{r},\mathfrak{k})[(1 - \mathbb{S}(\mathfrak{r},\mathfrak{k}))(\mathbb{S}(\mathfrak{r},\mathfrak{k}) - \psi)] = 0, \\[2ex]
\mathbb{S}(\mathfrak{r},0) = (\mathbb{S}_1, \mathbb{S}_2, \mathbb{S}_3) \odot \tilde{\ell}(\mathfrak{r}) \quad 0 < \mathfrak{r} < 1, \\[2ex]
\mathbb{S}'(\mathfrak{r},0) = (\mathbb{S}_1, \mathbb{S}_2, \mathbb{S}_3) \odot \tilde{j}(\mathfrak{r}) \quad 0 < \mathfrak{r} < 1,
\end{cases}
\tag{20}
$$

where $\left[{}_{gH}^{\mathbb{CF}}\mathfrak{D}_+^{\varsigma}\mathbb{S}\right](\mathfrak{r},\mathfrak{k})$ and $\left[{}_{gH}^{\mathbb{CF}}\mathfrak{D}_+^{\gamma}\mathbb{S}\right](\mathfrak{r},\mathfrak{k})$ represents the $\mathbb{CFF}gH$-derivatives of FVF $\mathbb{S}(\mathfrak{r},\mathfrak{k})$ having orders $0 < \gamma \leq 1$ and $0 \leq \varsigma \leq 2$ respectively. The initial conditions $\mathbb{S}(\mathfrak{r},0)$ and $\mathbb{S}'(\mathfrak{r},0)$ represent triangular FVFs with respect to spatial variable $\mathfrak{r}$. The FFGFH-NDEs act as an extension of classical FFH-NDEs in the fuzzy environments, where the $\mathbb{CFF}gH$-derivatives of $\mathbb{S}(\mathfrak{r},\mathfrak{k})$ deals with the memory effects in time, the orders $\varsigma$ and $\gamma$ of the fractional derivatives control the memory strength, fuzzy threshold function $\psi$ acts as a control function with the graded values ranging with in [0,1] and the initial conditions as an triangular FVFs helps to investigates how uncertainty propagates through fractional derivatives and non-linear dynamics. We present some necessary theorems that will play a key role in solving the FFGFH-NDE. First, we present a theorem dealing with different types of *FLT* of ${}_{gH}^{\mathbb{CF}}\mathfrak{D}_+^2\mathbb{S}(\mathfrak{r},\mathfrak{k})$, which play a significant role for obtaining the results of generalized *FFCgH*-derivatives of FVF $\mathbb{S}(\mathfrak{r},\mathfrak{k})$. Moreover, we develop the results of *FLT* of ${}_{gH}^{\mathbb{CF}}\mathfrak{D}_+^{\varsigma}\mathbb{S}(\mathfrak{r},\mathfrak{k})$ and ${}_{gH}^{\mathbb{CF}}\mathfrak{D}_+^{\gamma}\mathbb{S}(\mathfrak{r},\mathfrak{k})$ under the types of fuzzy differentiability for the fractional order $1 < \varsigma \leq 2$.

**Theorem 11.** Suppose that a FVF $\mathbb{S}(\mathfrak{r},\mathfrak{k})$ is primitive provided that $e^{\mathfrak{mk}} \odot \mathbb{S}(\mathfrak{r},\mathfrak{k})$, $e^{\mathfrak{mk}} \odot {}_{gH}^{\mathbb{CF}}\mathfrak{D}_+\mathbb{S}(\mathfrak{r},\mathfrak{k})$ and $e^{\mathfrak{mk}} \odot {}_{gH}^{\mathbb{CF}}\mathfrak{D}_+^2\mathbb{S}(\mathfrak{r},\mathfrak{k})$ are continuous and fuzzy Riemann integrable on the interval $[0,\infty)$, then the following results arise

(a) If a FVF $\mathbb{S}(\mathfrak{r},\mathfrak{k})$ and ${}_{gH}^{\mathbb{CF}}\mathfrak{D}_+\mathbb{S}(\mathfrak{r},\mathfrak{k})$ are *FFCgH*-differentiable in its $\mathbb{F}^* - \mathbb{F}^*$, then

$$
\mathsf{L}\left[{}_{gH}^{\mathbb{CF}}\mathfrak{D}_+^2\mathbb{S}(\mathfrak{r},\mathfrak{k})\right](\mathfrak{m}) = \left\{\mathfrak{m}^2 \odot \mathsf{L}[\mathbb{S}(\mathfrak{r},\mathfrak{k})](\mathfrak{m}) \ominus \mathfrak{m} \odot \mathbb{S}(\mathfrak{r},0)\right\} \ominus \mathbb{S}'(\mathfrak{r},0)).
\tag{21}
$$

(b) If a FVF $\mathbb{S}(\mathfrak{r},\mathfrak{k})$ is *FFCgH*-differentiable in its $\mathbb{F}^* - \mathbb{F}^*$ and ${}_{gH}^{\mathbb{CF}}\mathfrak{D}_+\mathbb{S}(\mathfrak{r},\mathfrak{k})$ is *FFCgH*-differentiable in its $\mathbb{S}^{\circledR} - \mathbb{F}^*$, then

$$
\mathsf{L}\left[{}_{gH}^{\mathbb{CF}}\mathfrak{D}_+^2\mathbb{S}(\mathfrak{r},\mathfrak{k})\right](\mathfrak{m}) = -\mathbb{S}'(\mathfrak{r},0) \ominus \left\{(-\mathfrak{m}^2 \odot \mathsf{L}[\mathbb{S}(\mathfrak{r},\mathfrak{k})](\mathfrak{m})) \ominus (-\mathfrak{m} \odot \mathbb{S}(\mathfrak{r},0))\right\}.
\tag{22}
$$

(c) If a FVF $\mathbb{S}(\mathfrak{r},\mathfrak{k})$ is *FFCgH*-differentiable in its $\mathbb{S}^{\circledR} - \mathbb{F}^*$ and ${}_{gH}^{\mathbb{CF}}\mathfrak{D}_+\mathbb{S}(\mathfrak{r},\mathfrak{k})$ is *FFCgH*-differentiable in its $\mathbb{F}^* - \mathbb{F}^*$, then

$$
\mathsf{L}\left[{}_{gH}^{\mathbb{CF}}\mathfrak{D}_+^2\mathbb{S}(\mathfrak{r},\mathfrak{k})\right](\mathfrak{m}) = \left\{(-\mathfrak{m} \odot \mathbb{S}(\mathfrak{r},0)) \ominus (-\mathfrak{m}^2 \odot \mathsf{L}[\mathbb{S}(\mathfrak{r},\mathfrak{k})](\mathfrak{m}))\right\} \ominus \mathbb{S}'(\mathfrak{r},0)).
\tag{23}
$$

(d) If a FVF $\mathbb{S}(\mathfrak{r},\mathfrak{k})$ and ${}_{gH}^{\mathbb{CF}}\mathfrak{D}_+\mathbb{S}(\mathfrak{r},\mathfrak{k})$ are *FFCgH*-differentiable in its $\mathbb{S}^{\circledR} - \mathbb{F}^*$, then

$$
\mathsf{L}\left[{}_{gH}^{\mathbb{CF}}\mathfrak{D}_+^2\mathbb{S}(\mathfrak{r},\mathfrak{k})\right](\mathfrak{m}) = (-\mathbb{S}'(\mathfrak{r},0)) \ominus \left\{(-\mathfrak{m} \odot \mathbb{S}(\mathfrak{r},0)) \ominus \mathfrak{m}^2 \odot \mathsf{L}[\mathbb{S}(\mathfrak{r},\mathfrak{k})](\mathfrak{m})\right\}.
\tag{24}
$$

**Proof 12.** It is easy to prove the results of cases (a), (b), (c) and (d), therefore left as an exercise. □

**Theorem 13.** Let $\mathbb{S} : (0, \mathfrak{u}) \longrightarrow \natural_{\mathbb{R}}$ provided that $\mathbb{S}(\mathfrak{r}, \mathfrak{k}) \in \mathbb{C}^{\natural_{\mathbb{R}}}(0, \mathfrak{u}) \cap L^{\natural_{\mathbb{R}}}(0, \mathfrak{u})$. Suppose that $\left[{}^{\mathrm{CF}}_{gH}\mathfrak{D}^{\varsigma}_{+}\mathbb{S}\right](\mathfrak{r}, \mathfrak{k})$ follows piecewise continuity on the interval $[0, \infty)$ and $\mathbb{S}$ is of exponential order $\varsigma > 0$ provided that $\varsigma < \mathrm{Re}(\mathfrak{m})$, then the following cases arises:

(a) If a FVF $\mathbb{S}(\mathfrak{r}, \mathfrak{k})$ and ${}^{\mathrm{CF}}_{gH}\mathfrak{D}^{\varsigma}_{+}\mathbb{S}(\mathfrak{r}, \mathfrak{k})$ are *FFCgH-differentiable* in its $\mathbb{F}^{\circledast} - \mathbb{F}^{\circledast}$, then $\mathbb{FFLT}$ of ${}^{\mathrm{CF}}_{gH}\mathfrak{D}^{\varsigma}_{+}\mathbb{S}(\mathfrak{r}, \mathfrak{k})$ is related as

$$\mathsf{L}\left[{}^{\mathrm{CF}}_{gH}\mathfrak{D}^{\varsigma}_{+}\mathbb{S}(\mathfrak{r}, \mathfrak{k})\right](\mathfrak{m}) = \left\{\mathfrak{m}^{\varsigma} \odot \mathsf{L}[\mathbb{S}(\mathfrak{r}, \mathfrak{k})](\mathfrak{m}) \ominus \mathfrak{m}^{\varsigma-1} \odot \mathbb{S}(\mathfrak{r}, 0)\right\} \ominus (\mathfrak{m}^{\varsigma-2} \odot \mathbb{S}'(\mathfrak{r}, 0)). \tag{25}$$

(b) If a FVF $\mathbb{S}(\mathfrak{r}, \mathfrak{k})$ is *FFCgH-differentiable* in its $\mathbb{F}^{\circledast} - \mathbb{F}^{\circledast}$ and ${}^{\mathrm{CF}}_{gH}\mathfrak{D}^{\varsigma}_{+}\mathbb{S}(\mathfrak{r}, \mathfrak{k})$ is *FFCgH-differentiable* in its $\mathbb{S}^{\circledast} - \mathbb{F}^{\circledast}$, then $\mathbb{FFLT}$ of ${}^{\mathrm{CF}}_{gH}\mathfrak{D}^{\varsigma}_{+}\mathbb{S}(\mathfrak{r}, \mathfrak{k})$ is related as

$$\mathsf{L}\left[{}^{\mathrm{CF}}_{gH}\mathfrak{D}^{\varsigma}_{+}\mathbb{S}(\mathfrak{r}, \mathfrak{k})\right](\mathfrak{m}) = (-\mathfrak{m}^{\varsigma-2} \odot \mathbb{S}'(\mathfrak{r}, 0)) \ominus \left\{(-\mathfrak{m}^{\varsigma} \odot \mathsf{L}[\mathbb{S}(\mathfrak{r}, \mathfrak{k})](\mathfrak{m})) \ominus (-\mathfrak{m}^{\varsigma-1} \odot \mathbb{S}(\mathfrak{r}, 0))\right\}. \tag{26}$$

(c) If a FVF $\mathbb{S}(\mathfrak{r}, \mathfrak{k})$ is *FFCgH-differentiable* in its $\mathbb{S}^{\circledast} - \mathbb{F}^{\circledast}$ and ${}^{\mathrm{CF}}_{gH}\mathfrak{D}^{\varsigma}_{+}\mathbb{S}(\mathfrak{r}, \mathfrak{k})$ is *FFCgH-differentiable* in its $\mathbb{F}^{\circledast} - \mathbb{F}^{\circledast}$, then $\mathbb{FFLT}$ of ${}^{\mathrm{CF}}_{gH}\mathfrak{D}^{\varsigma}_{+}\mathbb{S}(\mathfrak{r}, \mathfrak{k})$ is related as

$$\mathsf{L}\left[{}^{\mathrm{CF}}_{gH}\mathfrak{D}^{\varsigma}_{+}\mathbb{S}(\mathfrak{r}, \mathfrak{k})\right](\mathfrak{m}) = \left\{(-\mathfrak{m}^{\varsigma-1} \odot \mathbb{S}(\mathfrak{r}, 0)) \ominus (-\mathfrak{m}^{\varsigma} \odot \mathsf{L}[\mathbb{S}(\mathfrak{r}, \mathfrak{k})](\mathfrak{m}))\right\} \ominus (\mathfrak{m}^{\varsigma-2} \odot \mathbb{S}'(\mathfrak{r}, 0)). \tag{27}$$

(d) If a FVF $\mathbb{S}(\mathfrak{r}, \mathfrak{k})$ and ${}^{\mathrm{CF}}_{gH}\mathfrak{D}^{\varsigma}_{+}\mathbb{S}(\mathfrak{r}, \mathfrak{k})$ are *FFCgH-differentiable* in its $\mathbb{S}^{\circledast} - \mathbb{F}^{\circledast}$, then $\mathbb{FFLT}$ of ${}^{\mathrm{CF}}_{gH}\mathfrak{D}^{\varsigma}_{+}\mathbb{S}(\mathfrak{r}, \mathfrak{k})$ is related as

$$\mathsf{L}\left[{}^{\mathrm{CF}}_{gH}\mathfrak{D}^{\varsigma}_{+}\mathbb{S}(\mathfrak{r}, \mathfrak{k})\right](\mathfrak{m}) = (-\mathfrak{m}^{\varsigma-2} \odot \mathbb{S}'(\mathfrak{r}, 0)) \ominus \left\{\mathfrak{m}^{\varsigma-1} \odot \mathbb{S}(\mathfrak{r}, 0) \ominus \mathfrak{m}^{\varsigma} \odot \mathsf{L}[\mathbb{S}(\mathfrak{r}, \mathfrak{k})](\mathfrak{m})\right\}. \tag{28}$$

**Proof 14.**     (a). Suppose that $\mathbb{S}(\mathfrak{r}, \mathfrak{k})$ and ${}^{\mathrm{CF}}_{gH}\mathfrak{D}^{\varsigma}_{+}\mathbb{S}(\mathfrak{r}, \mathfrak{k})$ are *FFCgH-differentiable* in its $\mathbb{F}^{\circledast} - \mathbb{F}^{\circledast}$, then from Eq (3), Definition 7 and convolution theorem of [67]

$$\mathsf{L}\left[{}^{\mathrm{CF}}_{gH}\mathfrak{D}^{\varsigma}_{+}\mathbb{S}(\mathfrak{r}, \mathfrak{k})\right](\mathfrak{m}) = \frac{1}{\Gamma(2-\varsigma)} \odot \mathsf{L}[\mathfrak{k}^{1-\varsigma}] \odot \mathsf{L}\left[{}^{\mathrm{CF}}_{gH}\mathfrak{D}^{2}_{+}\mathbb{S}(\mathfrak{r}, \mathfrak{k})\right](\mathfrak{m}). \tag{29}$$

From Eqs (21) and (29), we obtain

$$\mathsf{L}\left[{}^{\mathrm{CF}}_{gH}\mathfrak{D}^{\varsigma}_{+}\mathbb{S}(\mathfrak{r}, \mathfrak{k})\right](\mathfrak{m}) = \frac{1}{\Gamma(2-\varsigma)} \odot \mathsf{L}[\mathfrak{k}^{1-\varsigma}] \odot \left\{\mathfrak{m}^{2} \odot \mathsf{L}[\mathbb{S}(\mathfrak{r}, \mathfrak{k})](\mathfrak{m}) \ominus \mathfrak{m} \odot \mathbb{S}(\mathfrak{r}, 0)\right\} \ominus \mathbb{S}'(\mathfrak{r}, 0)) \tag{30}$$

Using the Eq (21) and setting $\frac{1}{\Gamma(2-\varsigma)} \odot \mathsf{L}[\mathfrak{k}^{1-\varsigma}] = \mathfrak{m}^{\varsigma-2}$ in Eq (30), we obtain the required result as follows

$$\mathsf{L}\left[{}^{\mathrm{CF}}_{gH}\mathfrak{D}^{\varsigma}_{+}\mathbb{S}(\mathfrak{r}, \mathfrak{k})\right](\mathfrak{m}) = \left\{\mathfrak{m}^{\varsigma} \odot \mathsf{L}[\mathbb{S}(\mathfrak{r}, \mathfrak{k})](\mathfrak{m}) \ominus \mathfrak{m}^{\varsigma-1} \odot \mathbb{S}(\mathfrak{r}, 0)\right\} \ominus (\mathfrak{m}^{\varsigma-2} \odot \mathbb{S}'(\mathfrak{r}, 0)). \tag{31}$$

The results of cases (b), (c) and (d) can be proved in the similar way.     $\square$

**Theorem 15.** Let $\mathbb{S} : (0, \mathfrak{u}) \longrightarrow \natural_{\mathbb{R}}$ provided that $\mathbb{S}(\mathfrak{r}, \mathfrak{k}) \in \mathbb{C}^{\natural_{\mathbb{R}}}(0, \mathfrak{u}) \cap L^{\natural_{\mathbb{R}}}(0, \mathfrak{u})$. Suppose that $\left[{}^{\mathrm{CF}}_{gH}\mathfrak{D}^{\varsigma}_{+}\mathbb{S}\right](\mathfrak{r}, \mathfrak{k})$ follows piecewise continuity on the interval $[0, \infty)$ and $\mathbb{S}$ is of exponential order provided that $1 < \varsigma \leq 2$, then the following cases arises:

(a) If a FVF $\mathbb{S}(\mathfrak{r},\mathfrak{k})$ and $^{\mathrm{CF}}_{gH}\mathfrak{D}^{\varsigma}_{+}\mathbb{S}(\mathfrak{r},\mathfrak{k})$ are *FFCgH-differentiable* in its $\mathbb{F}^{*} - \mathbb{F}^{*}$, then $\mathbb{FFLT}$ of $^{\mathrm{CF}}_{gH}\mathfrak{D}^{\varsigma}_{+}\mathbb{S}(\mathfrak{r},\mathfrak{k})$ provided that $1 < \varsigma \leq 2$ is related as

$$L\left[^{\mathrm{CF}}_{gH}\mathfrak{D}^{\varsigma}_{+}\mathbb{S}(\mathfrak{r},\mathfrak{k})\right](\mathfrak{m}) = \left\{\mathfrak{m}^{\varsigma} \odot L[\mathbb{S}(\mathfrak{r},\mathfrak{k})](\mathfrak{m}) \ominus \mathfrak{m}^{\varsigma-1} \odot \mathbb{S}(\mathfrak{r},0)\right\} \ominus (\mathfrak{m}^{\varsigma-2} \odot \mathbb{S}'(\mathfrak{r},0)). \tag{32}$$

(b) If a FVF $\mathbb{S}(\mathfrak{r},\mathfrak{k})$ is *FFCgH-differentiable* in its $\mathbb{F}^{*} - \mathbb{F}^{*}$ and $^{\mathrm{CF}}_{gH}\mathfrak{D}^{\varsigma}_{+}\mathbb{S}(\mathfrak{r},\mathfrak{k})$ is *FFCgH-differentiable* in its $\mathbb{S}^{\circledast} - \mathbb{F}^{*}$, then $\mathbb{FFLT}$ of $^{\mathrm{CF}}_{gH}\mathfrak{D}^{\varsigma}_{+}\mathbb{S}(\mathfrak{r},\mathfrak{k})$ provided that $1 < \varsigma \leq 2$ is related as

$$L\left[^{\mathrm{CF}}_{gH}\mathfrak{D}^{\varsigma}_{+}\mathbb{S}(\mathfrak{r},\mathfrak{k})\right](\mathfrak{m}) = (-\mathfrak{m}^{\varsigma-2} \odot \mathbb{S}'(\mathfrak{r},0)) \ominus \left\{(-\mathfrak{m}^{\varsigma} \odot L[\mathbb{S}(\mathfrak{r},\mathfrak{k})](\mathfrak{m})) \ominus (-\mathfrak{m}^{\varsigma-1} \odot \mathbb{S}(\mathfrak{r},0))\right\}. \tag{33}$$

(c) If a FVF $\mathbb{S}(\mathfrak{r},\mathfrak{k})$ is *FFCgH-differentiable* in its $\mathbb{S}^{\circledast} - \mathbb{F}^{*}$ and $^{\mathrm{CF}}_{gH}\mathfrak{D}^{\varsigma}_{+}\mathbb{S}(\mathfrak{r},\mathfrak{k})$ is *FFCgH-differentiable* in its $\mathbb{F}^{*} - \mathbb{F}^{*}$, then $\mathbb{FFLT}$ of $^{\mathrm{CF}}_{gH}\mathfrak{D}^{\varsigma}_{+}\mathbb{S}(\mathfrak{r},\mathfrak{k})$ provided that $1 < \varsigma \leq 2$ is related as

$$L\left[^{\mathrm{CF}}_{gH}\mathfrak{D}^{\varsigma}_{+}\mathbb{S}(\mathfrak{r},\mathfrak{k})\right](\mathfrak{m}) = \left\{(-\mathfrak{m}^{\varsigma-1} \odot \mathbb{S}(\mathfrak{r},0)) \ominus (-\mathfrak{m}^{\varsigma} \odot L[\mathbb{S}(\mathfrak{r},\mathfrak{k})](\mathfrak{m}))\right\} \ominus \mathfrak{m}^{\varsigma-2} \odot \mathbb{S}'(\mathfrak{r},0). \tag{34}$$

(d) If a FVF $\mathbb{S}(\mathfrak{r},\mathfrak{k})$ and $^{\mathrm{CF}}_{gH}\mathfrak{D}^{\varsigma}_{+}\mathbb{S}(\mathfrak{r},\mathfrak{k})$ are *FFCgH-differentiable* in its $\mathbb{S}^{\circledast} - \mathbb{F}^{*}$, then $\mathbb{FFLT}$ of $^{\mathrm{CF}}_{gH}\mathfrak{D}^{\varsigma}_{+}\mathbb{S}(\mathfrak{r},\mathfrak{k})$ provided that $1 < \varsigma \leq 2$ is related as

$$L\left[^{\mathrm{CF}}_{gH}\mathfrak{D}^{\varsigma}_{+}\mathbb{S}(\mathfrak{r},\mathfrak{k})\right](\mathfrak{m}) = -\mathfrak{m}^{\varsigma-2} \odot \mathbb{S}'(\mathfrak{r},0) \ominus \left\{\mathfrak{m}^{\varsigma-1} \odot \mathbb{S}(\mathfrak{r},0) \ominus (\mathfrak{m}^{\varsigma} \odot L[\mathbb{S}(\mathfrak{r},\mathfrak{k})])(\mathfrak{m})\right\}. \tag{35}$$

**Proof 16.** For any $\mathfrak{z}$ provided that $0 \leq \mathfrak{z} \leq 1$, the $\mathfrak{z}$-cut form of $\mathbb{FFLT}$ of $^{\mathrm{CF}}_{gH}\mathfrak{D}^{\varsigma}_{+}\mathbb{S}(\mathfrak{r},\mathfrak{k})$ is related as

$$\left[L[^{\mathrm{CF}}_{gH}\mathfrak{D}^{\varsigma}_{+}\mathbb{S}(\mathfrak{r},\mathfrak{k})](\mathfrak{m})\right]^{\mathfrak{z}} = \left[L[(^{\mathrm{CF}}_{gH}\mathfrak{D}^{\varsigma}_{+}\mathbb{S})^{-}_{\mathfrak{z}}(\mathfrak{r},\mathfrak{k})](\mathfrak{m}), [L[(^{\mathrm{CF}}_{gH}\mathfrak{D}^{\varsigma}_{+}\mathbb{S})^{+}_{\mathfrak{z}}(\mathfrak{r},\mathfrak{k})](\mathfrak{m})\right]. \tag{36}$$

(b). Suppose that $\mathbb{S}(\mathfrak{r},\mathfrak{k})$ is *FFCgH-differentiable* in its $\mathbb{F}^{*} - \mathbb{F}^{*}$ and $^{\mathrm{CF}}_{gH}\mathfrak{D}^{\varsigma}_{+}\mathbb{S}(\mathfrak{r},\mathfrak{k})$ is *FFCgH-differentiable* in its $\mathbb{S}^{\circledast} - \mathbb{F}^{*}$, then

$$\left[^{\mathrm{CF}}_{gH}\mathfrak{D}^{\varsigma}_{+}\mathbb{S}(\mathfrak{r},\mathfrak{k})\right]^{\mathfrak{z}} = \left[^{\mathrm{CF}}_{gH}\mathfrak{D}^{\varsigma}_{+}\mathbb{S}^{+}_{\mathfrak{z}}(\mathfrak{r},\mathfrak{k}), {}^{\mathrm{CF}}_{gH}\mathfrak{D}^{\varsigma}_{+}\mathbb{S}^{-}_{\mathfrak{z}}(\mathfrak{r},\mathfrak{k})\right]. \tag{37}$$

Applying $\mathbb{FFLT}$ on Eq (37), we deduce that

$$\left[L[^{\mathrm{CF}}_{gH}\mathfrak{D}^{\varsigma}_{+}\mathbb{S}(\mathfrak{r},\mathfrak{k})](\mathfrak{m})\right]^{\mathfrak{z}} = \left[L[(^{\mathrm{CF}}_{gH}\mathfrak{D}^{\varsigma}_{+}\mathbb{S}^{+}_{\mathfrak{z}})(\mathfrak{r},\mathfrak{k})](\mathfrak{m}), L[(^{\mathrm{CF}}_{gH}\mathfrak{D}^{\varsigma}_{+}\mathbb{S}^{-}_{\mathfrak{z}})(\mathfrak{r},\mathfrak{k})](\mathfrak{m})\right]. \tag{38}$$

From Eqs (26) and (38), we have

$$\left[L[^{\mathrm{CF}}_{gH}\mathfrak{D}^{\varsigma}_{+}\mathbb{S}(\mathfrak{r},\mathfrak{k})](\mathfrak{m})\right]^{\mathfrak{z}} = \left[(-\mathfrak{m}^{\varsigma-2} \odot \mathbb{S}^{+'}_{\mathfrak{z}}(\mathfrak{r},0)) \ominus \left\{(-\mathfrak{m}^{\varsigma} \odot L[\mathbb{S}^{+}_{\mathfrak{z}}(\mathfrak{r},\mathfrak{k})](\mathfrak{m})) \ominus (-\mathfrak{m}^{\varsigma-1}\right.\right.$$

$$\left.\left.\odot \mathbb{S}^{+}_{\mathfrak{z}}(\mathfrak{r},0))\right\}, (-\mathfrak{m}^{\varsigma-2} \odot \mathbb{S}^{-'}_{\mathfrak{z}}(\mathfrak{r},0)) \ominus \left\{(-\mathfrak{m}^{\varsigma} \odot L[\mathbb{S}^{-}_{\mathfrak{z}}(\mathfrak{r},\mathfrak{k})](\mathfrak{m})) \ominus (-\mathfrak{m}^{\varsigma-1} \odot \mathbb{S}^{-}_{\mathfrak{z}}(\mathfrak{r},0))\right\}\right]. \tag{39}$$

Rearranging the Eq (39) and using the fact that ${}^{\mathrm{CF}}_{gH}\mathfrak{D}^{\varsigma}_{+}\mathbb{S}(\mathfrak{r},\mathfrak{k})$ is *FFCgH*-differentiable in its $\mathbb{S}^{\circledast}-\mathbb{F}^{\circledast}$, we obtain

$$\mathsf{L}\left[{}^{\mathrm{CF}}_{gH}\mathfrak{D}^{\varsigma}_{+}\mathbb{S}(\mathfrak{r},\mathfrak{k})\right](\mathfrak{m}) = (-\mathfrak{m}^{\varsigma-2}\odot\mathbb{S}'(\mathfrak{r},0))\ominus\left\{(-\mathfrak{m}^{\varsigma}\odot\mathsf{L}[\mathbb{S}(\mathfrak{r},\mathfrak{k})](\mathfrak{m}))\ominus(-\mathfrak{m}^{\varsigma-1}\odot\mathbb{S}(\mathfrak{r},0))\right\}.$$

The results of cases (a), (c) and (d) can also be proved in the similar fashion, therefore left as an exercise. □

**Theorem 17.** Let $\mathbb{S}:(0,\mathfrak{u})\longrightarrow\natural_{\mathbb{R}}$ provided that $\mathbb{S}(\mathfrak{r},\mathfrak{k})\in\mathbb{C}^{\natural_{\mathbb{R}}}(0,\mathfrak{u})\cap L^{\natural_{\mathbb{R}}}(0,\mathfrak{u})$. Suppose that $\left[{}^{\mathrm{CF}}_{gH}\mathfrak{D}^{\varsigma}_{+}\mathbb{S}\right](\mathfrak{r},\mathfrak{k})$ and $\left[{}^{\mathrm{CF}}_{gH}\mathfrak{D}^{\gamma}_{+}\mathbb{S}\right](\mathfrak{r},\mathfrak{k})$ follow piecewise continuity on the interval $[0,\infty)$ and $\mathbb{S}$ is of exponential orders $\varsigma,\gamma$ respectively provided that $1<\varsigma\leq2$ and $0<\gamma\leq1$, then the following solutions of system (1) arises:

(a) If a FVF $\mathbb{S}(\mathfrak{r},\mathfrak{k}),{}^{\mathrm{CF}}_{gH}\mathfrak{D}^{\varsigma}_{+}\mathbb{S}(\mathfrak{r},\mathfrak{k})$ and ${}^{\mathrm{CF}}_{gH}\mathfrak{D}^{\gamma}_{+}\mathbb{S}(\mathfrak{r},\mathfrak{k})$ are *FFCgH*-differentiable in its $\mathbb{F}^{\circledast}-\mathbb{F}^{\circledast}$, then for exponential orders $\varsigma,\gamma$ such that $1<\varsigma\leq2$ and $0<\gamma\leq1$, the system (1) has solution of the form

$$\mathbb{S}(\mathfrak{r},\mathfrak{k}) = (\mathbb{S}_1,\mathbb{S}_2,\mathbb{S}_3)\odot\tilde{\ell}(\mathfrak{r})\odot\mathbb{E}_{\varsigma,\varsigma-\gamma,1}(\psi\mathfrak{k}^{\varsigma},\mathfrak{k}^{\varsigma-\gamma})\oplus(\mathbb{S}_1,\mathbb{S}_2,\mathbb{S}_3)\odot\tilde{j}(\mathfrak{r})\odot\mathfrak{k}\mathbb{E}_{\varsigma,\varsigma-\gamma,2}(\psi\mathfrak{k}^{\varsigma},\mathfrak{k}^{\varsigma-\gamma})\ominus$$

$$(\mathbb{S}_1,\mathbb{S}_2,\mathbb{S}_3)\odot\tilde{\ell}(\mathfrak{r})\odot\mathfrak{k}^{\varsigma-\gamma+1}\mathbb{E}_{\varsigma,\varsigma-\gamma,\varsigma-\gamma+1}(\psi\mathfrak{k}^{\varsigma},\mathfrak{k}^{\varsigma-\gamma})\ominus\left(\mathbb{I}^{1;\psi,1}_{\varsigma,\varsigma-\gamma,\varsigma}[(1+\psi)\odot\mathbb{S}^2(\mathfrak{r},\mathfrak{k})\ominus\mathbb{S}^3(\mathfrak{r},\mathfrak{k})]\right)(\mathfrak{k}). \qquad (40)$$

(b) If a FVF $\mathbb{S}(\mathfrak{r},\mathfrak{k})$ is *FFCgH*-differentiable in its $\mathbb{F}^{\circledast}-\mathbb{F}^{\circledast}$ and ${}^{\mathrm{CF}}_{gH}\mathfrak{D}^{\varsigma}_{+}\mathbb{S}(\mathfrak{r},\mathfrak{k}),{}^{\mathrm{CF}}_{gH}\mathfrak{D}^{\gamma}_{+}\mathbb{S}(\mathfrak{r},\mathfrak{k})$ are *FFCgH*-differentiable in its $\mathbb{S}^{\circledast}-\mathbb{F}^{\circledast}$, then for exponential orders $\varsigma,\gamma$ such that $1<\varsigma\leq2$ and $0<\gamma\leq1$, the system (1) has solution of the form

$$\mathbb{S}(\mathfrak{r},\mathfrak{k}) = (\mathbb{S}_1,\mathbb{S}_2,\mathbb{S}_3)\odot\tilde{j}(\mathfrak{r})\odot\mathfrak{k}\mathbb{E}_{\varsigma,\varsigma-\gamma,2}(\psi\mathfrak{k}^{\varsigma},\mathfrak{k}^{\varsigma-\gamma})\ominus(-1)(\mathbb{S}_1,\mathbb{S}_2,\mathbb{S}_3)\odot\tilde{\ell}(\mathfrak{r})\odot\mathbb{E}_{\varsigma,\varsigma-\gamma,1}$$

$$(\psi\mathfrak{k}^{\varsigma},\mathfrak{k}^{\varsigma-\gamma})\oplus(-1)(\mathbb{S}_1,\mathbb{S}_2,\mathbb{S}_3)\odot\tilde{\ell}(\mathfrak{r})\odot\mathfrak{k}^{\varsigma-\gamma}\mathbb{E}_{\varsigma,\varsigma-\gamma,\varsigma-\gamma+1}(\psi\mathfrak{k}^{\varsigma},\mathfrak{k}^{\varsigma-\gamma})\oplus(-1)(\mathbb{I}^{1;\psi,1}_{\varsigma,\varsigma-\gamma,\varsigma}[(1+\psi)$$

$$\odot\mathbb{S}^2(\mathfrak{r},\mathfrak{k})\ominus\mathbb{S}^3(\mathfrak{r},\mathfrak{k})])(\mathfrak{k}). \qquad (41)$$

(c) If a FVF $\mathbb{S}(\mathfrak{r},\mathfrak{k})$ is *FFCgH*-differentiable in its $\mathbb{S}^{\circledast}-\mathbb{F}^{\circledast}$ and ${}^{\mathrm{CF}}_{gH}\mathfrak{D}^{\varsigma}_{+}\mathbb{S}(\mathfrak{r},\mathfrak{k}),{}^{\mathrm{CF}}_{gH}\mathfrak{D}^{\gamma}_{+}\mathbb{S}(\mathfrak{r},\mathfrak{k})$ are *FFCgH*-differentiable in its $\mathbb{F}^{\circledast}-\mathbb{F}^{\circledast}$, then for exponential orders $\varsigma,\gamma$ such that $1<\varsigma\leq2$ and $0<\gamma\leq1$, the system (1) has solution of the form

$$\mathbb{S}(\mathfrak{r},\mathfrak{k}) = (\mathbb{S}_1,\mathbb{S}_2,\mathbb{S}_3)\odot\tilde{\ell}(\mathfrak{r})\odot\mathbb{E}_{\varsigma,\varsigma-\gamma,1}(\psi\mathfrak{k}^{\varsigma},\mathfrak{k}^{\varsigma-\gamma})\oplus(\mathbb{S}_1,\mathbb{S}_2,\mathbb{S}_3)\odot\tilde{j}(\mathfrak{r})\odot\mathfrak{k}\mathbb{E}_{\varsigma,\varsigma-\gamma,\varsigma-\gamma+1}$$

$$(\psi\mathfrak{k}^{\varsigma},\mathfrak{k}^{\varsigma-\gamma})\oplus(-1)(\mathbb{S}_1,\mathbb{S}_2,\mathbb{S}_3)\odot\tilde{\ell}(\mathfrak{r})\odot\mathfrak{k}^{\varsigma-\gamma}\mathbb{E}_{\varsigma,\varsigma-\gamma,\varsigma-\gamma+1}(\psi\mathfrak{k}^{\varsigma},\mathfrak{k}^{\varsigma-\gamma})\oplus(-1)(\mathbb{I}^{1;\psi,1}_{\varsigma,\varsigma-\gamma,\varsigma}[(1+\psi)$$

$$\odot\mathbb{S}^2(\mathfrak{r},\mathfrak{k})\ominus\mathbb{S}^3(\mathfrak{r},\mathfrak{k})])(\mathfrak{k}). \qquad (42)$$

(d) If a FVF $\mathbb{S}(\mathfrak{r},\mathfrak{k}),{}^{\mathrm{CF}}_{gH}\mathfrak{D}^{\gamma}_{+}\mathbb{S}(\mathfrak{r},\mathfrak{k})$ and ${}^{\mathrm{CF}}_{gH}\mathfrak{D}^{\varsigma}_{+}\mathbb{S}(\mathfrak{r},\mathfrak{k})$ are *FFCgH*-differentiable in its $\mathbb{S}^{\circledast}-\mathbb{F}^{\circledast}$, then for exponential orders $\varsigma,\gamma$ such that $1<\varsigma\leq2$ and $0<\gamma\leq1$, the system (1) has solution of the form

$$\mathbb{S}(\mathfrak{r},\mathfrak{k}) = (\mathbb{S}_1,\mathbb{S}_2,\mathbb{S}_3) \odot \tilde{\ell}(\mathfrak{r}) \odot \mathbb{E}_{\varsigma,\varsigma-\gamma,1}(\psi\mathfrak{k}^\varsigma,\mathfrak{k}^{\varsigma-\gamma}) \ominus (-1)(\mathbb{S}_1,\mathbb{S}_2,\mathbb{S}_3) \odot \tilde{j}(\mathfrak{r}) \odot \mathfrak{k}\mathbb{E}_{\varsigma,\varsigma-\gamma,\varsigma-\gamma+2}$$

$$(\psi\mathfrak{k}^\varsigma,\mathfrak{k}^{\varsigma-\gamma}) \ominus (\mathbb{S}_1,\mathbb{S}_2,\mathbb{S}_3) \odot \tilde{\ell}(\mathfrak{r}) \odot \mathfrak{k}^{\varsigma-\gamma}\mathbb{E}_{\varsigma,\varsigma-\gamma,\varsigma-\gamma+1}(\psi\mathfrak{k}^\varsigma,\mathfrak{k}^{\varsigma-\gamma}) \ominus \big(\mathbb{I}^{1;\psi,1}_{\varsigma,\varsigma-\gamma,\varsigma}[(1+\psi)$$

$$\odot \mathbb{S}^2(\mathfrak{r},\mathfrak{k}) \ominus \mathbb{S}^3(\mathfrak{r},\mathfrak{k})]\big)(\mathfrak{k}). \tag{43}$$

(e) If a FVF $\mathbb{S}(\mathfrak{r},\mathfrak{k}), {}^{\mathrm{CF}}_{gH}\mathfrak{D}^\gamma_+\mathbb{S}(\mathfrak{r},\mathfrak{k})$ are *FFCgH*-differentiable in its $\mathbb{F}^* - \mathbb{F}^*$ and ${}^{\mathrm{CF}}_{gH}\mathfrak{D}^\varsigma_+\mathbb{S}(\mathfrak{r},\mathfrak{k})$ is *FFCgH*-differentiable in its $\mathbb{S}^\circledR - \mathbb{F}^*$, then for exponential orders $\varsigma,\gamma$ such that $1 < \varsigma \le 2$ and $0 < \gamma \le 1$, the system (1) has solution of the form

$$\mathbb{S}(\mathfrak{r},\mathfrak{k}) = (\mathbb{S}_1,\mathbb{S}_2,\mathbb{S}_3) \odot \tilde{j}(\mathfrak{r}) \odot \mathfrak{k}\mathbb{E}_{\varsigma,\varsigma-\gamma,2}(\psi\mathfrak{k}^\varsigma,\mathfrak{k}^{\varsigma-\gamma}) \oplus (-1)(\mathbb{S}_1,\mathbb{S}_2,\mathbb{S}_3) \odot \tilde{\ell}(\mathfrak{r}) \odot \mathbb{E}_{\varsigma,\varsigma-\gamma,1}$$

$$(\psi\mathfrak{k}^\varsigma,\mathfrak{k}^{\varsigma-\gamma}) \ominus (\mathbb{S}_1,\mathbb{S}_2,\mathbb{S}_3) \odot \tilde{\ell}(\mathfrak{r}) \odot \mathfrak{k}^{\varsigma-\gamma}\mathbb{E}_{\varsigma,\varsigma-\gamma,\varsigma-\gamma+1}(\psi\mathfrak{k}^\varsigma,\mathfrak{k}^{\varsigma-\gamma}) \ominus \big(\mathbb{I}^{1;\psi,1}_{\varsigma,\varsigma-\gamma,\varsigma}[(1+\psi)$$

$$\odot \mathbb{S}^2(\mathfrak{r},\mathfrak{k}) \ominus \mathbb{S}^3(\mathfrak{r},\mathfrak{k})]\big)(\mathfrak{k}). \tag{44}$$

(f) If a FVF $\mathbb{S}(\mathfrak{r},\mathfrak{k}), {}^{\mathrm{CF}}_{gH}\mathfrak{D}^\varsigma_+\mathbb{S}(\mathfrak{r},\mathfrak{k})$ are *FFCgH*-differentiable in its $\mathbb{F}^* - \mathbb{F}^*$ and ${}^{\mathrm{CF}}_{gH}\mathfrak{D}^\gamma_+\mathbb{S}(\mathfrak{r},\mathfrak{k})$ is *FFCgH*-differentiable in its $\mathbb{S}^\circledR - \mathbb{F}^*$, then for exponential orders $\varsigma,\gamma$ such that $1 < \varsigma \le 2$ and $0 < \gamma \le 1$, the system (1) has solution of the form

$$\mathbb{S}(\mathfrak{r},\mathfrak{k}) = (\mathbb{S}_1,\mathbb{S}_2,\mathbb{S}_3) \odot \tilde{\ell}(\mathfrak{r}) \odot \mathbb{E}_{\varsigma,\varsigma-\gamma,1}(\psi\mathfrak{k}^\varsigma,-\mathfrak{k}^{\varsigma-\gamma}) \oplus (\mathbb{S}_1,\mathbb{S}_2,\mathbb{S}_3) \odot \tilde{j}(\mathfrak{r}) \odot \mathfrak{k}\mathbb{E}_{\varsigma,\varsigma-\gamma,2}$$

$$(\psi\mathfrak{k}^\varsigma,-\mathfrak{k}^{\varsigma-\gamma}) \oplus (\mathbb{S}_1,\mathbb{S}_2,\mathbb{S}_3) \odot \tilde{\ell}(\mathfrak{r}) \odot \mathfrak{k}^{\varsigma-\gamma}\mathbb{E}_{\varsigma,\varsigma-\gamma,\varsigma-\gamma+1}(\psi\mathfrak{k}^\varsigma,-\mathfrak{k}^{\varsigma-\gamma}) \ominus \big(\mathbb{I}^{1;\psi,1}_{\varsigma,\varsigma-\gamma,\varsigma}[(1+\psi)$$

$$\odot \mathbb{S}^2(\mathfrak{r},\mathfrak{k}) \ominus \mathbb{S}^3(\mathfrak{r},\mathfrak{k})]\big)(\mathfrak{k}). \tag{45}$$

**Proof 18.** Suppose that $\mathbb{S}(\mathfrak{r},\mathfrak{k}) \in \mathbb{C}^{\natural_\mathbb{R}}(0,\mathfrak{u}) \cap L^{\natural_\mathbb{R}}(0,\mathfrak{u})$. Consider the FFGFH-NDE problem as follows:

$$\begin{cases} \big[{}^{\mathrm{CF}}_{gH}\mathfrak{D}^\varsigma_+\mathbb{S}\big](\mathfrak{r},\mathfrak{k}) \ominus \big[{}^{\mathrm{CF}}_{gH}\mathfrak{D}^\gamma_+\mathbb{S}\big](\mathfrak{r},\mathfrak{k}) \oplus \mathbb{S}(\mathfrak{r},\mathfrak{k})[(1-\mathbb{S}(\mathfrak{r},\mathfrak{k}))(\mathbb{S}(\mathfrak{r},\mathfrak{k})-\psi)] = 0, \\ \\ \mathbb{S}(\mathfrak{r},0) = (\mathbb{S}_1,\mathbb{S}_2,\mathbb{S}_3) \odot \tilde{\ell}(\mathfrak{r}), \\ \\ \mathbb{S}'(\mathfrak{r},0) = (\mathbb{S}_1,\mathbb{S}_2,\mathbb{S}_3) \odot \tilde{j}(\mathfrak{r}), \end{cases} \tag{46}$$

Now applying $\mathbb{FFLT}$ to problem (46), we have

$$\mathsf{L}\Big[\big[{}^{\mathrm{CF}}_{gH}\mathfrak{D}^\varsigma_+\mathbb{S}\big](\mathfrak{r},\mathfrak{k}) \ominus \big[{}^{\mathrm{CF}}_{gH}\mathfrak{D}^\gamma_+\mathbb{S}\big](\mathfrak{r},\mathfrak{k}) \oplus \mathbb{S}(\mathfrak{r},\mathfrak{k})[(1-\mathbb{S}(\mathfrak{r},\mathfrak{k}))(\mathbb{S}(\mathfrak{r},\mathfrak{k})-\psi)]\Big] = 0 \tag{47}$$

By Lemma 1, the aforementioned Eq (47) reduces as follows:

$$\mathsf{L}\big[{}^{\mathrm{CF}}_{gH}\mathfrak{D}^\varsigma_+\mathbb{S}\big](\mathfrak{r},\mathfrak{k}) \ominus \mathsf{L}\big[{}^{\mathrm{CF}}_{gH}\mathfrak{D}^\gamma_+\mathbb{S}\big](\mathfrak{r},\mathfrak{k}) \oplus \mathsf{L}\Big[\mathbb{S}(\mathfrak{r},\mathfrak{k})[(1-\mathbb{S}(\mathfrak{r},\mathfrak{k}))(\mathbb{S}(\mathfrak{r},\mathfrak{k})-\psi)]\Big] = 0. \tag{48}$$

(a) Suppose that FVF $\mathbb{S}(\mathfrak{r},\mathfrak{k}), {}_{gH}^{\mathrm{CF}}\mathfrak{D}_+^\gamma \mathbb{S}(\mathfrak{r},\mathfrak{k})$ and ${}_{gH}^{\mathrm{CF}}\mathfrak{D}_+^\varsigma \mathbb{S}(\mathfrak{r},\mathfrak{k})$ are *FFCgH*-differentiable in its $\mathbb{F}^\divideontimes - \mathbb{F}^\divideontimes$, then from Eqs (25) and (48), we have

$$\left\{\mathfrak{m}^\varsigma \odot L[\mathbb{S}(\mathfrak{r},\mathfrak{k})](\mathfrak{m}) \ominus \mathfrak{m}^{\varsigma-1} \odot \mathbb{S}(\mathfrak{r},0)\right\} \ominus (\mathfrak{m}^{\varsigma-2} \odot \mathbb{S}'(\mathfrak{r},0)) \ominus \left\{\mathfrak{m}^\gamma \odot L[\mathbb{S}(\mathfrak{r},\mathfrak{k})](\mathfrak{m}) \ominus \right.$$

$$\left.\mathfrak{m}^{\gamma-1} \odot \mathbb{S}(\mathfrak{r},0)\right\} \oplus L\left[-\psi \odot \mathbb{S}(\mathfrak{r},\mathfrak{k}) \oplus \mathbb{S}^*(\mathfrak{r},\mathfrak{k})\right] = 0, \tag{49}$$

where $\mathbb{S}^*(\mathfrak{r},\mathfrak{k}) = (1+\psi) \odot \mathbb{S}^2(\mathfrak{r},\mathfrak{k}) \ominus \mathbb{S}^3(\mathfrak{r},\mathfrak{k})$ is a nonlinear function. From lemma 1 and Theorem 8, Eq (49) implies

$$L[\mathbb{S}(\mathfrak{r},\mathfrak{k})](\mathfrak{m}) = \frac{\mathfrak{m}^{\varsigma-1} \odot (\mathbb{S}_1,\mathbb{S}_2,\mathbb{S}_3) \odot \tilde{\ell}(\mathfrak{r})}{\mathfrak{m}^\varsigma - \mathfrak{m}^\gamma - \psi} \oplus \frac{(\mathfrak{m}^{\varsigma-2} \odot (\mathbb{S}_1,\mathbb{S}_2,\mathbb{S}_3) \odot \tilde{j}(\mathfrak{r}))}{\mathfrak{m}^\varsigma - \mathfrak{m}^\gamma - \psi} \ominus$$

$$\frac{(\mathfrak{m}^{\gamma-1} \odot (\mathbb{S}_1,\mathbb{S}_2,\mathbb{S}_3) \odot \tilde{\ell}(\mathfrak{r}))}{\mathfrak{m}^\varsigma - \mathfrak{m}^\gamma - \psi} \ominus \frac{1}{\mathfrak{m}^\varsigma - \mathfrak{m}^\gamma - \psi} \odot L\left[\mathbb{S}^*(\mathfrak{r},\mathfrak{k})\right](\mathfrak{m}). \tag{50}$$

From Theorem 9, Eq (50) implies

$$L[\mathbb{S}(\mathfrak{r},\mathfrak{k})](\mathfrak{m}) = \frac{\mathfrak{m}^{\varsigma-1} \odot (\mathbb{S}_1,\mathbb{S}_2,\mathbb{S}_3) \odot \tilde{\ell}(\mathfrak{r})}{\mathfrak{m}^\varsigma - \mathfrak{m}^\gamma - \psi} \oplus \frac{(\mathfrak{m}^{\varsigma-2} \odot (\mathbb{S}_1,\mathbb{S}_2,\mathbb{S}_3) \odot \tilde{j}(\mathfrak{r}))}{\mathfrak{m}^\varsigma - \mathfrak{m}^\gamma - \psi} \ominus$$

$$\frac{(\mathfrak{m}^{\gamma-1} \odot (\mathbb{S}_1,\mathbb{S}_2,\mathbb{S}_3) \odot \tilde{\ell}(\mathfrak{r}))}{\mathfrak{m}^\varsigma - \mathfrak{m}^\gamma - \psi} \ominus L\left[\mathfrak{k}^{\varsigma-1}\mathbb{E}_{\varsigma,\varsigma-\gamma,\varsigma}(\psi\mathfrak{k}^\varsigma, \mathfrak{k}^{\varsigma-\gamma})\right](\mathfrak{m}) \odot L\left[\mathbb{S}^*(\mathfrak{r},\mathfrak{k})\right](\mathfrak{m}). \tag{51}$$

Applying Theorem 8 on Eq (51), we obtain

$$L[\mathbb{S}(\mathfrak{r},\mathfrak{k})](\mathfrak{m}) = \frac{\mathfrak{m}^{\varsigma-1} \odot (\mathbb{S}_1,\mathbb{S}_2,\mathbb{S}_3) \odot \tilde{\ell}(\mathfrak{r})}{\mathfrak{m}^\varsigma - \mathfrak{m}^\gamma - \psi} \oplus \frac{(\mathfrak{m}^{\varsigma-2} \odot (\mathbb{S}_1,\mathbb{S}_2,\mathbb{S}_3) \odot \tilde{j}(\mathfrak{r}))}{\mathfrak{m}^\varsigma - \mathfrak{m}^\gamma - \psi} \ominus (\mathfrak{m}^{\gamma-1}\odot$$

$$\frac{(\mathbb{S}_1,\mathbb{S}_2,\mathbb{S}_3) \odot \tilde{\ell}(\mathfrak{r}))}{\mathfrak{m}^\varsigma - \mathfrak{m}^\gamma - \psi} \ominus L\left[\int_0^{\mathfrak{k}} (\mathfrak{k}-\mathfrak{s})^{\varsigma-1}\mathbb{E}_{\varsigma,\varsigma-\gamma,\varsigma}(\psi(\mathfrak{k}-\mathfrak{s})^\varsigma, (\mathfrak{k}-\mathfrak{s})^{\varsigma-\gamma}) \odot \mathbb{S}^*(\mathfrak{r},\mathfrak{s})d\mathfrak{s}\right](\mathfrak{m}). \tag{52}$$

Using $\mathbb{IFFLT}$ and Theorem 9, we deduce

$$\mathbb{S}(\mathfrak{r},\mathfrak{k}) = (\mathbb{S}_1,\mathbb{S}_2,\mathbb{S}_3) \odot \tilde{\ell}(\mathfrak{r}) \odot \mathbb{E}_{\varsigma,\varsigma-\gamma,1}(\psi\mathfrak{k}^\varsigma, \mathfrak{k}^{\varsigma-\gamma}) \oplus (\mathbb{S}_1,\mathbb{S}_2,\mathbb{S}_3) \odot \tilde{j}(\mathfrak{r}) \odot \mathfrak{k}\mathbb{E}_{\varsigma,\varsigma-\gamma,2}(\psi\mathfrak{k}^\varsigma, \mathfrak{k}^{\varsigma-\gamma})$$

$$\ominus (\mathbb{S}_1,\mathbb{S}_2,\mathbb{S}_3) \odot \tilde{\ell}(\mathfrak{r}) \odot \mathfrak{k}^{\varsigma-\gamma+1}\mathbb{E}_{\varsigma,\varsigma-\gamma,\varsigma-\gamma+1}(\psi\mathfrak{k}^\varsigma, \mathfrak{k}^{\varsigma-\gamma}) \ominus \int_0^{\mathfrak{k}} (\mathfrak{k}-\mathfrak{s})^{\varsigma-1}\mathbb{E}_{\varsigma,\varsigma-\gamma},$$

$$\varsigma(\psi(\mathfrak{k}-\mathfrak{s})^\varsigma, (\mathfrak{k}-\mathfrak{s})^{\varsigma-\gamma}) \odot \mathbb{S}^*(\mathfrak{r},\mathfrak{s})d\mathfrak{s}. \tag{53}$$

The most general form of the solution is obtained by using [Theorem 10] in Eq [(53)] as follows

$$\mathbb{S}(\mathfrak{r},\mathfrak{k}) = (\mathbb{S}_1,\mathbb{S}_2,\mathbb{S}_3) \odot \tilde{\ell}(\mathfrak{r}) \odot \mathbb{E}_{\varsigma,\varsigma-\gamma,1}(\psi \mathfrak{k}^\varsigma, \mathfrak{k}^{\varsigma-\gamma}) \oplus (\mathbb{S}_1,\mathbb{S}_2,\mathbb{S}_3) \odot \tilde{j}(\mathfrak{r}) \odot \mathfrak{k}\mathbb{E}_{\varsigma,\varsigma-\gamma,2}(\psi \mathfrak{k}^\varsigma, \mathfrak{k}^{\varsigma-\gamma}) \ominus$$

$$(\mathbb{S}_1,\mathbb{S}_2,\mathbb{S}_3) \odot \tilde{\ell}(\mathfrak{r}) \odot \mathfrak{k}^{\varsigma-\gamma+1}\mathbb{E}_{\varsigma,\varsigma-\gamma,\varsigma-\gamma+1}(\psi \mathfrak{k}^\varsigma, \mathfrak{k}^{\varsigma-\gamma}) \ominus \big(\mathbb{I}^{1;\psi,1}_{\varsigma,\varsigma-\gamma,\varsigma}[(1+\psi)\odot \mathbb{S}^2(\mathfrak{r},\mathfrak{k}) \ominus \mathbb{S}^3(\mathfrak{r},\mathfrak{k})]\big)(\mathfrak{k}).$$

The rest of the parts $(b),(c),(d),(e)$, and $(f)$ can be proved in a similar way as mentioned in Part $(a)$.  □

**Theorem 19.** Let $\mathbb{S}:(0,\mathfrak{u}) \longrightarrow \natural_\mathbb{R}$ provided that $\mathbb{S}(\mathfrak{r},\mathfrak{k}) \in \mathbb{C}^{\natural_\mathbb{R}}(0,\mathfrak{u}) \cap L^{\natural_\mathbb{R}}(0,\mathfrak{u})$. Suppose that $\big[{}^{\mathrm{CF}}_{gH}\mathfrak{D}^\varsigma_+\mathbb{S}\big](\mathfrak{r},\mathfrak{k})$ and $\big[{}^{\mathrm{CF}}_{gH}\mathfrak{D}^\gamma_+\mathbb{S}\big](\mathfrak{r},\mathfrak{k})$ follow piecewise continuity on the interval $[0,\infty)$ and $\mathbb{S}$ is of exponential orders $\varsigma,\gamma$ respectively where $1<\varsigma\le 2; 0<\gamma\le 1$ provided that $\psi=1$, then the following solutions of system [(1)] arises:

(a) If a FVF $\mathbb{S}(\mathfrak{r},\mathfrak{k}), {}^{\mathrm{CF}}_{gH}\mathfrak{D}^\gamma_+\mathbb{S}(\mathfrak{r},\mathfrak{k})$ and ${}^{\mathrm{CF}}_{gH}\mathfrak{D}^\varsigma_+\mathbb{S}(\mathfrak{r},\mathfrak{k})$ are *FFCgH-differentiable* in its $\mathbb{F}^* - \mathbb{F}^*$, then for exponential orders $\varsigma,\gamma$ where $1<\varsigma\le 2; 0<\gamma\le 1$, the system [(1)] has solution of the form

$$\mathbb{S}(\mathfrak{r},\mathfrak{k}) = (\mathbb{S}_1,\mathbb{S}_2,\mathbb{S}_3) \odot \tilde{\ell}(\mathfrak{r}) \odot \mathbb{E}_{\varsigma,\varsigma-\gamma,1}(\mathfrak{k}^\varsigma, \mathfrak{k}^{\varsigma-\gamma}) \oplus (\mathbb{S}_1,\mathbb{S}_2,\mathbb{S}_3) \odot \tilde{j}(\mathfrak{r}) \odot \mathfrak{k}\mathbb{E}_{\varsigma,\varsigma-\gamma,2}(\mathfrak{k}^\varsigma, \mathfrak{k}^{\varsigma-\gamma}) \ominus$$

$$(\mathbb{S}_1,\mathbb{S}_2,\mathbb{S}_3) \odot \tilde{\ell}(\mathfrak{r}) \odot \mathfrak{k}^{\varsigma-\gamma}\mathbb{E}_{\varsigma,\varsigma-\gamma,\varsigma-\gamma+1}(\mathfrak{k}^\varsigma, \mathfrak{k}^{\varsigma-\gamma}) \ominus (-1)\big(\mathbb{I}^{1;-1,1}_{\varsigma,\varsigma-\gamma,\varsigma}[\mathbb{S}^3(\mathfrak{r},\mathfrak{k}) \ominus 2\mathbb{S}^2(\mathfrak{r},\mathfrak{k})]\big)(\mathfrak{k}). \qquad (54)$$

(b) If a FVF $\mathbb{S}(\mathfrak{r},\mathfrak{k})$ is *FFCgH-differentiable* in its $\mathbb{F}^* - \mathbb{F}^*$ and ${}^{\mathrm{CF}}_{gH}\mathfrak{D}^\varsigma_+\mathbb{S}(\mathfrak{r},\mathfrak{k}), {}^{\mathrm{CF}}_{gH}\mathfrak{D}^\gamma_+\mathbb{S}(\mathfrak{r},\mathfrak{k})$ are *FFCgH-differentiable* in its $\mathbb{S}^\circledast - \mathbb{F}^*$, then for exponential orders $\varsigma,\gamma$ where $1<\varsigma\le 2; 0<\gamma\le 1$, the system [(1)] has solution of the form

$$\mathbb{S}(\mathfrak{r},\mathfrak{k}) = (\mathbb{S}_1,\mathbb{S}_2,\mathbb{S}_3) \odot \tilde{j}(\mathfrak{r}) \odot \mathfrak{k} \odot \mathbb{E}_{\varsigma,\varsigma-\gamma,2}(\mathfrak{k}^\varsigma, \mathfrak{k}^{\varsigma-\gamma}) \ominus (-1)(\mathbb{S}_1,\mathbb{S}_2,\mathbb{S}_3) \odot \tilde{\ell}(\mathfrak{r}) \odot \mathbb{E}_{\varsigma,\varsigma-\gamma,1}$$

$$(\mathfrak{k}^\varsigma, \mathfrak{k}^{\varsigma-\gamma}) \oplus (-1)(\mathbb{S}_1,\mathbb{S}_2,\mathbb{S}_3) \odot \tilde{\ell}(\mathfrak{r}) \odot \mathfrak{k}^{\varsigma-\gamma}\mathbb{E}_{\varsigma,\varsigma-\gamma,\varsigma-\gamma+1}(\mathfrak{k}^\varsigma, \mathfrak{k}^{\varsigma-\gamma}) \ominus (-1)\big(\mathbb{I}^{1;1,1}_{\varsigma,\varsigma-\gamma,\varsigma}[\mathbb{S}^3(\mathfrak{r},\mathfrak{k}) \ominus$$

$$2\mathbb{S}^2(\mathfrak{r},\mathfrak{k})]\big)(\mathfrak{k}). \qquad (55)$$

(c) If a FVF $\mathbb{S}(\mathfrak{r},\mathfrak{k})$ is *FFCgH-differentiable* in its $\mathbb{S}^\circledast - \mathbb{F}^*$ and ${}^{\mathrm{CF}}_{gH}\mathfrak{D}^\varsigma_+\mathbb{S}(\mathfrak{r},\mathfrak{k}), {}^{\mathrm{CF}}_{gH}\mathfrak{D}^\gamma_+\mathbb{S}(\mathfrak{r},\mathfrak{k})$ are *FFCgH-differentiable* in its $\mathbb{F}^* - \mathbb{F}^*$, then for exponential orders $\varsigma,\gamma$ where $1<\varsigma\le 2; 0<\gamma\le 1$, then the system [(1)] has solution of the form

$$\mathbb{S}(\mathfrak{r},\mathfrak{k}) = (\mathbb{S}_1,\mathbb{S}_2,\mathbb{S}_3) \odot \tilde{j}(\mathfrak{r}) \odot \mathfrak{k} \odot \mathbb{E}_{\varsigma,\varsigma-\gamma,2}(\mathfrak{k}^\varsigma, \mathfrak{k}^{\varsigma-\gamma}) \oplus (\mathbb{S}_1,\mathbb{S}_2,\mathbb{S}_3) \odot \tilde{\ell}(\mathfrak{r}) \odot \mathbb{E}_{\varsigma,\varsigma-\gamma,1}(\mathfrak{k}^\varsigma, \mathfrak{k}^{\varsigma-\gamma}) \oplus$$

$$(-1)(\mathbb{S}_1,\mathbb{S}_2,\mathbb{S}_3) \odot \tilde{\ell}(\mathfrak{r}) \odot \mathfrak{k}^{\varsigma-\gamma}\mathbb{E}_{\varsigma,\varsigma-\gamma,\varsigma-\gamma+1}(\mathfrak{k}^\varsigma, \mathfrak{k}^{\varsigma-\gamma}) \ominus (-1)\big(\mathbb{I}^{1;1,1}_{\varsigma,\varsigma-\gamma,\varsigma}[\mathbb{S}^3(\mathfrak{r},\mathfrak{k}) \ominus 2\mathbb{S}^2(\mathfrak{r},\mathfrak{k})]\big)(\mathfrak{k}). \qquad (56)$$

(d) If a FVF $\mathbb{S}(\mathfrak{r},\mathfrak{k}), {}^{\mathrm{CF}}_{gH}\mathfrak{D}^\gamma_+\mathbb{S}(\mathfrak{r},\mathfrak{k})$ and ${}^{\mathrm{CF}}_{gH}\mathfrak{D}^\varsigma_+\mathbb{S}(\mathfrak{r},\mathfrak{k})$ are *FFCgH-differentiable* in its $\mathbb{S}^\circledast - \mathbb{F}^*$, then for exponential orders $\varsigma,\gamma$ where $1<\varsigma\le 2; 0<\gamma\le 1$, then the system [(1)] has solution of the form

$$\mathbb{S}(\mathbf{r}, \mathfrak{k}) = (-1)(\mathbb{S}_1, \mathbb{S}_2, \mathbb{S}_3) \odot \tilde{\ell}(\mathbf{r}) \odot \mathbb{E}_{\varsigma, \varsigma-\gamma, 1}(\mathfrak{k}^\varsigma, \mathfrak{k}^{\varsigma-\gamma}) \oplus (\mathbb{S}_1, \mathbb{S}_2, \mathbb{S}_3) \odot \tilde{j}(\mathbf{r}) \odot \mathfrak{k} \mathbb{E}_{\varsigma, \varsigma-\gamma, 2}(\mathfrak{k}^\varsigma,$$

$$\mathfrak{k}^{\varsigma-\gamma}) \ominus (-1)(\mathbb{S}_1, \mathbb{S}_2, \mathbb{S}_3) \odot \tilde{\ell}(\mathbf{r}) \odot \mathfrak{k}^{\varsigma-\gamma} \mathbb{E}_{\varsigma, \varsigma-\gamma, \varsigma-\gamma+1}((-1)\mathfrak{k}^\varsigma, \mathfrak{k}^{\varsigma-\gamma}) \ominus (-1)(\mathbb{I}_{\varsigma, \varsigma-\gamma, \varsigma}^{1;1,1}[\mathbb{S}^3(\mathbf{r}, \mathfrak{k})$$

$$\ominus 2\mathbb{S}^2(\mathbf{r}, \mathfrak{k})])(\mathfrak{k}). \tag{57}$$

(e) If a FVF $\mathbb{S}(\mathbf{r}, \mathfrak{k}), {}^{\mathrm{CF}}_{gH}\mathfrak{D}_+^\gamma \mathbb{S}(\mathbf{r}, \mathfrak{k})$ are *FFCgH*-differentiable in its $\mathbb{F}^* - \mathbb{F}^*$ and ${}^{\mathrm{CF}}_{gH}\mathfrak{D}_+^\varsigma \mathbb{S}(\mathbf{r}, \mathfrak{k})$ is *FFCgH*-differentiable in its $\mathbb{S}^\circledast - \mathbb{F}^*$, then for exponential orders $\varsigma, \gamma$ such that $1 < \varsigma \leq 2$ and $0 < \gamma \leq 1$, the system (1) has solution of the form

$$\mathbb{S}(\mathbf{r}, \mathfrak{k}) = (\mathbb{S}_1, \mathbb{S}_2, \mathbb{S}_3) \odot \tilde{j}(\mathbf{r}) \odot \mathfrak{k} \mathbb{E}_{\varsigma, \varsigma-\gamma, 2}(\mathfrak{k}^\varsigma, \mathfrak{k}^{\varsigma-\gamma}) \oplus (-1)(\mathbb{S}_1, \mathbb{S}_2, \mathbb{S}_3) \odot \tilde{\ell}(\mathbf{r}) \odot \mathbb{E}_{\varsigma, \varsigma-\gamma, 1}(\mathfrak{k}^\varsigma,$$

$$\mathfrak{k}^{\varsigma-\gamma}) \ominus (\mathbb{S}_1, \mathbb{S}_2, \mathbb{S}_3) \odot \tilde{\ell}(\mathbf{r}) \odot \mathfrak{k}^{\varsigma-\gamma} \mathbb{E}_{\varsigma, \varsigma-\gamma, \varsigma-\gamma+1}(\mathfrak{k}^\varsigma, \mathfrak{k}^{\varsigma-\gamma}) \ominus (\mathbb{I}_{\varsigma, \varsigma-\gamma, \varsigma}^{1;1,1}[2 \odot \mathbb{S}^2(\mathbf{r}, \mathfrak{k}) \ominus \mathbb{S}^3(\mathbf{r}, \mathfrak{k})])(\mathfrak{k}). \tag{58}$$

(f) If a FVF $\mathbb{S}(\mathbf{r}, \mathfrak{k}), {}^{\mathrm{CF}}_{gH}\mathfrak{D}_+^\varsigma \mathbb{S}(\mathbf{r}, \mathfrak{k})$ are *FFCgH*-differentiable in its $\mathbb{F}^* - \mathbb{F}^*$ and ${}^{\mathrm{CF}}_{gH}\mathfrak{D}_+^\gamma \mathbb{S}(\mathbf{r}, \mathfrak{k})$ is *FFCgH*-differentiable in its $\mathbb{S}^\circledast - \mathbb{F}^*$, then for exponential orders $\varsigma, \gamma$ such that $1 < \varsigma \leq 2$ and $0 < \gamma \leq 1$, the system (1) has solution of the form

$$\mathbb{S}(\mathbf{r}, \mathfrak{k}) = (\mathbb{S}_1, \mathbb{S}_2, \mathbb{S}_3) \odot \tilde{\ell}(\mathbf{r}) \odot \mathbb{E}_{\varsigma, \varsigma-\gamma, 1}(\psi \mathfrak{k}^\varsigma, -\mathfrak{k}^{\varsigma-\gamma}) \oplus (\mathbb{S}_1, \mathbb{S}_2, \mathbb{S}_3) \odot \tilde{j}(\mathbf{r}) \odot \mathfrak{k} \mathbb{E}_{\varsigma, \varsigma-\gamma, 2}$$

$$(\mathfrak{k}^\varsigma, -\mathfrak{k}^{\varsigma-\gamma}) \oplus (\mathbb{S}_1, \mathbb{S}_2, \mathbb{S}_3) \odot \tilde{\ell}(\mathbf{r}) \odot \mathfrak{k}^{\varsigma-\gamma} \mathbb{E}_{\varsigma, \varsigma-\gamma, \varsigma-\gamma+1}(\mathfrak{k}^\varsigma, -\mathfrak{k}^{\varsigma-\gamma}) \ominus (\mathbb{I}_{\varsigma, \varsigma-\gamma, \varsigma}^{1;1,1}[2 \odot \mathbb{S}^2(\mathbf{r}, \mathfrak{k})$$

$$\ominus \mathbb{S}^3(\mathbf{r}, \mathfrak{k})])(\mathfrak{k}). \tag{59}$$

**Proof 20.** This theorem can be proved on the similar way as Theorem 17 with the condition that $\psi = 1$. $\qquad\square$

**Theorem 21.** Let $\mathbb{S} : (0, \mathfrak{u}) \longrightarrow \natural_\mathbb{R}$ provided that $\mathbb{S}(\mathbf{r}, \mathfrak{k}) \in \mathbb{C}^{\natural_\mathbb{R}}(0, \mathfrak{u}) \cap L^{\natural_\mathbb{R}}(0, \mathfrak{u})$. Suppose that $[{}^{\mathrm{CF}}_{gH}\mathfrak{D}_+^\varsigma \mathbb{S}](\mathbf{r}, \mathfrak{k})$ and $[{}^{\mathrm{CF}}_{gH}\mathfrak{D}_+^\gamma \mathbb{S}](\mathbf{r}, \mathfrak{k})$ follow piecewise continuity on the interval $[0, \infty)$ and $\mathbb{S}$ is of exponential orders $\varsigma, \gamma$ respectively where $1 < \varsigma \leq 2; 0 < \gamma \leq 1$ provided that $\psi = 0$, then the following solutions of system (1) arises:

(a) If a FVF $\mathbb{S}(\mathbf{r}, \mathfrak{k}), {}^{\mathrm{CF}}_{gH}\mathfrak{D}_+^\gamma \mathbb{S}(\mathbf{r}, \mathfrak{k})$ and ${}^{\mathrm{CF}}_{gH}\mathfrak{D}_+^\varsigma \mathbb{S}(\mathbf{r}, \mathfrak{k})$ are *FFCgH*-differentiable in its $\mathbb{F}^* - \mathbb{F}^*$, then for exponential orders $\varsigma, \gamma$ where $1 < \varsigma \leq 2; 0 < \gamma \leq 1$, the system (1) has solution of the form

$$\mathbb{S}(\mathfrak{r},\mathfrak{k}) = (\mathbb{S}_1,\mathbb{S}_2,\mathbb{S}_3) \odot \tilde{\ell}(\mathfrak{r}) \odot \mathbb{E}_{\varsigma,\varsigma-\gamma,1}(0,\mathfrak{k}^{\varsigma-\gamma}) \ominus (\mathbb{S}_1,\mathbb{S}_2,\mathbb{S}_3) \odot \tilde{\ell}(\mathfrak{r}) \odot \mathfrak{k}^{\varsigma-\gamma} \odot \mathbb{E}_{\varsigma,\varsigma-\gamma,\varsigma-\gamma+1}$$

$$(0,\mathfrak{k}^{\varsigma-\gamma}) \oplus (\mathbb{S}_1,\mathbb{S}_2,\mathbb{S}_3) \odot \tilde{j}(\mathfrak{r}) \odot \mathfrak{k} \odot \mathbb{E}_{\varsigma,\varsigma-\gamma,2}(0,\mathfrak{k}^{\varsigma-\gamma}) \oplus \left(\mathbb{I}_{\varsigma,\varsigma-\gamma,\varsigma}^{1;0,1}[\mathbb{S}^3(\mathfrak{r},\mathfrak{k}) \ominus \mathbb{S}^2(\mathfrak{r},\mathfrak{k})]\right)(\mathfrak{k}). \tag{60}$$

(b) If a FVF $\mathbb{S}(\mathfrak{r},\mathfrak{k})$ is *FFCgH*-differentiable in its $\mathbb{F}^* - \mathbb{F}^*$ and $_{gH}^{\mathrm{CF}}\mathfrak{D}_+^{\varsigma}\mathbb{S}(\mathfrak{r},\mathfrak{k}), {}_{gH}^{\mathrm{CF}}\mathfrak{D}_+^{\gamma}\mathbb{S}(\mathfrak{r},\mathfrak{k})$ are *FFCgH*-differentiable in its $\mathbb{S}^{\circledR} - \mathbb{F}^*$, then for exponential orders $\varsigma,\gamma$ where $1 < \varsigma \le 2; 0 < \gamma \le 1$, the system (1) has solution of the form

$$\mathbb{S}(\mathfrak{r},\mathfrak{k}) = (\mathbb{S}_1,\mathbb{S}_2,\mathbb{S}_3) \odot \tilde{j}(\mathfrak{r}) \odot \mathfrak{k} \odot \mathbb{E}_{\varsigma,\varsigma-\gamma,1}(0,\mathfrak{k}^{\varsigma-\gamma}) \ominus (-1)(\mathbb{S}_1,\mathbb{S}_2,\mathbb{S}_3) \odot \tilde{\ell}(\mathfrak{r}) \odot \mathbb{E}_{\varsigma,\varsigma-\gamma,1}(0,\mathfrak{k}^{\varsigma-\gamma})$$

$$\oplus(-1)(\mathbb{S}_1,\mathbb{S}_2,\mathbb{S}_3) \odot \tilde{\ell}(\mathfrak{r}) \odot \mathfrak{k}^{\varsigma-\gamma} \odot \mathbb{E}_{\varsigma,\varsigma-\gamma,\varsigma-\gamma+1}(0,\mathfrak{k}^{\varsigma-\gamma}) \oplus \left(\mathbb{I}_{\varsigma,\varsigma-\gamma,\varsigma}^{1;0,1}[\mathbb{S}^3(\mathfrak{r},\mathfrak{k}) \ominus \mathbb{S}^2(\mathfrak{r},\mathfrak{k})]\right)(\mathfrak{k}). \tag{61}$$

(c) If a FVF $\mathbb{S}(\mathfrak{r},\mathfrak{k})$ is *FFCgH*-differentiable in its $\mathbb{S}^{\circledR} - \mathbb{F}^*$ and $_{gH}^{\mathrm{CF}}\mathfrak{D}_+^{\varsigma}\mathbb{S}(\mathfrak{r},\mathfrak{k}), {}_{gH}^{\mathrm{CF}}\mathfrak{D}_+^{\gamma}\mathbb{S}(\mathfrak{r},\mathfrak{k})$ are *FFCgH*-differentiable in its $\mathbb{F}^* - \mathbb{F}^*$, then for exponential orders $\varsigma,\gamma$ where $1 < \varsigma \le 2; 0 < \gamma \le 1$, then the system (1) has solution of the form

$$\mathbb{S}(\mathfrak{r},\mathfrak{k}) = (\mathbb{S}_1,\mathbb{S}_2,\mathbb{S}_3) \odot \tilde{j}(\mathfrak{r}) \odot \mathfrak{k} \odot \mathbb{E}_{\varsigma,\varsigma-\gamma,2}(0,\mathfrak{k}^{\varsigma-\gamma}) \ominus (-1)(\mathbb{S}_1,\mathbb{S}_2,\mathbb{S}_3) \odot \tilde{\ell}(\mathfrak{r}) \odot \mathbb{E}_{\varsigma,\varsigma-\gamma,1}(0,\mathfrak{k}^{\varsigma-\gamma})$$

$$\oplus(-1)(\mathbb{S}_1,\mathbb{S}_2,\mathbb{S}_3) \odot \tilde{\ell}(\mathfrak{r}) \odot \mathfrak{k}^{\varsigma-\gamma+1} \odot \mathbb{E}_{\varsigma,\varsigma-\gamma,\varsigma-\gamma+1}(0,\mathfrak{k}^{\varsigma-\gamma}) \oplus \left(\mathbb{I}_{\varsigma,\varsigma-\gamma,\varsigma}^{1;0,1}[\mathbb{S}^3(\mathfrak{r},\mathfrak{k}) \ominus \mathbb{S}^2(\mathfrak{r},\mathfrak{k})]\right)(\mathfrak{k}). \tag{62}$$

(d) If a FVF $\mathbb{S}(\mathfrak{r},\mathfrak{k}), {}_{gH}^{\mathrm{CF}}\mathfrak{D}_+^{\gamma}\mathbb{S}(\mathfrak{r},\mathfrak{k})$ and $_{gH}^{\mathrm{CF}}\mathfrak{D}_+^{\varsigma}\mathbb{S}(\mathfrak{r},\mathfrak{k})$ are *FFCgH*-differentiable in its $\mathbb{S}^{\circledR} - \mathbb{F}^*$, then for exponential orders $\varsigma,\gamma$ where $1 < \varsigma \le 2; 0 < \gamma \le 1$, then the system (1) has solution of the form

$$\mathbb{S}(\mathfrak{r},\mathfrak{k}) = (\mathbb{S}_1,\mathbb{S}_2,\mathbb{S}_3) \odot \tilde{\ell}(\mathfrak{r}) \odot \mathbb{E}_{\varsigma,\varsigma-\gamma,1}(0,\mathfrak{k}^{\varsigma-\gamma}) \oplus (\mathbb{S}_1,\mathbb{S}_2,\mathbb{S}_3) \odot \tilde{j}(\mathfrak{r}) \odot \mathbb{E}_{\varsigma,\varsigma-\gamma,2}(0,\mathfrak{k}^{\varsigma-\gamma}) \ominus (\mathbb{S}_1,$$

$$\mathbb{S}_2,\mathbb{S}_3) \odot \tilde{\ell}(\mathfrak{r}) \odot \mathfrak{k}^{\varsigma-\gamma}\mathbb{E}_{\varsigma,\varsigma-\gamma,\varsigma-\gamma+1}(0,\mathfrak{k}^{\varsigma-\gamma}) \oplus \left(\mathbb{I}_{\varsigma,\varsigma-\gamma,\varsigma}^{1;0,1}[\mathbb{S}^3(\mathfrak{r},\mathfrak{k}) \ominus \mathbb{S}^2(\mathfrak{r},\mathfrak{k})]\right)(\mathfrak{k}). \tag{63}$$

(e) If a FVF $\mathbb{S}(\mathfrak{r},\mathfrak{k}), {}_{gH}^{\mathrm{CF}}\mathfrak{D}_+^{\gamma}\mathbb{S}(\mathfrak{r},\mathfrak{k})$ are *FFCgH*-differentiable in its $\mathbb{F}^* - \mathbb{F}^*$ and $_{gH}^{\mathrm{CF}}\mathfrak{D}_+^{\varsigma}\mathbb{S}(\mathfrak{r},\mathfrak{k})$ is *FFCgH*-differentiable in its $\mathbb{S}^{\circledR} - \mathbb{F}^*$, then for exponential orders $\varsigma,\gamma$ such that $1 < \varsigma \le 2$ and $0 < \gamma \le 1$, the system (1) has solution of the form

$$\mathbb{S}(\mathfrak{r},\mathfrak{k}) = (\mathbb{S}_1,\mathbb{S}_2,\mathbb{S}_3) \odot \tilde{j}(\mathfrak{r}) \odot \mathfrak{k}\mathbb{E}_{\varsigma,\varsigma-\gamma,2}(0,\mathfrak{k}^{\varsigma-\gamma}) \oplus (-1)(\mathbb{S}_1,\mathbb{S}_2,\mathbb{S}_3) \odot \tilde{\ell}(\mathfrak{r}) \odot \mathbb{E}_{\varsigma,\varsigma-\gamma,1}(0,\mathfrak{k}^{\varsigma-\gamma})$$

$$\ominus(\mathbb{S}_1,\mathbb{S}_2,\mathbb{S}_3) \odot \tilde{\ell}(\mathfrak{r}) \odot \mathfrak{k}^{\varsigma-\gamma}\mathbb{E}_{\varsigma,\varsigma-\gamma,\varsigma-\gamma+1}(0,\mathfrak{k}^{\varsigma-\gamma}) \ominus \left(\mathbb{I}_{\varsigma,\varsigma-\gamma,\varsigma}^{1;0,1}[\mathbb{S}^2(\mathfrak{r},\mathfrak{k}) \ominus \mathbb{S}^3(\mathfrak{r},\mathfrak{k})]\right)(\mathfrak{k}). \tag{64}$$

(f) If a FVF $\mathbb{S}(\mathfrak{r},\mathfrak{k}), {}_{gH}^{\mathrm{CF}}\mathfrak{D}_+^{\varsigma}\mathbb{S}(\mathfrak{r},\mathfrak{k})$ are *FFCgH*-differentiable in its $\mathbb{F}^* - \mathbb{F}^*$ and $_{gH}^{\mathrm{CF}}\mathfrak{D}_+^{\gamma}\mathbb{S}(\mathfrak{r},\mathfrak{k})$ is *FFCgH*-differentiable in its $\mathbb{S}^{\circledR} - \mathbb{F}^*$, then for exponential orders $\varsigma,\gamma$ such that $1 < \varsigma \le 2$ and $0 < \gamma \le 1$, the system (1) has solution of the form

$$\mathbb{S}(\mathfrak{r},\mathfrak{k}) = (\mathbb{S}_1,\mathbb{S}_2,\mathbb{S}_3) \odot \tilde{\ell}(\mathfrak{r}) \odot \mathbb{E}_{\varsigma,\varsigma-\gamma,1}(0,-\mathfrak{k}^{\varsigma-\gamma}) \oplus (\mathbb{S}_1,\mathbb{S}_2,\mathbb{S}_3) \odot \tilde{j}(\mathfrak{r}) \odot \mathfrak{k}\mathbb{E}_{\varsigma,\varsigma-\gamma,2}(0,-\mathfrak{k}^{\varsigma-\gamma})$$

$$\oplus(\mathbb{S}_1,\mathbb{S}_2,\mathbb{S}_3) \odot \tilde{\ell}(\mathfrak{r}) \odot \mathfrak{k}^{\varsigma-\gamma}\mathbb{E}_{\varsigma,\varsigma-\gamma,\varsigma-\gamma+1}(0,-\mathfrak{k}^{\varsigma-\gamma}) \ominus \left(\mathbb{I}_{\varsigma,\varsigma-\gamma,\varsigma}^{1;0,1}[\mathbb{S}^2(\mathfrak{r},\mathfrak{k}) \ominus \mathbb{S}^3(\mathfrak{r},\mathfrak{k})]\right)(\mathfrak{k}). \tag{65}$$

**Proof 22.** This theorem can be proved on the similar way as Theorem 17 with the condition that $\psi = 0$. □

**Theorem 23.** Let $\mathbb{S} : (0, \mathfrak{u}) \longrightarrow \natural_{\mathbb{R}}$ such that $\mathbb{S}(\mathfrak{r}, \mathfrak{k}) \in \mathbb{C}^{\natural_{\mathbb{R}}}(0, \mathfrak{u}) \cap L^{\natural_{\mathbb{R}}}(0, \mathfrak{u})$. Suppose that $\mathbb{S}(\mathfrak{r}, \mathfrak{k})$, $\left[{}^{\mathrm{CF}}_{gH}\mathfrak{D}^{\varsigma}_{+}\mathbb{S}\right](\mathfrak{r}, \mathfrak{k})$ and $\left[{}^{\mathrm{CF}}_{gH}\mathfrak{D}^{\gamma}_{+}\mathbb{S}\right](\mathfrak{r}, \mathfrak{k})$ follow piecewise continuity on the interval $[0, \infty)$, where $\varsigma$ and $\gamma$ are exponential orders provided that $1 < \varsigma \leq 2$ and $1 < \gamma \leq 2$, then the system (1) contains the following cases of solution:

(a) If a FVF $\mathbb{S}(\mathfrak{r}, \mathfrak{k})$, ${}^{\mathrm{CF}}_{gH}\mathfrak{D}^{\gamma}_{+}\mathbb{S}(\mathfrak{r}, \mathfrak{k})$ and ${}^{\mathrm{CF}}_{gH}\mathfrak{D}^{\varsigma}_{+}\mathbb{S}(\mathfrak{r}, \mathfrak{k})$ are *FFCgH*-differentiable in its $\mathbb{F}^{*} - \mathbb{F}^{*}$, then for exponential orders $\varsigma, \gamma$ such that $1 < \varsigma \leq 2$ and $1 < \gamma \leq 2$, the system (1) contains the solution which is given as

$$\mathbb{S}(\mathfrak{r}, \mathfrak{k}) = (\mathbb{S}_1, \mathbb{S}_2, \mathbb{S}_3) \odot \tilde{\ell}(\mathfrak{r}) \odot \mathbb{E}_{\varsigma, \varsigma - \gamma, 1}(\psi \mathfrak{k}^{\varsigma}, \mathfrak{k}^{\varsigma - \gamma}) \oplus (\mathbb{S}_1, \mathbb{S}_2, \mathbb{S}_3) \odot \tilde{j}(\mathfrak{r}) \odot \mathfrak{k} \odot \mathbb{E}_{\varsigma, \varsigma - \gamma, 2}$$

$$(\psi \mathfrak{k}^{\varsigma}, \mathfrak{k}^{\varsigma - \gamma}) \ominus (\mathbb{S}_1, \mathbb{S}_2, \mathbb{S}_3) \odot \tilde{\ell}(\mathfrak{r}) \odot \mathfrak{k}^{\varsigma - \gamma + 1} \odot \mathbb{E}_{\varsigma, \varsigma - \gamma, \varsigma - \gamma + 1}(\psi \mathfrak{k}^{\varsigma}, \mathfrak{k}^{\varsigma - \gamma}) \ominus (\mathbb{S}_1, \mathbb{S}_2, \mathbb{S}_3) \odot \tilde{j}(\mathfrak{r}) \odot$$

$$\mathfrak{k}^{\varsigma - \gamma + 1} \odot \mathbb{E}_{\varsigma, \varsigma - \gamma, \varsigma - \gamma + 2}(\psi \mathfrak{k}^{\varsigma}, \mathfrak{k}^{\varsigma - \gamma}) \ominus \left(\mathbb{I}^{1; \psi, 1}_{\varsigma, \varsigma - \gamma, \varsigma}[(1 + \psi) \odot \mathbb{S}^2(\mathfrak{r}, \mathfrak{k}) \ominus \mathbb{S}^3(\mathfrak{r}, \mathfrak{k})]\right)(\mathfrak{k}). \tag{66}$$

(b) If a FVF $\mathbb{S}(\mathfrak{r}, \mathfrak{k})$ is *FFCgH*-differentiable in its $\mathbb{F}^{*} - \mathbb{F}^{*}$ and ${}^{\mathrm{CF}}_{gH}\mathfrak{D}^{\varsigma}_{+}\mathbb{S}(\mathfrak{r}, \mathfrak{k})$, ${}^{\mathrm{CF}}_{gH}\mathfrak{D}^{\gamma}_{+}\mathbb{S}(\mathfrak{r}, \mathfrak{k})$ are *FFCgH*-differentiable in its $\mathbb{S}^{\circledast} - \mathbb{F}^{*}$, then for exponential orders $\varsigma, \gamma$ such that $1 < \varsigma \leq 2$ and $1 < \gamma \leq 2$, the system (1) contains the solution which is given as

$$\mathbb{S}(\mathfrak{r}, \mathfrak{k}) = (\mathbb{S}_1, \mathbb{S}_2, \mathbb{S}_3) \odot \tilde{j}(\mathfrak{r}) \odot \mathfrak{k} \odot \mathbb{E}_{\varsigma, \varsigma - \gamma, 2}(\psi \mathfrak{k}^{\varsigma}, \mathfrak{k}^{\varsigma - \gamma}) \ominus (-1)(\mathbb{S}_1, \mathbb{S}_2, \mathbb{S}_3) \odot \tilde{\ell}(\mathfrak{r}) \odot \mathbb{E}_{\varsigma, \varsigma - \gamma, 1}$$

$$(\psi \mathfrak{k}^{\varsigma}, \mathfrak{k}^{\varsigma - \gamma}) \oplus (-1)(\mathbb{S}_1, \mathbb{S}_2, \mathbb{S}_3) \odot \tilde{\ell}(\mathfrak{r}) \odot \mathfrak{k}^{\varsigma - \gamma} \odot \mathbb{E}_{\varsigma, \varsigma - \gamma, \varsigma - \gamma + 1}(\psi \mathfrak{k}^{\varsigma}, \mathfrak{k}^{\varsigma - \gamma}) \oplus (-1)(\mathbb{S}_1, \mathbb{S}_2, \mathbb{S}_3) \odot$$

$$\tilde{j}(\mathfrak{r}) \odot \mathfrak{k}^{\varsigma - \gamma + 1} \odot \mathbb{E}_{\varsigma, \varsigma - \gamma, \varsigma - \gamma + 2}(-\psi \mathfrak{k}^{\varsigma}, \mathfrak{k}^{\varsigma - \gamma}) \oplus (-1)\left(\mathbb{I}^{1; \psi, 1}_{\varsigma, \varsigma - \gamma, \varsigma}[(1 + \psi) \odot \mathbb{S}^2(\mathfrak{r}, \mathfrak{k}) \ominus \mathbb{S}^3(\mathfrak{r}, \mathfrak{k})]\right)(\mathfrak{k}). \tag{67}$$

(c) If a FVF $\mathbb{S}(\mathfrak{r}, \mathfrak{k})$ is *FFCgH*-differentiable in its $\mathbb{S}^{\circledast} - \mathbb{F}^{*}$ and ${}^{\mathrm{CF}}_{gH}\mathfrak{D}^{\varsigma}_{+}\mathbb{S}(\mathfrak{r}, \mathfrak{k})$, ${}^{\mathrm{CF}}_{gH}\mathfrak{D}^{\gamma}_{+}\mathbb{S}(\mathfrak{r}, \mathfrak{k})$ are *FFCgH*-differentiable in its $\mathbb{F}^{*} - \mathbb{F}^{*}$, then for exponential orders $\varsigma, \gamma$ such that $1 < \varsigma \leq 2$ and $1 < \gamma \leq 2$, the system (1) contains the solution which is given as

$$\mathbb{S}(\mathfrak{r}, \mathfrak{k}) = (\mathbb{S}_1, \mathbb{S}_2, \mathbb{S}_3) \odot \tilde{\ell}(\mathfrak{r}) \odot \mathbb{E}_{\varsigma, \varsigma - \gamma, 1}(\psi \mathfrak{k}^{\varsigma}, \mathfrak{k}^{\varsigma - \gamma}) \oplus (\mathbb{S}_1, \mathbb{S}_2, \mathbb{S}_3) \odot \tilde{j}(\mathfrak{r}) \odot \mathfrak{k} \odot \mathbb{E}_{\varsigma, \varsigma - \gamma, \varsigma - \gamma + 1}$$

$$(\psi \mathfrak{k}^{\varsigma}, \mathfrak{k}^{\varsigma - \gamma}) \oplus (-1)(\mathbb{S}_1, \mathbb{S}_2, \mathbb{S}_3) \odot \tilde{\ell}(\mathfrak{r}) \odot \mathfrak{k}^{\varsigma - \gamma} \odot \mathbb{E}_{\varsigma, \varsigma - \gamma, \varsigma - \gamma + 1}(\psi \mathfrak{k}^{\varsigma}, \mathfrak{k}^{\varsigma - \gamma}) \ominus (\mathbb{S}_1, \mathbb{S}_2, \mathbb{S}_3) \odot \tilde{j}(\mathfrak{r})$$

$$\odot \mathfrak{k}^{\varsigma - \gamma + 1} \odot \mathbb{E}_{\varsigma, \varsigma - \gamma, \varsigma - \gamma + 2}(\psi \mathfrak{k}^{\varsigma}, \mathfrak{k}^{\varsigma - \gamma}) \oplus (-1)\left(\mathbb{I}^{1; \psi, 1}_{\varsigma, \varsigma - \gamma, \varsigma}[(1 + \psi) \odot \mathbb{S}^2(\mathfrak{r}, \mathfrak{k}) \ominus \mathbb{S}^3(\mathfrak{r}, \mathfrak{k})]\right)(\mathfrak{k}). \tag{68}$$

(d) If a FVF $\mathbb{S}(\mathfrak{r}, \mathfrak{k})$, ${}^{\mathrm{CF}}_{gH}\mathfrak{D}^{\gamma}_{+}\mathbb{S}(\mathfrak{r}, \mathfrak{k})$ and ${}^{\mathrm{CF}}_{gH}\mathfrak{D}^{\varsigma}_{+}\mathbb{S}(\mathfrak{r}, \mathfrak{k})$ are *FFCgH*-differentiable in its $\mathbb{S}^{\circledast} - \mathbb{F}^{*}$, then for exponential orders $\varsigma, \gamma$ such that $1 < \varsigma \leq 2$ and $1 < \gamma \leq 2$, the system (1) contains the solution which is given as

$$\mathbb{S}(\mathfrak{r},\mathfrak{k}) = (\mathbb{S}_1,\mathbb{S}_2,\mathbb{S}_3) \odot \tilde{\ell}(\mathfrak{r}) \odot \mathbb{E}_{\varsigma,\varsigma-\gamma,1}(\psi\mathfrak{k}^\varsigma,\mathfrak{k}^{\varsigma-\gamma}) \ominus (-1)(\mathbb{S}_1,\mathbb{S}_2,\mathbb{S}_3) \odot \tilde{j}(\mathfrak{r}) \odot \mathfrak{k} \odot \mathbb{E}_{\varsigma,\varsigma-\gamma,\varsigma-\gamma+2}$$

$$(\psi\mathfrak{k}^\varsigma,\mathfrak{k}^{\varsigma-\gamma}) \oplus (-1)(\mathbb{S}_1,\mathbb{S}_2,\mathbb{S}_3) \odot \tilde{j}(\mathfrak{r}) \odot \mathfrak{k}^{\varsigma-\gamma+1} \odot \mathbb{E}_{\varsigma,\varsigma-\gamma,\varsigma-\gamma+2}(\psi\mathfrak{k}^\varsigma,\mathfrak{k}^{\varsigma-\gamma}) \ominus (\mathbb{S}_1,\mathbb{S}_2,\mathbb{S}_3) \odot \tilde{\ell}(\mathfrak{r})$$

$$\odot\mathfrak{k}^{\varsigma-\gamma} \odot \mathbb{E}_{\varsigma,\varsigma-\gamma,\varsigma-\gamma+1}(\psi\mathfrak{k}^\varsigma,\mathfrak{k}^{\varsigma-\gamma}) \ominus (\mathbb{I}^{1;\psi,1}_{\varsigma,\varsigma-\gamma,\varsigma}[(1+\psi) \odot \mathbb{S}^2(\mathfrak{r},\mathfrak{k}) \ominus \mathbb{S}^3(\mathfrak{r},\mathfrak{k})])(\mathfrak{k}). \tag{69}$$

(e) If a FVF $\mathbb{S}(\mathfrak{r},\mathfrak{k}), {}^{\mathrm{CF}}_{gH}\mathfrak{D}^\gamma_+\mathbb{S}(\mathfrak{r},\mathfrak{k})$ are *FFCgH*-differentiable in its $\mathbb{F}^* - \mathbb{F}^*$ and ${}^{\mathrm{CF}}_{gH}\mathfrak{D}^\varsigma_+\mathbb{S}(\mathfrak{r},\mathfrak{k})$ is *FFCgH*-differentiable in its $\mathbb{S}^\circledR - \mathbb{F}^*$, then for exponential orders $\varsigma,\gamma$ such that $1<\varsigma\leq 2$ and $1<\gamma\leq 2$, the system (1) has solution which is given as

$$\mathbb{S}(\mathfrak{r},\mathfrak{k}) = (\mathbb{S}_1,\mathbb{S}_2,\mathbb{S}_3) \odot \tilde{j}(\mathfrak{r}) \odot \mathfrak{k} \odot \mathbb{E}_{\varsigma,\varsigma-\gamma,2}(\psi\mathfrak{k}^\varsigma,\mathfrak{k}^{\varsigma-\gamma}) \oplus (-1)(\mathbb{S}_1,\mathbb{S}_2,\mathbb{S}_3) \odot \tilde{\ell}(\mathfrak{r}) \odot \mathbb{E}_{\varsigma,\varsigma-\gamma,1}$$

$$(\psi\mathfrak{k}^\varsigma,\mathfrak{k}^{\varsigma-\gamma}) \ominus (\mathbb{S}_1,\mathbb{S}_2,\mathbb{S}_3) \odot \tilde{\ell}(\mathfrak{r}) \odot \mathfrak{k}^{\varsigma-\gamma} \odot \mathbb{E}_{\varsigma,\varsigma-\gamma,\varsigma-\gamma+1}(\psi\mathfrak{k}^\varsigma,\mathfrak{k}^{\varsigma-\gamma}) \ominus (\mathbb{S}_1,\mathbb{S}_2,\mathbb{S}_3) \odot \tilde{j}(\mathfrak{r})\odot$$

$$\mathfrak{k}^{\varsigma-\gamma+1} \odot \mathbb{E}_{\varsigma,\varsigma-\gamma,\varsigma-\gamma+2}(\psi\mathfrak{k}^\varsigma,\mathfrak{k}^{\varsigma-\gamma}) \oplus (-1)(\mathbb{I}^{1;\psi,1}_{\varsigma,\varsigma-\gamma,\varsigma}[(1+\psi) \odot \mathbb{S}^2(\mathfrak{r},\mathfrak{k}) \ominus \mathbb{S}^3(\mathfrak{r},\mathfrak{k})])(\mathfrak{k}). \tag{70}$$

(f) If a FVF $\mathbb{S}(\mathfrak{r},\mathfrak{k}), {}^{\mathrm{CF}}_{gH}\mathfrak{D}^\varsigma_+\mathbb{S}(\mathfrak{r},\mathfrak{k})$ are *FFCgH*-differentiable in its $\mathbb{F}^* - \mathbb{F}^*$ and ${}^{\mathrm{CF}}_{gH}\mathfrak{D}^\gamma_+\mathbb{S}(\mathfrak{r},\mathfrak{k})$ is *FFCgH*-differentiable in its $\mathbb{S}^\circledR - \mathbb{F}^*$, then for exponential orders $\varsigma,\gamma$ such that $1<\varsigma\leq 2$ and $1<\gamma\leq 2$, the system (1) has solution which is given as

$$\mathbb{S}(\mathfrak{r},\mathfrak{k}) = (\mathbb{S}_1,\mathbb{S}_2,\mathbb{S}_3) \odot \tilde{\ell}(\mathfrak{r}) \odot \mathbb{E}_{\varsigma,\varsigma-\gamma,1}(\psi\mathfrak{k}^\varsigma,\mathfrak{k}^{\varsigma-\gamma}) \oplus (\mathbb{S}_1,\mathbb{S}_2,\mathbb{S}_3) \odot \tilde{j}(\mathfrak{r}) \odot \mathfrak{k} \odot \mathbb{E}_{\varsigma,\varsigma-\gamma,2}$$

$$(\psi\mathfrak{k}^\varsigma,\mathfrak{k}^{\varsigma-\gamma}) \oplus (-1)(\mathbb{S}_1,\mathbb{S}_2,\mathbb{S}_3) \odot \tilde{j}(\mathfrak{r}) \odot \mathfrak{k}^{\varsigma-\gamma+1} \odot \mathbb{E}_{\varsigma,\varsigma-\gamma,\varsigma-\gamma+2}(\psi\mathfrak{k}^\varsigma,\mathfrak{k}^{\varsigma-\gamma}) \oplus (-1)(\mathbb{S}_1,\mathbb{S}_2,\mathbb{S}_3)$$

$$\odot\tilde{\ell}(\mathfrak{r}) \odot \mathfrak{k}^{\varsigma-\gamma} \odot \mathbb{E}_{\varsigma,\varsigma-\gamma,\varsigma-\gamma+1}(\psi\mathfrak{k}^\varsigma,\mathfrak{k}^{\varsigma-\gamma}) \oplus (-1)(\mathbb{I}^{1;\psi,1}_{\varsigma,\varsigma-\gamma,\varsigma}[(1+\psi) \odot \mathbb{S}^2(\mathfrak{r},\mathfrak{k}) \ominus \mathbb{S}^3(\mathfrak{r},\mathfrak{k})])(\mathfrak{k}). \tag{71}$$

**Proof 24.** The proof is on the similar steps as mentioned in Theorem 17. □

**Theorem 25.** Let $\mathbb{S} : (0,\mathfrak{u}) \longrightarrow \natural_\mathbb{R}$ such that $\mathbb{S}(\mathfrak{r},\mathfrak{k}) \in \mathbb{C}^{\natural_\mathbb{R}}(0,\mathfrak{u}) \cap L^{\natural_\mathbb{R}}(0,\mathfrak{u})$. Suppose that $\mathbb{S}(\mathfrak{r},\mathfrak{k}), [{}^{\mathrm{CF}}_{gH}\mathfrak{D}^\varsigma_+\mathbb{S}](\mathfrak{r},\mathfrak{k})$ and $[{}^{\mathrm{CF}}_{gH}\mathfrak{D}^\gamma_+\mathbb{S}](\mathfrak{r},\mathfrak{k})$ follow piecewise continuity on the interval $[0,\infty)$, where $\varsigma$ and $\gamma$ are exponential orders provided that $1<\varsigma\leq 2$ and $1<\gamma\leq 2$ with $\psi=1$, then the system (1) contains the following cases of solution:

(a) If a FVF $\mathbb{S}(\mathfrak{r},\mathfrak{k}), {}^{\mathrm{CF}}_{gH}\mathfrak{D}^\gamma_+\mathbb{S}(\mathfrak{r},\mathfrak{k})$ and ${}^{\mathrm{CF}}_{gH}\mathfrak{D}^\varsigma_+\mathbb{S}(\mathfrak{r},\mathfrak{k})$ are *FFCgH*-differentiable in its $\mathbb{F}^* - \mathbb{F}^*$, then for exponential orders $\varsigma,\gamma$ such that $1<\varsigma\leq 2$ and $1<\gamma\leq 2$, the system (1) contains the solution which is given as

$$\mathbb{S}(\mathfrak{r},\mathfrak{k}) = (\mathbb{S}_1,\mathbb{S}_2,\mathbb{S}_3) \odot \tilde{\ell}(\mathfrak{r}) \odot \mathbb{E}_{\varsigma,\varsigma-\gamma,1}(\mathfrak{k}^\varsigma,\mathfrak{k}^{\varsigma-\gamma}) \oplus (\mathbb{S}_1,\mathbb{S}_2,\mathbb{S}_3) \odot \tilde{j}(\mathfrak{r}) \odot \mathfrak{k} \odot \mathbb{E}_{\varsigma,\varsigma-\gamma,2}$$

$$(\mathfrak{k}^\varsigma,\mathfrak{k}^{\varsigma-\gamma}) \ominus \quad (\mathbb{S}_1,\mathbb{S}_2,\mathbb{S}_3) \odot \tilde{\ell}(\mathfrak{r}) \odot \mathfrak{k}^{\varsigma-\gamma+1} \odot \mathbb{E}_{\varsigma,\varsigma-\gamma,\varsigma-\gamma+1}(\mathfrak{k}^\varsigma,\mathfrak{k}^{\varsigma-\gamma}) \ominus (\mathbb{S}_1,\mathbb{S}_2,\mathbb{S}_3) \odot \tilde{j}(\mathfrak{r})\odot$$

$$\mathfrak{k}^{\varsigma-\gamma+1} \odot \mathbb{E}_{\varsigma,\varsigma-\gamma,\varsigma-\gamma+2}(\mathfrak{k}^\varsigma,\mathfrak{k}^{\varsigma-\gamma}) \ominus (\mathbb{I}^{1;1,1}_{\varsigma,\varsigma-\gamma,\varsigma}[2 \odot \mathbb{S}^2(\mathfrak{r},\mathfrak{k}) \ominus \mathbb{S}^3(\mathfrak{r},\mathfrak{k})])(\mathfrak{k}). \tag{72}$$

(b) If a FVF $\mathbb{S}(\mathfrak{r}, \mathfrak{k})$ is *FFCgH*-differentiable in its $\mathbb{F}^* - \mathbb{F}^*$ and ${}_{gH}^{\mathrm{CF}}\mathfrak{D}_+^\varsigma \mathbb{S}(\mathfrak{r}, \mathfrak{k}), {}_{gH}^{\mathrm{CF}}\mathfrak{D}_+^\gamma \mathbb{S}(\mathfrak{r}, \mathfrak{k})$ are *FFCgH*-differentiable in its $\mathbb{S}^\circledast - \mathbb{F}^*$, then for exponential orders $\varsigma, \gamma$ such that $1 < \varsigma \le 2$ and $1 < \gamma \le 2$, the system (1) contains the solution which is given as

$$\mathbb{S}(\mathfrak{r}, \mathfrak{k}) = (\mathbb{S}_1, \mathbb{S}_2, \mathbb{S}_3) \odot \tilde{j}(\mathfrak{r}) \odot \mathfrak{k} \odot \mathbb{E}_{\varsigma, \varsigma - \gamma, 2}(\mathfrak{k}^\varsigma, \mathfrak{k}^{\varsigma - \gamma}) \ominus (-1)(\mathbb{S}_1, \mathbb{S}_2, \mathbb{S}_3) \odot \tilde{\ell}(\mathfrak{r}) \odot \mathbb{E}_{\varsigma, \varsigma - \gamma, 1}$$

$$(\mathfrak{k}^\varsigma, \mathfrak{k}^{\varsigma - \gamma}) \oplus (-1)(\mathbb{S}_1, \mathbb{S}_2, \mathbb{S}_3) \odot \tilde{\ell}(\mathfrak{r}) \odot \mathfrak{k}^{\varsigma - \gamma} \odot \mathbb{E}_{\varsigma, \varsigma - \gamma, \varsigma - \gamma + 1}(\mathfrak{k}^\varsigma, \mathfrak{k}^{\varsigma - \gamma}) \oplus (-1)(\mathbb{S}_1, \mathbb{S}_2, \mathbb{S}_3) \odot \tilde{j}(\mathfrak{r})$$

$$\odot \mathfrak{k}^{\varsigma - \gamma + 1} \odot \mathbb{E}_{\varsigma, \varsigma - \gamma, \varsigma - \gamma + 2}(\mathfrak{k}^\varsigma, \mathfrak{k}^{\varsigma - \gamma}) \oplus (-1)(\mathbb{I}_{\varsigma, \varsigma - \gamma, \varsigma}^{1;1,1}[2 \odot \mathbb{S}^2(\mathfrak{r}, \mathfrak{k}) \ominus \mathbb{S}^3(\mathfrak{r}, \mathfrak{k})])(\mathfrak{k}). \tag{73}$$

(c) If a FVF $\mathbb{S}(\mathfrak{r}, \mathfrak{k})$ is *FFCgH*-differentiable in its $\mathbb{S}^\circledast - \mathbb{F}^*$ and ${}_{gH}^{\mathrm{CF}}\mathfrak{D}_+^\varsigma \mathbb{S}(\mathfrak{r}, \mathfrak{k}), {}_{gH}^{\mathrm{CF}}\mathfrak{D}_+^\gamma \mathbb{S}(\mathfrak{r}, \mathfrak{k})$ are *FFCgH*-differentiable in its $\mathbb{F}^* - \mathbb{F}^*$, then for exponential orders $\varsigma, \gamma$ such that $1 < \varsigma \le 2$ and $1 < \gamma \le 2$, the system (1) contains the solution which is given as

$$\mathbb{S}(\mathfrak{r}, \mathfrak{k}) = (\mathbb{S}_1, \mathbb{S}_2, \mathbb{S}_3) \odot \tilde{\ell}(\mathfrak{r}) \odot \mathbb{E}_{\varsigma, \varsigma - \gamma, 1}(\mathfrak{k}^\varsigma, \mathfrak{k}^{\varsigma - \gamma}) \oplus (\mathbb{S}_1, \mathbb{S}_2, \mathbb{S}_3) \odot \tilde{j}(\mathfrak{r}) \odot \mathfrak{k} \odot \mathbb{E}_{\varsigma, \varsigma - \gamma, \varsigma - \gamma + 1}$$

$$(\mathfrak{k}^\varsigma, \mathfrak{k}^{\varsigma - \gamma}) \oplus (-1)(\mathbb{S}_1, \mathbb{S}_2, \mathbb{S}_3) \odot \tilde{\ell}(\mathfrak{r}) \odot \mathfrak{k}^{\varsigma - \gamma} \odot \mathbb{E}_{\varsigma, \varsigma - \gamma, \varsigma - \gamma + 1}(\mathfrak{k}^\varsigma, \mathfrak{k}^{\varsigma - \gamma}) \ominus (\mathbb{S}_1, \mathbb{S}_2, \mathbb{S}_3) \odot$$

$$\tilde{j}(\mathfrak{r}) \odot \mathfrak{k}^{\varsigma - \gamma + 1} \odot \mathbb{E}_{\varsigma, \varsigma - \gamma, \varsigma - \gamma + 2}(\mathfrak{k}^\varsigma, \mathfrak{k}^{\varsigma - \gamma}) \oplus (-1)(\mathbb{I}_{\varsigma, \varsigma - \gamma, \varsigma}^{1;1,1}[2 \odot \mathbb{S}^2(\mathfrak{r}, \mathfrak{k}) \ominus \mathbb{S}^3(\mathfrak{r}, \mathfrak{k})])(\mathfrak{k}). \tag{74}$$

(d) If a FVF $\mathbb{S}(\mathfrak{r}, \mathfrak{k}), {}_{gH}^{\mathrm{CF}}\mathfrak{D}_+^\gamma \mathbb{S}(\mathfrak{r}, \mathfrak{k})$ and ${}_{gH}^{\mathrm{CF}}\mathfrak{D}_+^\varsigma \mathbb{S}(\mathfrak{r}, \mathfrak{k})$ are *FFCgH*-differentiable in its $\mathbb{S}^\circledast - \mathbb{F}^*$, then for exponential orders $\varsigma, \gamma$ such that $1 < \varsigma \le 2$ and $1 < \gamma \le 2$, the system (1) contains the solution which is given as

$$\mathbb{S}(\mathfrak{r}, \mathfrak{k}) = (\mathbb{S}_1, \mathbb{S}_2, \mathbb{S}_3) \odot \tilde{\ell}(\mathfrak{r}) \odot \mathbb{E}_{\varsigma, \varsigma - \gamma, 1}(\mathfrak{k}^\varsigma, \mathfrak{k}^{\varsigma - \gamma}) \ominus (-1)(\mathbb{S}_1, \mathbb{S}_2, \mathbb{S}_3) \odot \tilde{j}(\mathfrak{r}) \odot \mathfrak{k} \odot \mathbb{E}_{\varsigma, \varsigma - \gamma, \varsigma - \gamma + 2}$$

$$(\mathfrak{k}^\varsigma, \mathfrak{k}^{\varsigma - \gamma}) \oplus (-1)(\mathbb{S}_1, \mathbb{S}_2, \mathbb{S}_3) \odot \tilde{j}(\mathfrak{r}) \odot \mathfrak{k}^{\varsigma - \gamma + 1} \odot \mathbb{E}_{\varsigma, \varsigma - \gamma, \varsigma - \gamma + 2}(\mathfrak{k}^\varsigma, \mathfrak{k}^{\varsigma - \gamma}) \ominus (\mathbb{S}_1, \mathbb{S}_2, \mathbb{S}_3) \odot$$

$$\tilde{\ell}(\mathfrak{r}) \odot \mathfrak{k}^{\varsigma - \gamma} \odot \mathbb{E}_{\varsigma, \varsigma - \gamma, \varsigma - \gamma + 1}(\mathfrak{k}^\varsigma, \mathfrak{k}^{\varsigma - \gamma}) \ominus (\mathbb{I}_{\varsigma, \varsigma - \gamma, \varsigma}^{1;1,1}[2 \odot \mathbb{S}^2(\mathfrak{r}, \mathfrak{k}) \ominus \mathbb{S}^3(\mathfrak{r}, \mathfrak{k})])(\mathfrak{k}). \tag{75}$$

(e) If a FVF $\mathbb{S}(\mathfrak{r}, \mathfrak{k}), {}_{gH}^{\mathrm{CF}}\mathfrak{D}_+^\gamma \mathbb{S}(\mathfrak{r}, \mathfrak{k})$ are *FFCgH*-differentiable in its $\mathbb{F}^* - \mathbb{F}^*$ and ${}_{gH}^{\mathrm{CF}}\mathfrak{D}_+^\varsigma \mathbb{S}(\mathfrak{r}, \mathfrak{k})$ is *FFCgH*-differentiable in its $\mathbb{S}^\circledast - \mathbb{F}^*$, then for exponential orders $\varsigma, \gamma$ such that $1 < \varsigma \le 2$ and $1 < \gamma \le 2$, the system (1) has solution which is given as

$$\mathbb{S}(\mathfrak{r}, \mathfrak{k}) = (\mathbb{S}_1, \mathbb{S}_2, \mathbb{S}_3) \odot \tilde{j}(\mathfrak{r}) \odot \mathfrak{k} \odot \mathbb{E}_{\varsigma, \varsigma - \gamma, 2}(\mathfrak{k}^\varsigma, \mathfrak{k}^{\varsigma - \gamma}) \oplus (-1)(\mathbb{S}_1, \mathbb{S}_2, \mathbb{S}_3) \odot \tilde{\ell}(\mathfrak{r}) \odot \mathbb{E}_{\varsigma, \varsigma - \gamma, 1}$$

$$(\mathfrak{k}^\varsigma, \mathfrak{k}^{\varsigma - \gamma}) \ominus (\mathbb{S}_1, \mathbb{S}_2, \mathbb{S}_3) \odot \tilde{\ell}(\mathfrak{r}) \odot \mathfrak{k}^{\varsigma - \gamma} \odot \mathbb{E}_{\varsigma, \varsigma - \gamma, \varsigma - \gamma + 1}(\mathfrak{k}^\varsigma, \mathfrak{k}^{\varsigma - \gamma}) \ominus (\mathbb{S}_1, \mathbb{S}_2, \mathbb{S}_3) \odot \tilde{j}(\mathfrak{r}) \odot \mathfrak{k}^{\varsigma - \gamma + 1}$$

$$\odot \mathbb{E}_{\varsigma, \varsigma - \gamma, \varsigma - \gamma + 2}(\mathfrak{k}^\varsigma, \mathfrak{k}^{\varsigma - \gamma}) \oplus (-1)(\mathbb{I}_{\varsigma, \varsigma - \gamma, \varsigma}^{1;1,1}[2 \odot \mathbb{S}^2(\mathfrak{r}, \mathfrak{k}) \ominus \mathbb{S}^3(\mathfrak{r}, \mathfrak{k})])(\mathfrak{k}). \tag{76}$$

(f) If a FVF $\mathbb{S}(\mathfrak{r},\mathfrak{k}), {}^{\mathrm{CF}}_{gH}\mathfrak{D}^{\varsigma}_+\mathbb{S}(\mathfrak{r},\mathfrak{k})$ are *FFCgH-differentiable* in its $\mathbb{F}^* - \mathbb{F}^*$ and ${}^{\mathrm{CF}}_{gH}\mathfrak{D}^{\gamma}_+\mathbb{S}(\mathfrak{r},\mathfrak{k})$ is *FFCgH-differentiable* in its $\mathbb{S}^{\circledast} - \mathbb{F}^*$, then for exponential orders $\varsigma, \gamma$ such that $1 < \varsigma \le 2$ and $1 < \gamma \le 2$, the system (1) has solution which is given as

$$\mathbb{S}(\mathfrak{r},\mathfrak{k}) = (\mathbb{S}_1, \mathbb{S}_2, \mathbb{S}_3) \odot \tilde{\ell}(\mathfrak{r}) \odot \mathbb{E}_{\varsigma,\varsigma-\gamma,1}(\mathfrak{k}^{\varsigma}, \mathfrak{k}^{\varsigma-\gamma}) \oplus (\mathbb{S}_1, \mathbb{S}_2, \mathbb{S}_3) \odot \tilde{j}(\mathfrak{r}) \odot \mathfrak{k} \odot \mathbb{E}_{\varsigma,\varsigma-\gamma,2}$$

$$(\mathfrak{k}^{\varsigma}, \mathfrak{k}^{\varsigma-\gamma}) \oplus (-1)(\mathbb{S}_1, \mathbb{S}_2, \mathbb{S}_3) \odot \tilde{j}(\mathfrak{r}) \odot \mathfrak{k}^{\varsigma-\gamma+1} \odot \mathbb{E}_{\varsigma,\varsigma-\gamma,\varsigma-\gamma+2}(\mathfrak{k}^{\varsigma}, \mathfrak{k}^{\varsigma-\gamma}) \oplus (-1)(\mathbb{S}_1, \mathbb{S}_2, \mathbb{S}_3) \odot$$

$$\tilde{\ell}(\mathfrak{r}) \odot \mathfrak{k}^{\varsigma-\gamma} \odot \mathbb{E}_{\varsigma,\varsigma-\gamma,\varsigma-\gamma+1}(\mathfrak{k}^{\varsigma}, \mathfrak{k}^{\varsigma-\gamma}) \oplus (-1)(\mathbb{I}^{1;1,1}_{\varsigma,\varsigma-\gamma,\varsigma}[2 \odot \mathbb{S}^2(\mathfrak{r},\mathfrak{k}) \ominus \mathbb{S}^3(\mathfrak{r},\mathfrak{k})])(\mathfrak{k}). \tag{77}$$

**Proof 26.** The proof is on the similar steps as mentioned in Theorem 19. □

**Theorem 27.** Let $\mathbb{S} : (0, \mathfrak{u}) \longrightarrow \natural_{\mathbb{R}}$ such that $\mathbb{S}(\mathfrak{r},\mathfrak{k}) \in \mathbb{C}^{\natural_{\mathbb{R}}}(0, \mathfrak{u}) \cap L^{\natural_{\mathbb{R}}}(0, \mathfrak{u})$. Suppose that $\mathbb{S}(\mathfrak{r},\mathfrak{k}), [{}^{\mathrm{CF}}_{gH}\mathfrak{D}^{\varsigma}_+\mathbb{S}](\mathfrak{r},\mathfrak{k})$ and $[{}^{\mathrm{CF}}_{gH}\mathfrak{D}^{\gamma}_+\mathbb{S}](\mathfrak{r},\mathfrak{k})$ follow piecewise continuity on the interval $[0, \infty)$, where $\varsigma$ and $\gamma$ are exponential orders provided that $1 < \varsigma \le 2$ and $1 < \gamma \le 2$ with $\psi = 0$, then the system (1) contains the following cases of solution:

(a) If a FVF $\mathbb{S}(\mathfrak{r},\mathfrak{k}), {}^{\mathrm{CF}}_{gH}\mathfrak{D}^{\gamma}_+\mathbb{S}(\mathfrak{r},\mathfrak{k})$ and ${}^{\mathrm{CF}}_{gH}\mathfrak{D}^{\varsigma}_+\mathbb{S}(\mathfrak{r},\mathfrak{k})$ are *FFCgH-differentiable* in its $\mathbb{F}^* - \mathbb{F}^*$, then for exponential orders $\varsigma, \gamma$ such that $1 < \varsigma \le 2$ and $1 < \gamma \le 2$, the system (1) contains the solution which is given as

$$\mathbb{S}(\mathfrak{r},\mathfrak{k}) = (\mathbb{S}_1, \mathbb{S}_2, \mathbb{S}_3) \odot \tilde{\ell}(\mathfrak{r}) \odot \mathbb{E}_{\varsigma,\varsigma-\gamma,1}(0, \mathfrak{k}^{\varsigma-\gamma}) \oplus (\mathbb{S}_1, \mathbb{S}_2, \mathbb{S}_3) \odot \tilde{j}(\mathfrak{r}) \odot \mathfrak{k} \odot \mathbb{E}_{\varsigma,\varsigma-\gamma,2}$$

$$(0, \mathfrak{k}^{\varsigma-\gamma}) \ominus (\mathbb{S}_1, \mathbb{S}_2, \mathbb{S}_3) \odot \tilde{\ell}(\mathfrak{r}) \odot \mathfrak{k}^{\varsigma-\gamma+1} \odot \mathbb{E}_{\varsigma,\varsigma-\gamma,\varsigma-\gamma+1}(0, \mathfrak{k}^{\varsigma-\gamma}) \ominus (\mathbb{S}_1, \mathbb{S}_2, \mathbb{S}_3) \odot \tilde{j}(\mathfrak{r}) \odot \mathfrak{k}^{\varsigma-\gamma+1}$$

$$\odot \mathbb{E}_{\varsigma,\varsigma-\gamma,\varsigma-\gamma+2}(0, \mathfrak{k}^{\varsigma-\gamma}) \ominus (\mathbb{I}^{1;0,1}_{\varsigma,\varsigma-\gamma,\varsigma}[\mathbb{S}^2(\mathfrak{r},\mathfrak{k}) \ominus \mathbb{S}^3(\mathfrak{r},\mathfrak{k})])(\mathfrak{k}). \tag{78}$$

(b) If a FVF $\mathbb{S}(\mathfrak{r},\mathfrak{k})$ is *FFCgH-differentiable* in its $\mathbb{F}^* - \mathbb{F}^*$ and ${}^{\mathrm{CF}}_{gH}\mathfrak{D}^{\varsigma}_+\mathbb{S}(\mathfrak{r},\mathfrak{k}), {}^{\mathrm{CF}}_{gH}\mathfrak{D}^{\gamma}_+\mathbb{S}(\mathfrak{r},\mathfrak{k})$ are *FFCgH-differentiable* in its $\mathbb{S}^{\circledast} - \mathbb{F}^*$, then for exponential orders $\varsigma, \gamma$ such that $1 < \varsigma \le 2$ and $1 < \gamma \le 2$, the system (1) contains the solution which is given as

$$\mathbb{S}(\mathfrak{r},\mathfrak{k}) = (\mathbb{S}_1, \mathbb{S}_2, \mathbb{S}_3) \odot \tilde{j}(\mathfrak{r}) \odot \mathfrak{k} \odot \mathbb{E}_{\varsigma,\varsigma-\gamma,2}(0, \mathfrak{k}^{\varsigma-\gamma}) \ominus (-1)(\mathbb{S}_1, \mathbb{S}_2, \mathbb{S}_3) \odot \tilde{\ell}(\mathfrak{r}) \odot \mathbb{E}_{\varsigma,\varsigma-\gamma,1}$$

$$(0, \mathfrak{k}^{\varsigma-\gamma}) \oplus (-1)(\mathbb{S}_1, \mathbb{S}_2, \mathbb{S}_3) \odot \tilde{\ell}(\mathfrak{r}) \odot \mathfrak{k}^{\varsigma-\gamma} \odot \mathbb{E}_{\varsigma,\varsigma-\gamma,\varsigma-\gamma+1}(0, \mathfrak{k}^{\varsigma-\gamma}) \oplus (-1)(\mathbb{S}_1, \mathbb{S}_2, \mathbb{S}_3) \odot \tilde{j}(\mathfrak{r})$$

$$\odot \mathfrak{k}^{\varsigma-\gamma+1} \odot \mathbb{E}_{\varsigma,\varsigma-\gamma,\varsigma-\gamma+2}(0, \mathfrak{k}^{\varsigma-\gamma}) \oplus (-1)(\mathbb{I}^{1;0,1}_{\varsigma,\varsigma-\gamma,\varsigma}[\mathbb{S}^2(\mathfrak{r},\mathfrak{k}) \ominus \mathbb{S}^3(\mathfrak{r},\mathfrak{k})])(\mathfrak{k}). \tag{79}$$

(c) If a FVF $\mathbb{S}(\mathfrak{r},\mathfrak{k})$ is *FFCgH-differentiable* in its $\mathbb{S}^{\circledast} - \mathbb{F}^*$ and ${}^{\mathrm{CF}}_{gH}\mathfrak{D}^{\varsigma}_+\mathbb{S}(\mathfrak{r},\mathfrak{k}), {}^{\mathrm{CF}}_{gH}\mathfrak{D}^{\gamma}_+\mathbb{S}(\mathfrak{r},\mathfrak{k})$ are *FFCgH-differentiable* in its $\mathbb{F}^* - \mathbb{F}^*$, then for exponential orders $\varsigma, \gamma$ such that $1 < \varsigma \le 2$ and $1 < \gamma \le 2$, the system (1) contains the solution which is given as

$$\mathbb{S}(\mathfrak{r},\mathfrak{k}) = (\mathbb{S}_1,\mathbb{S}_2,\mathbb{S}_3) \odot \tilde{\ell}(\mathfrak{r}) \odot \mathbb{E}_{\varsigma,\varsigma-\gamma,1}(0,\mathfrak{k}^{\varsigma-\gamma}) \oplus (\mathbb{S}_1,\mathbb{S}_2,\mathbb{S}_3) \odot \tilde{j}(\mathfrak{r}) \odot \mathfrak{k} \odot \mathbb{E}_{\varsigma,\varsigma-\gamma,\varsigma-\gamma+1}$$

$$(0,\mathfrak{k}^{\varsigma-\gamma}) \oplus (-1)(\mathbb{S}_1,\mathbb{S}_2,\mathbb{S}_3) \odot \tilde{\ell}(\mathfrak{r}) \odot \mathfrak{k}^{\varsigma-\gamma} \odot \mathbb{E}_{\varsigma,\varsigma-\gamma,\varsigma-\gamma+1}(0,\mathfrak{k}^{\varsigma-\gamma}) \ominus (\mathbb{S}_1,\mathbb{S}_2,\mathbb{S}_3) \odot \tilde{j}(\mathfrak{r}) \odot$$

$$\mathfrak{k}^{\varsigma-\gamma+1} \odot \mathbb{E}_{\varsigma,\varsigma-\gamma,\varsigma-\gamma+2}(0,\mathfrak{k}^{\varsigma-\gamma}) \oplus (-1)(\mathbb{I}^{1;0,1}_{\varsigma,\varsigma-\gamma,\varsigma}[\mathbb{S}^2(\mathfrak{r},\mathfrak{k}) \ominus \mathbb{S}^3(\mathfrak{r},\mathfrak{k})])(\mathfrak{k}). \tag{80}$$

(d) If a FVF $\mathbb{S}(\mathfrak{r},\mathfrak{k})$, $^{\mathrm{CF}}_{gH}\mathfrak{D}^{\gamma}_{+}\mathbb{S}(\mathfrak{r},\mathfrak{k})$ and $^{\mathrm{CF}}_{gH}\mathfrak{D}^{\varsigma}_{+}\mathbb{S}(\mathfrak{r},\mathfrak{k})$ are *FFCgH*-differentiable in its $\mathbb{S}^{\circledR} - \mathbb{F}^{*}$, then for exponential orders $\varsigma,\gamma$ such that $1 < \varsigma \leq 2$ and $1 < \gamma \leq 2$, the system (1) contains the solution which is given as

$$\mathbb{S}(\mathfrak{r},\mathfrak{k}) = (\mathbb{S}_1,\mathbb{S}_2,\mathbb{S}_3) \odot \tilde{\ell}(\mathfrak{r}) \odot \mathbb{E}_{\varsigma,\varsigma-\gamma,1}(0,\mathfrak{k}^{\varsigma-\gamma}) \ominus (-1)(\mathbb{S}_1,\mathbb{S}_2,\mathbb{S}_3) \odot \tilde{j}(\mathfrak{r}) \odot \mathfrak{k} \odot \mathbb{E}_{\varsigma,\varsigma-\gamma,\varsigma-\gamma+2}$$

$$(0,\mathfrak{k}^{\varsigma-\gamma}) \oplus (-1)(\mathbb{S}_1,\mathbb{S}_2,\mathbb{S}_3) \odot \tilde{j}(\mathfrak{r}) \odot \mathfrak{k}^{\varsigma-\gamma+1} \odot \mathbb{E}_{\varsigma,\varsigma-\gamma,\varsigma-\gamma+2}(0,\mathfrak{k}^{\varsigma-\gamma}) \ominus (\mathbb{S}_1,\mathbb{S}_2,\mathbb{S}_3) \odot \tilde{\ell}(\mathfrak{r}) \odot$$

$$\mathfrak{k}^{\varsigma-\gamma} \odot \mathbb{E}_{\varsigma,\varsigma-\gamma,\varsigma-\gamma+1}(0,\mathfrak{k}^{\varsigma-\gamma}) \ominus (\mathbb{I}^{1;0,1}_{\varsigma,\varsigma-\gamma,\varsigma}[\mathbb{S}^2(\mathfrak{r},\mathfrak{k}) \ominus \mathbb{S}^3(\mathfrak{r},\mathfrak{k})])(\mathfrak{k}). \tag{81}$$

(e) If a FVF $\mathbb{S}(\mathfrak{r},\mathfrak{k})$, $^{\mathrm{CF}}_{gH}\mathfrak{D}^{\gamma}_{+}\mathbb{S}(\mathfrak{r},\mathfrak{k})$ are *FFCgH*-differentiable in its $\mathbb{F}^{*} - \mathbb{F}^{*}$ and $^{\mathrm{CF}}_{gH}\mathfrak{D}^{\varsigma}_{+}\mathbb{S}(\mathfrak{r},\mathfrak{k})$ is *FFCgH*-differentiable in its $\mathbb{S}^{\circledR} - \mathbb{F}^{*}$, then for exponential orders $\varsigma,\gamma$ such that $1 < \varsigma \leq 2$ and $1 < \gamma \leq 2$, the system (1) has solution which is given as

$$\mathbb{S}(\mathfrak{r},\mathfrak{k}) = (\mathbb{S}_1,\mathbb{S}_2,\mathbb{S}_3) \odot \tilde{j}(\mathfrak{r}) \odot \mathfrak{k} \odot \mathbb{E}_{\varsigma,\varsigma-\gamma,2}(0,\mathfrak{k}^{\varsigma-\gamma}) \oplus (-1)(\mathbb{S}_1,\mathbb{S}_2,\mathbb{S}_3) \odot \tilde{\ell}(\mathfrak{r}) \odot \mathbb{E}_{\varsigma,\varsigma-\gamma,1}$$

$$(0,\mathfrak{k}^{\varsigma-\gamma}) \ominus (\mathbb{S}_1,\mathbb{S}_2,\mathbb{S}_3) \odot \tilde{\ell}(\mathfrak{r}) \odot \mathfrak{k}^{\varsigma-\gamma+1} \odot \mathbb{E}_{\varsigma,\varsigma-\gamma,\varsigma-\gamma+1}(0,\mathfrak{k}^{\varsigma-\gamma}) \ominus (\mathbb{S}_1,\mathbb{S}_2,\mathbb{S}_3) \odot \tilde{j}(\mathfrak{r}) \odot$$

$$\mathfrak{k}^{\varsigma-\gamma+1} \odot \mathbb{E}_{\varsigma,\varsigma-\gamma,\varsigma-\gamma+2}(0,\mathfrak{k}^{\varsigma-\gamma}) \oplus (-1)(\mathbb{I}^{1;0,1}_{\varsigma,\varsigma-\gamma,\varsigma}[\mathbb{S}^2(\mathfrak{r},\mathfrak{k}) \ominus \mathbb{S}^3(\mathfrak{r},\mathfrak{k})])(\mathfrak{k}). \tag{82}$$

(f) If a FVF $\mathbb{S}(\mathfrak{r},\mathfrak{k})$, $^{\mathrm{CF}}_{gH}\mathfrak{D}^{\varsigma}_{+}\mathbb{S}(\mathfrak{r},\mathfrak{k})$ are *FFCgH*-differentiable in its $\mathbb{F}^{*} - \mathbb{F}^{*}$ and $^{\mathrm{CF}}_{gH}\mathfrak{D}^{\gamma}_{+}\mathbb{S}(\mathfrak{r},\mathfrak{k})$ is *FFCgH*-differentiable in its $\mathbb{S}^{\circledR} - \mathbb{F}^{*}$, then for exponential orders $\varsigma,\gamma$ such that $1 < \varsigma \leq 2$ and $1 < \gamma \leq 2$, the system (1) has solution which is given as

$$\mathbb{S}(\mathfrak{r},\mathfrak{k}) = (\mathbb{S}_1,\mathbb{S}_2,\mathbb{S}_3) \odot \tilde{\ell}(\mathfrak{r}) \odot \mathbb{E}_{\varsigma,\varsigma-\gamma,1}(0,\mathfrak{k}^{\varsigma-\gamma}) \oplus (\mathbb{S}_1,\mathbb{S}_2,\mathbb{S}_3) \odot \tilde{j}(\mathfrak{r}) \odot \mathfrak{k} \odot \mathbb{E}_{\varsigma,\varsigma-\gamma,2}$$

$$(0,\mathfrak{k}^{\varsigma-\gamma}) \oplus (-1)(\mathbb{S}_1,\mathbb{S}_2,\mathbb{S}_3) \odot \tilde{j}(\mathfrak{r}) \odot \mathfrak{k}^{\varsigma-\gamma+1} \odot \mathbb{E}_{\varsigma,\varsigma-\gamma,\varsigma-\gamma+2}(0,\mathfrak{k}^{\varsigma-\gamma}) \oplus (-1)(\mathbb{S}_1,\mathbb{S}_2,\mathbb{S}_3) \odot$$

$$\tilde{\ell}(\mathfrak{r}) \odot \mathfrak{k}^{\varsigma-\gamma} \odot \mathbb{E}_{\varsigma,\varsigma-\gamma,\varsigma-\gamma+1}(0,\mathfrak{k}^{\varsigma-\gamma}) \oplus (-1)(\mathbb{I}^{1;0,1}_{\varsigma,\varsigma-\gamma,\varsigma}[\mathbb{S}^2(\mathfrak{r},\mathfrak{k}) \ominus \mathbb{S}^3(\mathfrak{r},\mathfrak{k})])(\mathfrak{k}). \tag{83}$$

**Proof 28.** The proof is on the similar steps as mentioned in Theorem 21. $\qquad\square$

**Structure of Solution**

The steps of finding the analytical solution of FFGFH-NDEs model with the given initial conditions are discussed as:

1. Consider the problem of FFGFH-NDEs (1) with the specified initial conditions in fuzzy environment.
2. Applying the $\mathbb{FFLT}$ to problem (1) and Lemma 1 in order to obtain the separated form of the aforementioned problem.
3. Apply Theorems 15, 9 and Theorem 8 to evaluate the $\mathbb{FFLT}$ of fractional order *FFCgH*-differentiability of FVF $\mathbb{S}(\mathfrak{r}, \mathfrak{k})$ with $0 < \gamma \leq 1$ and $0 < \varsigma \leq 2$.
4. Apply the Theorem 9 to transform the complicated results into UVMLF form so that the given system can be solved more effectively.
5. Apply the Theorem 8 to determined the combined form of the UVMLF and integral operator.
6. Apply $\mathbb{IFFLT}$ along with Theorem 10 in order to determine the analytical fuzzy solutions for specified type of *FFCgH*-differentiability.

## 4 Examples

In this section, we will present some examples to explain more specific general results. These examples show how practical those results actually are. First, we shall consider the example which concerns the analytical solution of FFGFH-NDEs having fractional orders $\gamma$ and $\varsigma$ such that $1 < \varsigma \leq 2$, $0 < \gamma \leq 1$. Secondly, we will present the solutions of aforementioned problem for $\psi = 1$ and $\psi = 0$ for $\gamma$ and $\varsigma$ such that $1 < \varsigma \leq 2$, $0 < \gamma \leq 1$. Furthermore, we will discuss an other example with fractional orders $\gamma, \varsigma$ such that $1 < \varsigma \leq 2$, $1 < \gamma \leq 2$. Finally, we will deduce the results for $\psi = 1$ and $\psi = 0$ with fractional orders $\gamma, \varsigma$ such that $1 < \varsigma \leq 2$, $1 < \gamma \leq 2$.

**Example 29.** Consider the FFGFH-NDE (1) for $\gamma = \dfrac{8}{9}; \varsigma = \dfrac{13}{9}$ along with the triangular fuzzy initial conditions $(\mathbb{S}_1, \mathbb{S}_2, \mathbb{S}_3) \odot \tilde{\ell}(\mathfrak{r}) = (4, 5, 8) \odot (\mathfrak{r}^2 + 1)$ and $(\mathbb{S}_1, \mathbb{S}_2, \mathbb{S}_3) \odot \tilde{j}(\mathfrak{r}) = (-3, -1, 5) \odot (\mathfrak{r} - 1)$. Then using Theorem 17, the FFGFH-NDE (1) contains the following forms of solutions:

(a) If a FVF $\mathbb{S}(\mathfrak{r}, \mathfrak{k}), {}^{\mathbb{CF}}_{gH}\mathfrak{D}^{\gamma}_{+}\mathbb{S}(\mathfrak{r}, \mathfrak{k})$ and ${}^{\mathbb{CF}}_{gH}\mathfrak{D}^{\varsigma}_{+}\mathbb{S}(\mathfrak{r}, \mathfrak{k})$ are *FFCgH*-differentiable in its $\mathbb{F}^{\circledast} - \mathbb{F}^{\circledast}$, then the problem (1) has solution of the form

$$\mathbb{S}(\mathfrak{r}, \mathfrak{k}) = (4, 5, 8) \odot (\mathfrak{r}^2 + 1) \odot \mathbb{E}_{\frac{13}{9}, \frac{5}{9}, 1}(\psi \mathfrak{k}^{\frac{13}{9}}, \mathfrak{k}^{\frac{5}{9}}) \oplus (-3, -1, 5) \odot (\mathfrak{r} - 1) \odot \mathfrak{k} \mathbb{E}_{\frac{13}{9}, \frac{5}{9}, 2}(\psi \mathfrak{k}^{\frac{13}{9}},$$

$$\mathfrak{k}^{\frac{5}{9}}) \ominus \mathbb{S} \odot \mathfrak{k}^{\frac{14}{9}} \mathbb{E}_{\frac{13}{9}, \frac{5}{9}, \frac{14}{9}}(\psi \mathfrak{k}^{\frac{13}{9}}, \mathfrak{k}^{\frac{5}{9}}) \ominus (\mathbb{I}^{1;\psi,1}_{\frac{13}{9}, \frac{5}{9}, \frac{13}{9}}[(1 + \psi) \odot \mathbb{S}^2(\mathfrak{r}, \mathfrak{k}) \ominus \mathbb{S}^3(\mathfrak{r}, \mathfrak{k})])(\mathfrak{k}). \tag{84}$$

(b) If a FVF $\mathbb{S}(\mathfrak{r}, \mathfrak{k})$ is *FFCgH*-differentiable in its $\mathbb{F}^{\circledast} - \mathbb{F}^{\circledast}$ and ${}^{\mathbb{CF}}_{gH}\mathfrak{D}^{\varsigma}_{+}\mathbb{S}(\mathfrak{r}, \mathfrak{k}), {}^{\mathbb{CF}}_{gH}\mathfrak{D}^{\gamma}_{+}\mathbb{S}(\mathfrak{r}, \mathfrak{k})$ are *FFCgH*-differentiable in its $\mathbb{S}^{\circledast} - \mathbb{F}^{\circledast}$, then the problem (1) has solution of the form

$$\mathbb{S}(\mathfrak{r},\mathfrak{f}) = (-3,-1,5) \odot (\mathfrak{r}-1) \odot \mathfrak{f}\mathbb{E}_{\frac{13}{9},\frac{5}{9},2}(\psi\mathfrak{f}^{\frac{13}{9}},\mathfrak{f}^{\frac{5}{9}}) \ominus (-1)(4,5,8) \odot (\mathfrak{r}^2+1) \odot \mathbb{E}_{\frac{13}{9},\frac{5}{9},1}$$

$$(\psi\mathfrak{f}^{\frac{13}{9}},\mathfrak{f}^{\frac{5}{9}}) \oplus (-1)(4,5,8) \odot (\mathfrak{r}^2+1) \odot \mathfrak{f}^{\frac{5}{9}}\mathbb{E}_{\frac{13}{9},\frac{5}{9},\frac{14}{9}}(\psi\mathfrak{f}^{\frac{13}{9}},\mathfrak{f}^{\frac{5}{9}}) \oplus (-1)(\mathbb{I}^{1;\psi,1}_{\frac{13}{9},\frac{5}{9},\frac{13}{9}}[(1+\psi)$$

$$\odot \mathbb{S}^2(\mathfrak{r},\mathfrak{f}) \ominus \mathbb{S}^3(\mathfrak{r},\mathfrak{f})])(\mathfrak{f}). \tag{85}$$

The cases of fuzzy solutions based on the types of differentiability are given in the Table 2.The graphical analysis of cases (*a*) and (*b*) is given by the Figs 1 and 2 respectively which are showing the behavior with respect to fractional orders $\varsigma = \frac{13}{9}$ and $\gamma = \frac{8}{9}$. We will present the discussion of these figures one another from left to right sequence. The figures in sequence from left to right show specific details regarding the effects of these fractional parameters on the Fuzzy-Valued Function. A three-dimensional display in the Figs 1 and 2 shows how the fuzzy solutions of case (*a*) and (*b*) changes throughout its entire domain while encompassing both the fractional orders. The graphical representation demonstrated in the above cases depicts that the fuzzy solutions are consistent and coherent for various values of parameters involved. At every point of FFGFH-NDEs, the solutions are fuzzy valued to demonstrate the consistent behavior. The structure of the solutions deeply connects the medelled profile, in particular when both the fuzziness and fractional orders are jointly considered.

(c) If a FVF $\mathbb{S}(\mathfrak{r},\mathfrak{f})$ is *FFCgH*-differentiable in its $\mathbb{S}^{\circledR} - \mathbb{F}^{\divideontimes}$ and ${}^{CF}_{gH}\mathfrak{D}^{\varsigma}_{+}\mathbb{S}(\mathfrak{r},\mathfrak{f}), {}^{CF}_{gH}\mathfrak{D}^{\gamma}_{+}\mathbb{S}(\mathfrak{r},\mathfrak{f})$ are *FFCgH*-differentiable in its $\mathbb{F}^{\divideontimes} - \mathbb{F}^{\divideontimes}$, then the problem (1) has solution of the form

$$\mathbb{S}(\mathfrak{r},\mathfrak{f}) = (4,5,8) \odot (\mathfrak{r}^2+1) \odot \mathbb{E}_{\frac{13}{9},\frac{5}{9},1}(\psi\mathfrak{f}^{\frac{13}{9}},\mathfrak{f}^{\frac{5}{9}}) \oplus (-3,-1,5) \odot (\mathfrak{r}-1) \odot \mathfrak{f}\mathbb{E}_{\frac{13}{9},\frac{5}{9},\frac{14}{9}}$$

$$(\psi\mathfrak{f}^{\frac{13}{9}},\mathfrak{f}^{\frac{5}{9}}) \oplus (-1)(4,5,8) \odot (\mathfrak{r}^2+1) \odot \mathfrak{f}^{\frac{5}{9}}\mathbb{E}_{\frac{13}{9},\frac{5}{9},\frac{14}{9}}(\psi\mathfrak{f}^{\frac{13}{9}},\mathfrak{f}^{\frac{5}{9}}) \oplus (-1)(\mathbb{I}^{1;\psi,1}_{\frac{13}{9},\frac{5}{9},\frac{13}{9}}[(1+\psi)$$

$$\odot \mathbb{S}^2(\mathfrak{r},\mathfrak{f}) \ominus \mathbb{S}^3(\mathfrak{r},\mathfrak{f})])(\mathfrak{f}). \tag{86}$$

**Table 2. Summary table of cases of *FFCgH*-differentiability.**

| $\mathbb{S}(\mathfrak{r},\mathfrak{f})$ | ${}^{CF}_{gH}\mathfrak{D}^{\varsigma}_{+}\mathbb{S}(\mathfrak{r},\mathfrak{f})$ | ${}^{CF}_{gH}\mathfrak{D}^{\gamma}_{+}\mathbb{S}(\mathfrak{r},\mathfrak{f})$ |
|---|---|---|
| $\mathbb{F}^{\divideontimes} - \mathbb{F}^{\divideontimes}$ | $\mathbb{F}^{\divideontimes} - \mathbb{F}^{\divideontimes}$ | $\mathbb{F}^{\divideontimes} - \mathbb{F}^{\divideontimes}$ |
| $\mathbb{F}^{\divideontimes} - \mathbb{F}^{\divideontimes}$ | $\mathbb{S}^{\circledR} - \mathbb{F}^{\divideontimes}$ | $\mathbb{S}^{\circledR} - \mathbb{F}^{\divideontimes}$ |
| $\mathbb{S}^{\circledR} - \mathbb{F}^{\divideontimes}$ | $\mathbb{F}^{\divideontimes} - \mathbb{F}^{\divideontimes}$ | $\mathbb{F}^{\divideontimes} - \mathbb{F}^{\divideontimes}$ |
| $\mathbb{F}^{\divideontimes} - \mathbb{F}^{\divideontimes}$ | $\mathbb{S}^{\circledR} - \mathbb{F}^{\divideontimes}$ | $\mathbb{F}^{\divideontimes} - \mathbb{F}^{\divideontimes}$ |
| $\mathbb{F}^{\divideontimes} - \mathbb{F}^{\divideontimes}$ | $\mathbb{F}^{\divideontimes} - \mathbb{F}^{\divideontimes}$ | $\mathbb{S}^{\circledR} - \mathbb{F}^{\divideontimes}$ |
| $\mathbb{S}^{\circledR} - \mathbb{F}^{\divideontimes}$ | $\mathbb{S}^{\circledR} - \mathbb{F}^{\divideontimes}$ | $\mathbb{S}^{\circledR} - \mathbb{F}^{\divideontimes}$ |

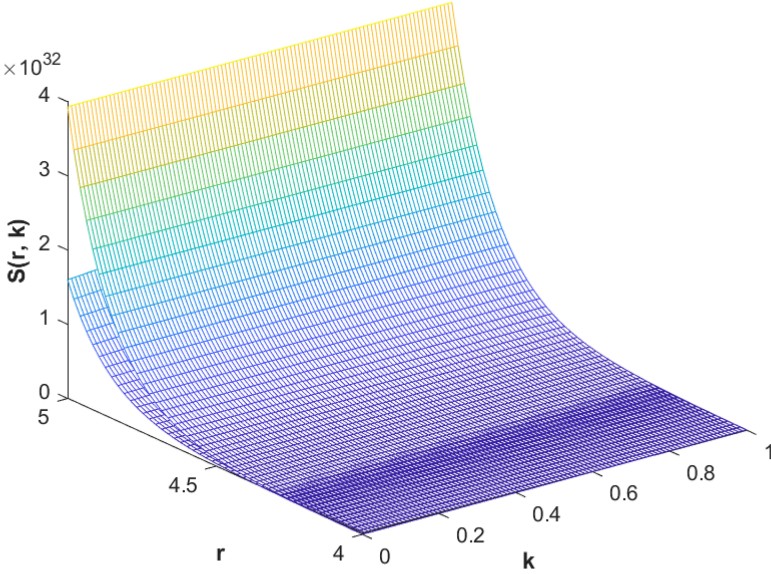

**Fig 1. Graphical representation of case (a) for** $\varsigma = \dfrac{13}{9}; \gamma = \dfrac{8}{9}$ **and** $\psi = 0.95$.

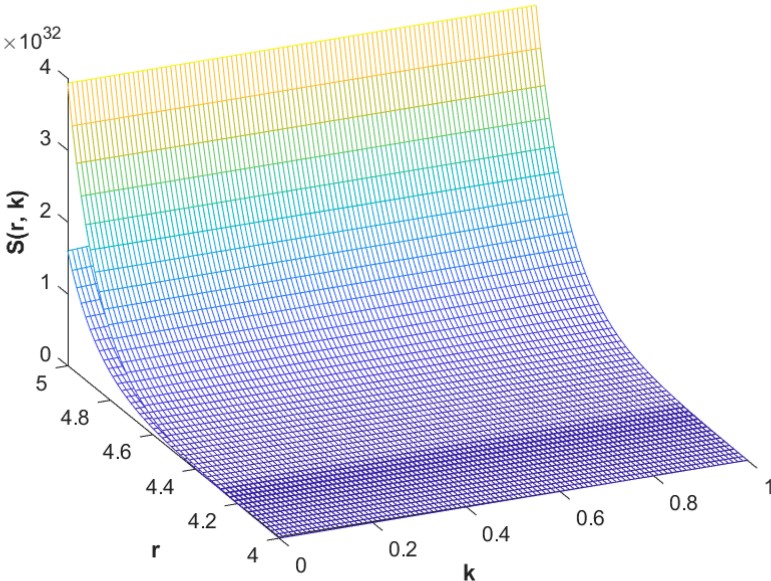

**Fig 2. Graphical representation of case (b) for** $\varsigma = \dfrac{13}{9}; \gamma = \dfrac{8}{9}$ **and** $\psi = 0.95$.

(d) If a FVF $\mathbb{S}(\mathfrak{r},\mathfrak{k})$, $_{gH}^{\mathrm{CF}}\mathfrak{D}_+^{\gamma}\mathbb{S}(\mathfrak{r},\mathfrak{k})$ and $_{gH}^{\mathrm{CF}}\mathfrak{D}_+^{\varsigma}\mathbb{S}(\mathfrak{r},\mathfrak{k})$ are *FFCgH*-differentiable in its $\mathbb{S}^{\circledR} - \mathbb{F}^{*}$, then the problem (1) has solution of the form

$$\mathbb{S}(\mathfrak{r},\mathfrak{k}) = (4,5,8) \odot (\mathfrak{r}^2+1) \odot \mathbb{E}_{\frac{13}{9},\frac{5}{9},1} (\psi\mathfrak{k}^{\frac{13}{9}},\mathfrak{k}^{\frac{5}{9}}) \ominus (-1)(-3,-1,5) \odot (\mathfrak{r}-1) \odot \mathfrak{k}\mathbb{E}_{\frac{13}{9},\frac{5}{9},\frac{5}{9}+2}$$

$$(\psi\mathfrak{k}^{\frac{13}{9}},\mathfrak{k}^{\frac{5}{9}}) \ominus (4,5,8) \odot (\mathfrak{r}^2+1) \odot \mathfrak{k}^{\frac{5}{9}}\mathbb{E}_{\frac{13}{9},\frac{5}{9},\frac{14}{9}} (\psi\mathfrak{k}^{\frac{13}{9}},\mathfrak{k}^{\frac{5}{9}}) \ominus (\mathbb{I}_{\frac{13}{9},\frac{5}{9},\frac{13}{9}}^{1;\psi,1}[(1+\psi) \odot \mathbb{S}^2(\mathfrak{r},\mathfrak{k})$$

$$\ominus \mathbb{S}^3(\mathfrak{r},\mathfrak{k})])(\mathfrak{k}). \tag{87}$$

The graphical analysis of cases (*c*) and (*d*) is given by the Figs 3 and 4 which are showing the behavior with respect to fractional orders $\varsigma = \dfrac{13}{9}$ and $\gamma = \dfrac{8}{9}$. A three-dimensional display in the Figs 3 and 4 shows how the fuzzy solutions of case (*c*) and (*d*) changes throughout its entire domain while encompassing both the fractional orders. The graphical representation demonstrated in the above cases depicts that the fuzzy solutions are consistent and coherent for various values of parameters involved. At every point of FFGFH-NDEs, the solutions are fuzzy valued to demonstrate the consistent behavior. The structure of the solutions deeply connects the medelled profile, in particular when both the fuzziness and fractional orders are jointly considered.

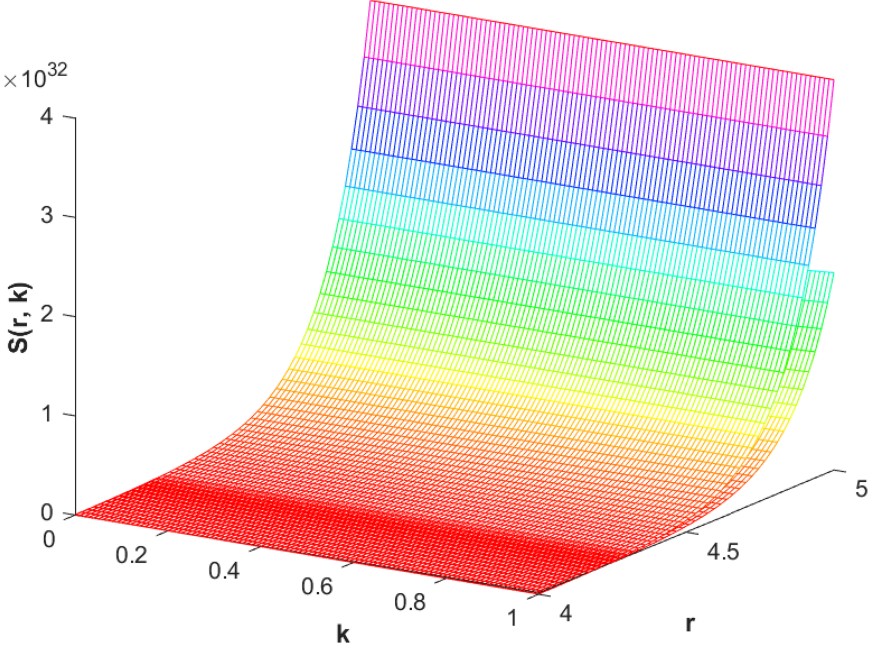

**Fig 3**. Graphical representation of case (c) for $\varsigma = \dfrac{13}{9}$; $\gamma = \dfrac{8}{9}$ and $\psi = 0.85$.

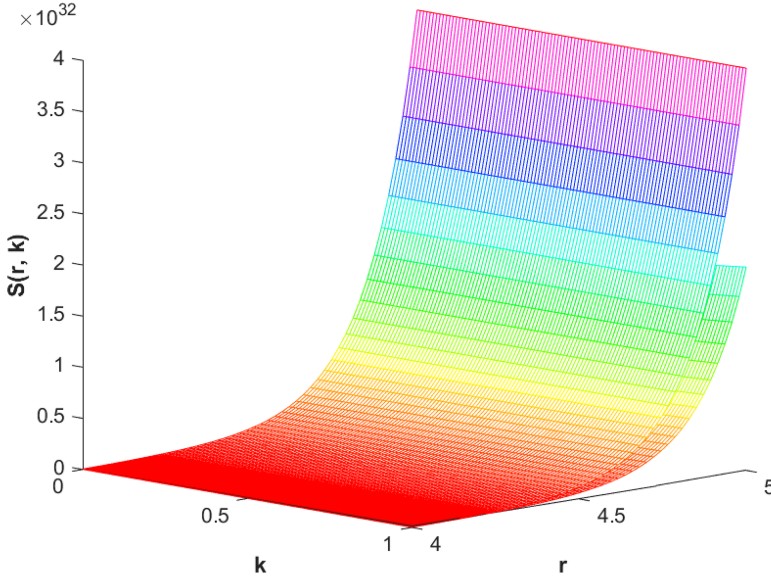

**Fig 4. Graphical representation of case (d) for** $\varsigma = \dfrac{13}{9}; \gamma = \dfrac{8}{9}$ **and** $\psi = 0.85$.

(e) If a FVF $\mathbb{S}(\mathfrak{r},\mathfrak{k}), {}^{\mathrm{CF}}_{gH}\mathfrak{D}^{\gamma}_{+}\mathbb{S}(\mathfrak{r},\mathfrak{k})$ are *FFCgH*-differentiable in its $\mathbb{F}^{\circledast} - \mathbb{F}^{\circledast}$ and ${}^{\mathrm{CF}}_{gH}\mathfrak{D}^{\varsigma}_{+}\mathbb{S}(\mathfrak{r},\mathfrak{k})$ is *FFCgH*-differentiable in its $\mathbb{S}^{\circledR} - \mathbb{F}^{\circledast}$,then the problem (1) has solution of the form

$$\mathbb{S}(\mathfrak{r},\mathfrak{k}) = (-3,-1,5) \odot (\mathfrak{r}-1) \odot \mathfrak{k} \mathbb{E}_{\frac{5}{9},\frac{5}{9},2}(\psi \mathfrak{k}^{\frac{13}{9}}, \mathfrak{k}^{\frac{5}{9}}) \oplus (-1)(4,5,8) \odot (\mathfrak{r}^2+1) \odot \mathbb{E}_{\frac{13}{9},\frac{5}{9},1}$$

$$(\psi \mathfrak{k}^{\frac{13}{9}}, \mathfrak{k}^{\frac{5}{9}}) \ominus (4,5,8) \odot (\mathfrak{r}^2+1) \odot \mathfrak{k}^{\frac{5}{9}} \mathbb{E}_{\frac{13}{9},\frac{5}{9},\frac{14}{9}}(\psi \mathfrak{k}^{\frac{13}{9}}, \mathfrak{k}^{\frac{5}{9}}) \ominus (\mathbb{I}^{1;\psi,1}_{\frac{13}{9},\frac{5}{9},\frac{13}{9}}[(1+\psi) \odot \mathbb{S}^2(\mathfrak{r},\mathfrak{k})$$

$$\ominus \mathbb{S}^3(\mathfrak{r},\mathfrak{k})])(\mathfrak{k}). \tag{88}$$

(f) If a FVF $\mathbb{S}(\mathfrak{r},\mathfrak{k}), {}^{\mathrm{CF}}_{gH}\mathfrak{D}^{\frac{13}{9}}_{+}\mathbb{S}(\mathfrak{r},\mathfrak{k})$ are *FFCgH*-differentiable in its $\mathbb{F}^{\circledast} - \mathbb{F}^{\circledast}$ and ${}^{\mathrm{CF}}_{gH}\mathfrak{D}^{\gamma}_{+}\mathbb{S}(\mathfrak{r},\mathfrak{k})$ is *FFCgH*-differentiable in its $\mathbb{S}^{\circledR} - \mathbb{F}^{\circledast}$, then the problem (1) has solution of the form

$$\mathbb{S}(\mathfrak{r},\mathfrak{k}) = (4,5,8) \odot (\mathfrak{r}^2+1) \odot \mathbb{E}_{\frac{13}{9},\frac{5}{9},1}(\psi \mathfrak{k}^{\frac{13}{9}}, -\mathfrak{k}^{\frac{5}{9}}) \oplus (-3,-1,5) \odot (\mathfrak{r}-1) \odot \mathfrak{k} \mathbb{E}_{\frac{13}{9},\frac{5}{9},2}$$

$$(\psi \mathfrak{k}^{\frac{13}{9}}, -\mathfrak{k}^{\frac{5}{9}}) \oplus (4,5,8) \odot (\mathfrak{r}^2+1) \odot \mathfrak{k}^{\frac{5}{9}} \mathbb{E}_{\frac{13}{9},\frac{5}{9},\frac{14}{9}}(\psi \mathfrak{k}^{\frac{13}{9}}, -\mathfrak{k}^{\frac{5}{9}}) \ominus (\mathbb{I}^{1;\psi,1}_{\frac{13}{9},\frac{5}{9},\frac{13}{9}}[(1+\psi) \odot$$

$$\mathbb{S}^2(\mathfrak{r},\mathfrak{k}) \ominus \mathbb{S}^3(\mathfrak{r},\mathfrak{k})])(\mathfrak{k}). \tag{89}$$

The graphical analysis of cases (*e*) and (*f*) is given by the Figs 5 and 6 respectively, which show the behavior with respect to fractional orders $\varsigma = \dfrac{13}{9}$ and $\gamma = \dfrac{8}{9}$. We will present the discussion of these figures one another from left to right sequence. The figures in sequence from left to right show specific details regarding the effects of these fractional

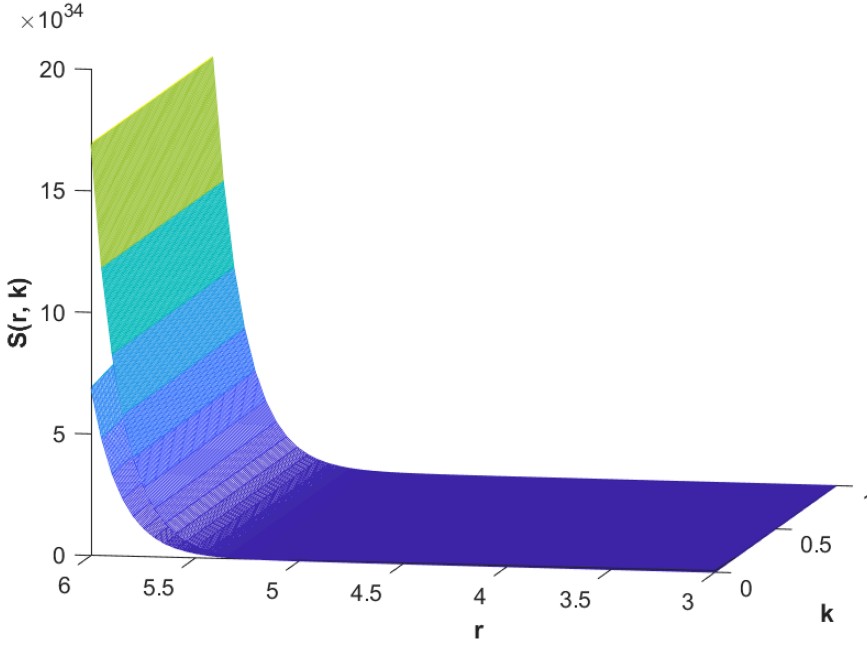

**Fig 5. Graphical representation of case (e) for** $\varsigma = \dfrac{13}{9}; \gamma = \dfrac{8}{9}$ **and** $\psi = 0.75$.

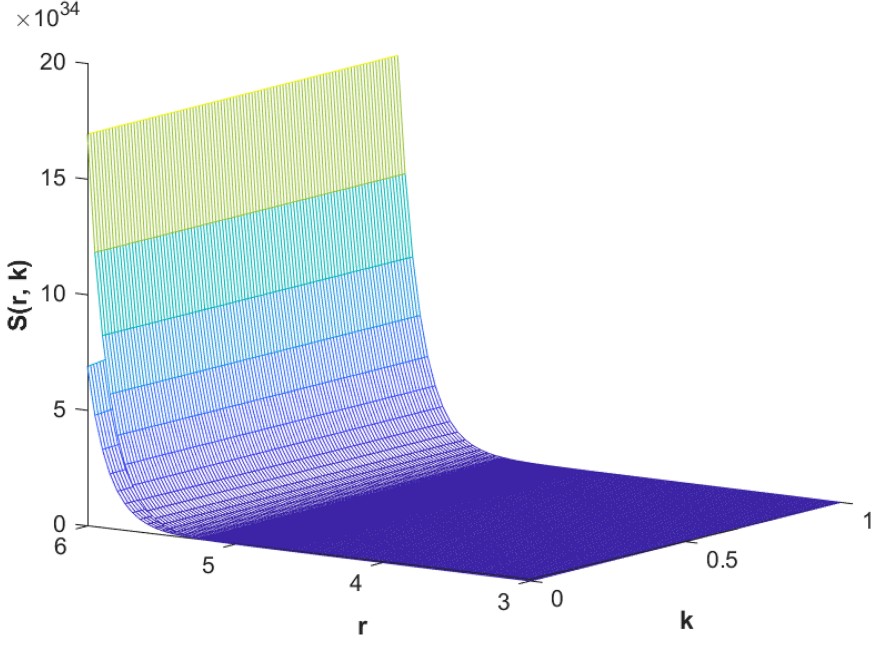

**Fig 6. Graphical representation of case (f) for** $\varsigma = \dfrac{13}{9}; \gamma = \dfrac{8}{9}$ **and** $\psi = 0.75$.

parameters on the FVFs. A three-dimensional display in the Figs 5 and 6 shows how the fuzzy solutions of case (*e*) and (*f*) changes throughout its entire domain while encompassing both the fractional orders. The graphical representation demonstrated in the above cases depicts that the fuzzy solutions are consistent and coherent for various values of parameters involved. At every point of FFGFH-NDEs, the solutions are fuzzy valued to demonstrate the consistent behavior. The structure of the solutions deeply connects the medelled profile, in particular when both the fuzziness and fractional orders are jointly considered. By fixing the fractional order $\gamma = 1$, the crisp solutions of the Example (29) can be determined. Fuzzified fractional models serve to represent system uncertainties and physical tolerances according to the research making them suitable for digital circuit theory and biological modeling. The graphical solution of the FFGFH-NDEs allows readers to grasp the uncertainty levels and tolerance ranges of the solutions in the much better way.

**Example 30.** Consider the FFGFH-NDE (1) for $\gamma = \dfrac{3}{5}; \varsigma = \dfrac{7}{4}$ along with the initial conditions $(\mathbb{S}_1, \mathbb{S}_2, \mathbb{S}_3) \odot \tilde{\ell}(\mathfrak{r}) = (1, 7, 9) \odot (\mathfrak{r}^2 - 1)$ and $(\mathbb{S}_1, \mathbb{S}_2, \mathbb{S}_3) \odot \tilde{j}(\mathfrak{r}) = (2, 4, 7) \odot (\mathfrak{r} + 1)$. Then using Theorem 19 and 21, the FFGFH-NDE (1) contains the following forms of solutions:

1. For $\psi = 1$
   (a) If a FVF $\mathbb{S}(\mathfrak{r}, \mathfrak{f}), {}_{gH}^{\mathbb{CF}}\mathfrak{D}_+^{\gamma}\mathbb{S}(\mathfrak{r}, \mathfrak{f})$ and ${}_{gH}^{\mathbb{CF}}\mathfrak{D}_+^{\varsigma}\mathbb{S}(\mathfrak{r}, \mathfrak{f})$ are *FFCgH-differentiable* in its $\mathbb{F}^* - \mathbb{F}^*$, then the system (1) has solution of the form

$$\mathbb{S}(\mathfrak{r}, \mathfrak{f}) = (1, 7, 9) \odot (\mathfrak{r}^2 - 1) \odot \mathbb{E}_{\frac{7}{4}, \frac{23}{20}, 1}(\mathfrak{f}^{\frac{7}{4}}, \mathfrak{f}^{\frac{23}{20}}) \oplus (2, 4, 7) \odot (\mathfrak{r} + 1) \odot \mathfrak{f} \mathbb{E}_{\frac{7}{4}, \frac{23}{20}, 2}$$

$$(\mathfrak{f}^{\frac{7}{4}}, \mathfrak{f}^{\frac{23}{20}}) \ominus (1, 7, 9) \odot (\mathfrak{r}^2 - 1) \odot \mathfrak{f}^{\frac{23}{20}} \mathbb{E}_{\frac{7}{4}, \frac{23}{20}, \frac{43}{20}}(\mathfrak{f}^{\frac{7}{4}}, \mathfrak{f}^{\frac{23}{20}}) \ominus (-1)(\mathbb{I}_{\frac{7}{4}, \frac{23}{20}, \frac{7}{4}}^{1;1,1} [\mathbb{S}^3(\mathfrak{r}, \mathfrak{f})$$

$$\ominus 2\mathbb{S}^2(\mathfrak{r}, \mathfrak{f})])(\mathfrak{f}). \tag{90}$$

   (b) If a FVF $\mathbb{S}(\mathfrak{r}, \mathfrak{f})$ is *FFCgH-differentiable* in its $\mathbb{F}^* - \mathbb{F}^*$ and ${}_{gH}^{\mathbb{CF}}\mathfrak{D}_+^{\varsigma}\mathbb{S}(\mathfrak{r}, \mathfrak{f}), {}_{gH}^{\mathbb{CF}}\mathfrak{D}_+^{\gamma}\mathbb{S}(\mathfrak{r}, \mathfrak{f})$ are *FFCgH-differentiable* in its $\mathbb{S}^{\circledR} - \mathbb{F}^*$, then the system (1) has solution of the form

$$\mathbb{S}(\mathfrak{r}, \mathfrak{f}) = (2, 4, 7) \odot (\mathfrak{r} + 1) \odot \mathfrak{f} \odot \mathbb{E}_{\frac{7}{4}, \frac{23}{20}, 2}(\mathfrak{f}^{\frac{7}{4}}, \mathfrak{f}^{\frac{23}{20}}) \ominus (-1)(1, 7, 9) \odot (\mathfrak{r}^2 - 1) \odot \mathbb{E}_{\frac{7}{4}, \frac{23}{20}, 1}$$

$$(\mathfrak{f}^{\frac{7}{4}}, \mathfrak{f}^{\frac{23}{20}}) \oplus (-1)(1, 7, 9) \odot (\mathfrak{r}^2 - 1) \odot \mathfrak{f}^{\frac{23}{20}} \mathbb{E}_{\frac{7}{4}, \frac{23}{20}, \frac{43}{20}}(\mathfrak{f}^{\frac{7}{4}}, \mathfrak{f}^{\frac{23}{20}}) \ominus (-1)(\mathbb{I}_{\frac{7}{4}, \frac{23}{20}, \frac{7}{4}}^{1;1,1} [$$

$$\mathbb{S}^3(\mathfrak{r}, \mathfrak{f}) \ominus 2\mathbb{S}^2(\mathfrak{r}, \mathfrak{f})](\mathfrak{f}). \tag{91}$$

(c) If a FVF $\mathbb{S}(\mathfrak{r},\mathfrak{f})$ is *FFCgH*-differentiable in its $\mathbb{S}^{\circledR} - \mathbb{F}^{\circledast}$ and ${}^{\mathrm{CF}}_{gH}\mathfrak{D}^{\varsigma}_{+}\mathbb{S}(\mathfrak{r},\mathfrak{f}), {}^{\mathrm{CF}}_{gH}\mathfrak{D}^{\gamma}_{+}\mathbb{S}(\mathfrak{r},\mathfrak{f})$ are *FFCgH*-differentiable in its $\mathbb{F}^{\circledast} - \mathbb{F}^{\circledast}$, then the system (1) has solution of the form

$$\mathbb{S}(\mathfrak{r},\mathfrak{f}) = (2,4,7) \odot (\mathfrak{r}+1) \odot \mathfrak{f} \odot \mathbb{E}_{\frac{7}{4},\frac{23}{20},2}(\mathfrak{f}^{\frac{7}{4}},\mathfrak{f}^{\frac{23}{20}}) \oplus (1,7,9) \odot (\mathfrak{r}^2-1) \odot \mathbb{E}_{\frac{7}{4},\frac{23}{20},1}$$

$$(\mathfrak{f}^{\frac{7}{4}},\mathfrak{f}^{\frac{23}{20}}) \oplus (-1)(1,7,9) \odot (\mathfrak{r}^2-1) \odot \mathfrak{f}^{\frac{23}{20}} \mathbb{E}_{\frac{7}{4},\frac{23}{20},\frac{43}{20}}(\mathfrak{f}^{\frac{7}{4}},\mathfrak{f}^{\frac{23}{20}}) \ominus (-1)(\mathbb{I}^{1;1,1}_{\frac{7}{4},\frac{23}{20},\frac{7}{4}}[$$

$$\mathbb{S}^3(\mathfrak{r},\mathfrak{f}) \ominus 2\mathbb{S}^2(\mathfrak{r},\mathfrak{f})])(\mathfrak{f}). \tag{92}$$

(d) If a FVF $\mathbb{S}(\mathfrak{r},\mathfrak{f}), {}^{\mathrm{CF}}_{gH}\mathfrak{D}^{\gamma}_{+}\mathbb{S}(\mathfrak{r},\mathfrak{f})$ and ${}^{\mathrm{CF}}_{gH}\mathfrak{D}^{\varsigma}_{+}\mathbb{S}(\mathfrak{r},\mathfrak{f})$ are *FFCgH*-differentiable in its $\mathbb{S}^{\circledR} - \mathbb{F}^{\circledast}$, then the system (1) has solution of the form

$$\mathbb{S}(\mathfrak{r},\mathfrak{f}) = (-1)(1,7,9) \odot (\mathfrak{r}^2-1) \odot \mathbb{E}_{\frac{7}{4},\frac{23}{20},1}(\mathfrak{f}^{\frac{7}{4}},\mathfrak{f}^{\frac{23}{20}}) \oplus (2,4,7) \odot (\mathfrak{r}+1) \odot \mathfrak{f} \mathbb{E}_{\frac{7}{4},\frac{23}{20},2}$$

$$(\mathfrak{f}^{\frac{7}{4}},\mathfrak{f}^{\frac{23}{20}}) \ominus (-1)(1,7,9) \odot (\mathfrak{r}^2-1) \odot \mathfrak{f}^{\frac{23}{20}} \mathbb{E}_{\frac{7}{4},\frac{23}{20},\frac{43}{20}}((-1)\mathfrak{f}^{\frac{7}{4}},\mathfrak{f}^{\frac{23}{20}}) \ominus (-1)(\mathbb{I}^{1;1,1}_{\gamma,\frac{23}{20},\gamma}$$

$$[\mathbb{S}^3(\mathfrak{r},\mathfrak{f}) \ominus 2\mathbb{S}^2(\mathfrak{r},\mathfrak{f})])(\mathfrak{f}). \tag{93}$$

(e) If a FVF $\mathbb{S}(\mathfrak{r},\mathfrak{f}), {}^{\mathrm{CF}}_{gH}\mathfrak{D}^{\gamma}_{+}\mathbb{S}(\mathfrak{r},\mathfrak{f})$ are *FFCgH*-differentiable in its $\mathbb{F}^{\circledast} - \mathbb{F}^{\circledast}$ and ${}^{\mathrm{CF}}_{gH}\mathfrak{D}^{\varsigma}_{+}\mathbb{S}(\mathfrak{r},\mathfrak{f})$ is *FFCgH*-differentiable in its $\mathbb{S}^{\circledR} - \mathbb{F}^{\circledast}$, then the system (1) has solution of the form

$$\mathbb{S}(\mathfrak{r},\mathfrak{f}) = (2,4,7) \odot (\mathfrak{r}+1) \odot \mathfrak{f} \mathbb{E}_{\frac{7}{4},\frac{23}{20},2}(\mathfrak{f}^{\frac{7}{4}},\mathfrak{f}^{\frac{23}{20}}) \oplus (-1)(1,7,9) \odot (\mathfrak{r}^2-1) \odot \mathbb{E}_{\frac{7}{4},\frac{23}{20},1}$$

$$(\mathfrak{f}^{\frac{7}{4}},\mathfrak{f}^{\frac{23}{20}}) \ominus (1,7,9) \odot (\mathfrak{r}^2-1) \odot \mathfrak{f}^{\frac{23}{20}} \mathbb{E}_{\frac{7}{4},\frac{23}{20},\frac{43}{20}}(\mathfrak{f}^{\frac{7}{4}},\mathfrak{f}^{\frac{23}{20}}) \ominus (\mathbb{I}^{1;1,1}_{\frac{7}{4},\frac{23}{20},\frac{7}{4}}[2 \odot \mathbb{S}^2(\mathfrak{r},\mathfrak{f})$$

$$\ominus \mathbb{S}^3(\mathfrak{r},\mathfrak{f})])(\mathfrak{f}). \tag{94}$$

(f) If a FVF $\mathbb{S}(\mathfrak{r},\mathfrak{f}), {}^{\mathrm{CF}}_{gH}\mathfrak{D}^{\varsigma}_{+}\mathbb{S}(\mathfrak{r},\mathfrak{f})$ are *FFCgH*-differentiable in its $\mathbb{F}^{\circledast} - \mathbb{F}^{\circledast}$ and ${}^{\mathrm{CF}}_{gH}\mathfrak{D}^{\gamma}_{+}\mathbb{S}(\mathfrak{r},\mathfrak{f})$ is *FFCgH*-differentiable in its $\mathbb{S}^{\circledR} - \mathbb{F}^{\circledast}$, then the system (1) has solution of the form

$$\mathbb{S}(\mathfrak{r},\mathfrak{k}) = (1,7,9) \odot (\mathfrak{r}^2 - 1) \odot \mathbb{E}_{\frac{7}{4},\frac{23}{20},1}(\psi\mathfrak{k}^{\frac{7}{4}}, -\mathfrak{k}^{\frac{23}{20}}) \oplus (2,4,7) \odot \mathfrak{k}\mathbb{E}_{\frac{7}{4},\frac{23}{20},2}$$

$$(\mathfrak{k}^{\frac{7}{4}}, -\mathfrak{k}^{\frac{23}{20}}) \oplus (1,7,9) \odot (\mathfrak{r}^2 - 1) \odot \mathfrak{k}^{\frac{23}{20}} \mathbb{E}_{\frac{7}{4},\frac{23}{20},\frac{43}{20}}(\mathfrak{k}^{\frac{7}{4}}, -\mathfrak{k}^{\frac{23}{20}}) \ominus (\mathbb{I}^{1;1,1}_{\frac{7}{4},\frac{23}{20},\frac{7}{4}}[2\odot$$

$$\mathbb{S}^2(\mathfrak{r},\mathfrak{k}) \ominus \mathbb{S}^3(\mathfrak{r},\mathfrak{k})])(\mathfrak{k}). \tag{95}$$

The fuzzy solutions of FFGFH-NDEs for the cases (*a*), (*b*), (*c*), (*d*), (*e*) and (*f*) are of significant importance as these solutions combine fuzziness with fractional dynamics hence offering a more realistic model of excitable media where variability of parameters are together with a long memory term. The parametrically varied fuzzy solution takes into consideration not only uncertainty in the initial condition but in the parameters of the associated system such as threshold function. Contrasting the crisp solution, where a single deterministic solution is produced, the fuzzified solutions of the FFGFH-NDE (1) for $\gamma = \frac{3}{5}; \varsigma = \frac{7}{4}$ are more reliable because they produces a family of solutions that has interval values taking account of uncertainty in both the spatial and time dynamics of the system. Similarly, we can construct the fuzzy solutions of FFGFH-NDEs for $\psi = 0$ in the same manner for the different cases of fuzzy differentiability.

**Example 31.** Consider the FFGFH-NDE (1) for $\gamma = \frac{4}{3}; \varsigma = \frac{5}{3}$ along with the initial conditions $(\mathbb{S}_1, \mathbb{S}_2, \mathbb{S}_3) \odot \tilde{\ell}(\mathfrak{r}) = (7,9,11) \odot (5 + \mathfrak{k})$ and $(\mathbb{S}_1, \mathbb{S}_2, \mathbb{S}_3) \odot \tilde{j}(\mathfrak{r}) = (2,4,7) \odot (2 - \mathfrak{r}^2)$. Then from Theorem 23, the FFGFH-NDE (1) contains the following forms of solutions:

(a) If a FVF $\mathbb{S}(\mathfrak{r},\mathfrak{k}), {}^{\mathrm{CF}}_{gH}\mathfrak{D}^\gamma_+\mathbb{S}(\mathfrak{r},\mathfrak{k})$ and ${}^{\mathrm{CF}}_{gH}\mathfrak{D}^\varsigma_+\mathbb{S}(\mathfrak{r},\mathfrak{k})$ are *FFCgH*-differentiable in its $\mathbb{F}^* - \mathbb{F}^*$, then the system (1) contains the solution which is given as

$$\mathbb{S}(\mathfrak{r},\mathfrak{k}) = (7,9,11) \odot (5+\mathfrak{r}) \odot \mathbb{E}_{\frac{5}{3},\frac{1}{3},1}(\psi\mathfrak{k}^{\frac{5}{3}}, \mathfrak{k}^{\frac{1}{3}}) \oplus (2,4,7) \odot (2-\mathfrak{r}^2) \odot \mathfrak{k} \odot \mathbb{E}_{\frac{5}{3},\frac{1}{3},2}(\psi\mathfrak{k}^{\frac{5}{3}}, \mathfrak{k}^{\frac{1}{3}}) \ominus$$

$$(7,9,11) \odot (5+\mathfrak{k}) \odot \odot \mathbb{E}_{\frac{5}{3},\frac{1}{3},\frac{4}{3}}(\psi\mathfrak{k}^{\frac{5}{3}}, \mathfrak{k}^{\frac{1}{3}}) \ominus (2,4,7) \odot (2-\mathfrak{r}^2) \odot \mathfrak{k}^{\frac{4}{3}} \odot \mathbb{E}_{\frac{5}{3},\frac{1}{3},\frac{7}{3}}(\psi\mathfrak{k}^{\frac{5}{3}}, \mathfrak{k}^{\frac{1}{3}}) \ominus (\mathbb{I}^{1;\psi,1}_{\frac{5}{3},\frac{1}{3},\frac{5}{3}}$$

$$[(1 + \psi) \odot \mathbb{S}^2(\mathfrak{r},\mathfrak{k}) \ominus \mathbb{S}^3(\mathfrak{r},\mathfrak{k})])(\mathfrak{k}). \tag{96}$$

(b) If a FVF $\mathbb{S}(\mathfrak{r},\mathfrak{k})$ is *FFCgH*-differentiable in its $\mathbb{F}^* - \mathbb{F}^*$ and ${}^{\mathrm{CF}}_{gH}\mathfrak{D}^\varsigma_+\mathbb{S}(\mathfrak{r},\mathfrak{k}), {}^{\mathrm{CF}}_{gH}\mathfrak{D}^\gamma_+\mathbb{S}(\mathfrak{r},\mathfrak{k})$ are *FFCgH*-differentiable in its $\mathbb{S}^\circledast - \mathbb{F}^*$, then the system (1) contains the solution which is given as

$$\mathbb{S}(\mathfrak{r},\mathfrak{f}) = (2,4,7) \odot (2-\mathfrak{r}^2) \odot \mathfrak{f} \odot \mathbb{E}_{\frac{5}{3},\frac{1}{3},2}(\psi\mathfrak{f}^{\frac{5}{3}},\mathfrak{f}^{\frac{1}{3}}) \ominus (-1)(7,9,11) \odot (5+\mathfrak{r}) \odot \mathbb{E}_{\frac{5}{3},\frac{1}{3},1}(\psi\mathfrak{f}^{\frac{5}{3}},\mathfrak{f}^{\frac{1}{3}}) \oplus$$

$$(-1)(7,9,11) \odot (5+\mathfrak{f}) \odot \mathfrak{f}^{\frac{1}{3}} \odot \mathbb{E}_{\frac{5}{3},\frac{1}{3},\frac{4}{3}}(\psi\mathfrak{f}^{\frac{5}{3}},\mathfrak{f}^{\frac{1}{3}}) \oplus (-1)(2,4,7) \odot (2-\mathfrak{r}^2) \odot \mathfrak{f}^{\frac{4}{3}} \odot \mathbb{E}_{\frac{5}{3},\frac{1}{3},\frac{7}{3}}(\psi\mathfrak{f}^{\frac{5}{3}},\mathfrak{f}^{\frac{1}{3}})$$

$$\oplus (-1)(\mathbb{I}^{1;\psi,1}_{\frac{5}{3},\frac{1}{3},\frac{5}{3}}[(1+\psi) \odot \mathbb{S}^2(\mathfrak{r},\mathfrak{f}) \ominus \mathbb{S}^3(\mathfrak{r},\mathfrak{f})])(\mathfrak{f}). \tag{97}$$

(c) If a FVF $\mathbb{S}(\mathfrak{r},\mathfrak{f})$ is *FFCgH-differentiable* in its $\mathbb{S}^{\circledR} - \mathbb{F}^{\divideontimes}$ and $^{\mathrm{CF}}_{gH}\mathfrak{D}^{\varsigma}_{+}\mathbb{S}(\mathfrak{r},\mathfrak{f})$, $^{\mathrm{CF}}_{gH}\mathfrak{D}^{\gamma}_{+}\mathbb{S}(\mathfrak{r},\mathfrak{f})$ are *FFCgH-differentiable* in its $\mathbb{F}^{\divideontimes} - \mathbb{F}^{\divideontimes}$, then the system (1) contains the solution which is given as

$$\mathbb{S}(\mathfrak{r},\mathfrak{f}) = (7,9,11) \odot (5+\mathfrak{r}) \odot \mathbb{E}_{\frac{5}{3},\frac{1}{3},1}(\psi\mathfrak{f}^{\frac{5}{3}},\mathfrak{f}^{\frac{1}{3}}) \oplus (2,4,7) \odot (2-\mathfrak{r}^2) \odot \mathfrak{f} \odot \mathbb{E}_{\frac{5}{3},\frac{1}{3},\frac{4}{3}}(\psi\mathfrak{f}^{\frac{5}{3}},\mathfrak{f}^{\frac{1}{3}}) \oplus$$

$$(-1)(7,9,11) \odot (5+\mathfrak{r}) \odot \mathfrak{f}^{\frac{1}{3}} \odot \mathbb{E}_{\frac{5}{3},\frac{1}{3},\frac{4}{3}}(\psi\mathfrak{f}^{\frac{5}{3}},\mathfrak{f}^{\frac{1}{3}}) \ominus (2,4,7) \odot (2-\mathfrak{r}^2) \odot \mathfrak{f}^{\frac{4}{3}} \odot \mathbb{E}_{\frac{5}{3},\frac{1}{3},\frac{7}{3}}(\psi\mathfrak{f}^{\frac{5}{3}},\mathfrak{f}^{\frac{1}{3}}) \oplus$$

$$(-1)(\mathbb{I}^{1;\psi,1}_{\frac{5}{3},\frac{1}{3},\frac{5}{3}}[(1+\psi) \odot \mathbb{S}^2(\mathfrak{r},\mathfrak{f}) \ominus \mathbb{S}^3(\mathfrak{r},\mathfrak{f})])(\mathfrak{f}). \tag{98}$$

(d) If a FVF $\mathbb{S}(\mathfrak{r},\mathfrak{f})$, $^{\mathrm{CF}}_{gH}\mathfrak{D}^{\gamma}_{+}\mathbb{S}(\mathfrak{r},\mathfrak{f})$ and $^{\mathrm{CF}}_{gH}\mathfrak{D}^{\varsigma}_{+}\mathbb{S}(\mathfrak{r},\mathfrak{f})$ are *FFCgH-differentiable* in its $\mathbb{S}^{\circledR} - \mathbb{F}^{\divideontimes}$, then the system (1) contains the solution which is given as

$$\mathbb{S}(\mathfrak{r},\mathfrak{f}) = (7,9,11) \odot (5+\mathfrak{r}) \odot \mathbb{E}_{\frac{5}{3},\frac{1}{3},1}(\psi\mathfrak{f}^{\frac{5}{3}},\mathfrak{f}^{\frac{1}{3}}) \ominus (-1)(2,4,7) \odot (2-\mathfrak{r}^2) \odot \mathfrak{f} \odot \mathbb{E}_{\frac{5}{3},\frac{1}{3},\frac{7}{3}}(\psi\mathfrak{f}^{\frac{5}{3}},\mathfrak{f}^{\frac{1}{3}}) \oplus$$

$$(-1)(2,4,7) \odot (2-\mathfrak{r}^2) \odot \mathfrak{f}^{\frac{4}{3}} \odot \mathbb{E}_{\frac{5}{3},\frac{1}{3},\frac{7}{3}}(\psi\mathfrak{f}^{\frac{5}{3}},\mathfrak{f}^{\frac{1}{3}}) \ominus (7,9,11) \odot (5+\mathfrak{r}) \odot \mathfrak{f}^{\frac{1}{3}} \odot \mathbb{E}_{\frac{5}{3},\frac{1}{3},\frac{4}{3}}(\psi\mathfrak{f}^{\frac{5}{3}},\mathfrak{f}^{\frac{1}{3}}) \ominus$$

$$(\mathbb{I}^{1;\psi,1}_{\frac{5}{3},\frac{1}{3},\frac{5}{3}}[(1+\psi) \odot \mathbb{S}^2(\mathfrak{r},\mathfrak{f}) \ominus \mathbb{S}^3(\mathfrak{r},\mathfrak{f})])(\mathfrak{f}). \tag{99}$$

(e) If a FVF $\mathbb{S}(\mathfrak{r},\mathfrak{f})$, $^{\mathrm{CF}}_{gH}\mathfrak{D}^{\gamma}_{+}\mathbb{S}(\mathfrak{r},\mathfrak{f})$ are *FFCgH-differentiable* in its $\mathbb{F}^{\divideontimes} - \mathbb{F}^{\divideontimes}$ and $^{\mathrm{CF}}_{gH}\mathfrak{D}^{\varsigma}_{+}\mathbb{S}(\mathfrak{r},\mathfrak{f})$ is *FFCgH-differentiable* in its $\mathbb{S}^{\circledR} - \mathbb{F}^{\divideontimes}$, then the system (1) contains the solution which is given as

$$\mathbb{S}(\mathfrak{r},\mathfrak{k}) = (2,4,7) \odot (2-\mathfrak{r}^2) \odot \mathfrak{k} \odot \mathbb{E}_{\frac{5}{3},\frac{1}{3},2}(\psi\mathfrak{k}^{\frac{5}{3}},\mathfrak{k}^{\frac{1}{3}}) \oplus (-1)(7,9,11) \odot (5+\mathfrak{r}) \odot \mathbb{E}_{\frac{5}{3},\frac{1}{3},1}(\psi\mathfrak{k}^{\frac{5}{3}},\mathfrak{k}^{\frac{1}{3}}) \ominus$$

$$(7,9,11) \odot (5+\mathfrak{r}) \odot \mathfrak{k}^{\frac{1}{3}} \odot \mathbb{E}_{\frac{5}{3},\frac{1}{3},\frac{4}{3}}(\psi\mathfrak{k}^{\frac{5}{3}},\mathfrak{k}^{\frac{1}{3}}) \ominus (2,4,7) \odot (2-\mathfrak{r}^2) \odot \mathfrak{k}^{\frac{4}{3}} \odot \mathbb{E}_{\frac{5}{3},\frac{1}{3},\frac{7}{3}}(\psi\mathfrak{k}^{\frac{5}{3}},\mathfrak{k}^{\frac{1}{3}}) \oplus (-1)$$

$$(\mathbb{I}_{\frac{5}{3},\frac{1}{3},\frac{5}{3}}^{1;\psi,1}[(1+\psi) \odot \mathbb{S}^2(\mathfrak{r},\mathfrak{k}) \ominus \mathbb{S}^3(\mathfrak{r},\mathfrak{k})])(\mathfrak{k}). \tag{100}$$

(f) If a FVF $\mathbb{S}(\mathfrak{r},\mathfrak{k}), {}_{gH}^{\mathrm{CF}}\mathfrak{D}_+^\varsigma \mathbb{S}(\mathfrak{r},\mathfrak{k})$ are *FFCgH*-differentiable in its $\mathbb{F}^* - \mathbb{F}^*$ and ${}_{gH}^{\mathrm{CF}}\mathfrak{D}_+^\gamma \mathbb{S}(\mathfrak{r},\mathfrak{k})$ is *FFCgH*-differentiable in its $\mathbb{S}^\circledast - \mathbb{F}^*$, then the system (1) contains the solution which is given as

$$\mathbb{S}(\mathfrak{r},\mathfrak{k}) = (7,9,11) \odot (5+\mathfrak{r}) \odot \mathbb{E}_{\frac{5}{3},\frac{1}{3},1}(\psi\mathfrak{k}^{\frac{5}{3}},\mathfrak{k}^{\frac{1}{3}}) \oplus (2,4,7) \odot (2-\mathfrak{r}^2) \odot \mathfrak{k} \odot \mathbb{E}_{\frac{5}{3},\frac{1}{3},2}(\psi\mathfrak{k}^{\frac{5}{3}},\mathfrak{k}^{\frac{1}{3}}) \oplus (-1)$$

$$(2,4,7) \odot (2-\mathfrak{r}^2) \odot \mathfrak{k}^{\frac{4}{3}} \odot \mathbb{E}_{\frac{5}{3},\frac{1}{3},\frac{7}{3}}(\psi\mathfrak{k}^{\frac{5}{3}},\mathfrak{k}^{\frac{1}{3}}) \oplus (-1)(7,9,11) \odot (5+\mathfrak{r}) \odot \mathfrak{k}^{\frac{1}{3}} \odot \mathbb{E}_{\frac{5}{3},\frac{1}{3},\frac{4}{3}}(\psi\mathfrak{k}^{\frac{5}{3}},\mathfrak{k}^{\frac{1}{3}}) \oplus$$

$$(-1)(\mathbb{I}_{\frac{5}{3},\frac{1}{3},\frac{5}{3}}^{1;\psi,1}[(1+\psi) \odot \mathbb{S}^2(\mathfrak{r},\mathfrak{k}) \ominus \mathbb{S}^3(\mathfrak{r},\mathfrak{k})])(\mathfrak{k}). \tag{101}$$

**Example 32.** Consider the FFGFH-NDE (1) for $\gamma = \dfrac{3}{2}; \varsigma = \dfrac{7}{4}$ along with the initial conditions $(\mathbb{S}_1,\mathbb{S}_2,\mathbb{S}_3) \odot \tilde{\ell}(\mathfrak{r}) = (-3,-1,4) \odot (7-\mathfrak{r}^2)$ and $(\mathbb{S}_1,\mathbb{S}_2,\mathbb{S}_3) \odot \tilde{j}(\mathfrak{r}) = (2,4,7) \odot (\mathfrak{r}^2+4)$. Then from Theorems 25 and 27, the problem (1) contains the following forms of solutions:

1. (a) If a FVF $\mathbb{S}(\mathfrak{r},\mathfrak{k}), {}_{gH}^{\mathrm{CF}}\mathfrak{D}_+^\gamma \mathbb{S}(\mathfrak{r},\mathfrak{k})$ and ${}_{gH}^{\mathrm{CF}}\mathfrak{D}_+^\varsigma \mathbb{S}(\mathfrak{r},\mathfrak{k})$ are *FFCgH*-differentiable in its $\mathbb{F}^* - \mathbb{F}^*$, then the system (1) contains the solution which is given by

$$\mathbb{S}(\mathfrak{r},\mathfrak{k}) = (-3,-1,4) \odot (7-\mathfrak{r}^2) \odot \mathbb{E}_{\frac{7}{4},\frac{1}{4},1}(\mathfrak{k}^{\frac{7}{4}},\mathfrak{k}^{\frac{1}{4}}) \oplus (2,4,7) \odot (\mathfrak{r}^2+4) \odot \mathfrak{k} \odot \mathbb{E}_{\frac{7}{4},\frac{1}{4},2}(\mathfrak{k}^{\frac{7}{4}},\mathfrak{k}^{\frac{1}{4}}) \ominus$$

$$(-3,-1,4) \odot (7-\mathfrak{r}^2) \odot \mathfrak{k}^{\frac{5}{4}} \odot \mathbb{E}_{\frac{7}{4},\frac{1}{4},\frac{5}{4}}(\mathfrak{k}^{\frac{7}{4}},\mathfrak{k}^{\frac{1}{4}}) \ominus (2,4,7) \odot (\mathfrak{r}^2+4) \odot \mathfrak{k}^{\frac{5}{4}} \odot \mathbb{E}_{\frac{7}{4},\frac{1}{4},\frac{9}{4}}(\mathfrak{k}^{\frac{7}{4}},\mathfrak{k}^{\frac{1}{4}}) \ominus$$

$$(\mathbb{I}_{\frac{7}{4},\frac{1}{4},\frac{7}{4}}^{1;1,1}[2 \odot \mathbb{S}^2(\mathfrak{r},\mathfrak{k}) \ominus \mathbb{S}^3(\mathfrak{r},\mathfrak{k})])(\mathfrak{k}). \tag{102}$$

(b) If a FVF $\mathbb{S}(\mathfrak{r},\mathfrak{k})$ is *FFCgH-differentiable* in its $\mathbb{F}^{\circledast}-\mathbb{F}^{\circledast}$ and ${}_{gH}^{\mathrm{CF}}\mathfrak{D}_+^{\varsigma}\mathbb{S}(\mathfrak{r},\mathfrak{k}), {}_{gH}^{\mathrm{CF}}\mathfrak{D}_+^{\gamma}\mathbb{S}(\mathfrak{r},\mathfrak{k})$ are *FFCgH-differentiable* in its $\mathbb{S}^{\circledast}-\mathbb{F}^{\circledast}$, then the system (1) contains the solution which is given by

$$\mathbb{S}(\mathfrak{r},\mathfrak{k}) = (2,4,7) \odot (\mathfrak{r}^2+4) \odot \mathfrak{k} \odot \mathbb{E}_{\frac{7}{4},\frac{1}{4},2}(\mathfrak{k}^{\frac{7}{4}},\mathfrak{k}^{\frac{1}{4}}) \ominus (-1)(-3,-1,4) \odot (7-\mathfrak{r}^2) \odot \mathbb{E}_{\frac{7}{4},\frac{1}{4},1}$$

$$(\mathfrak{k}^{\frac{7}{4}},\mathfrak{k}^{\frac{1}{4}}) \oplus (-1)(-3,-1,4) \odot (7-\mathfrak{k}^2) \odot \mathfrak{k}^{\frac{1}{4}} \odot \mathbb{E}_{\frac{7}{4},\frac{1}{4},\frac{5}{4}}(\mathfrak{k}^{\frac{7}{4}},\mathfrak{k}^{\frac{1}{4}}) \oplus (-1)(2,4,7) \odot (\mathfrak{r}^2+4) \odot \mathfrak{k}^{\frac{5}{4}}$$

$$\odot \mathbb{E}_{\frac{7}{4},\frac{1}{4},\frac{9}{4}}(\mathfrak{k}^{\frac{7}{4}},\mathfrak{k}^{\frac{1}{4}}) \oplus (-1)(\mathbb{I}_{\frac{7}{4},\frac{1}{4},\frac{7}{4}}^{1;1,1}[2 \odot \mathbb{S}^2(\mathfrak{r},\mathfrak{k}) \ominus \mathbb{S}^3(\mathfrak{r},\mathfrak{k})])(\mathfrak{k}). \tag{103}$$

(c) If a FVF $\mathbb{S}(\mathfrak{r},\mathfrak{k})$ is *FFCgH-differentiable* in its $\mathbb{S}^{\circledast}-\mathbb{F}^{\circledast}$ and ${}_{gH}^{\mathrm{CF}}\mathfrak{D}_+^{\varsigma}\mathbb{S}(\mathfrak{r},\mathfrak{k}), {}_{gH}^{\mathrm{CF}}\mathfrak{D}_+^{\gamma}\mathbb{S}(\mathfrak{r},\mathfrak{k})$ are *FFCgH-differentiable* in its $\mathbb{F}^{\circledast}-\mathbb{F}^{\circledast}$, then the system (1) contains the solution which is given by

$$\mathbb{S}(\mathfrak{r},\mathfrak{k}) = (-3,-1,4) \odot (7-\mathfrak{r}^2) \odot \mathbb{E}_{\frac{7}{4},\frac{1}{4},1}(\mathfrak{k}^{\frac{7}{4}},\mathfrak{k}^{\frac{1}{4}}) \oplus (2,4,7) \odot (\mathfrak{r}^2+4) \odot \mathfrak{k} \odot \mathbb{E}_{\frac{7}{4},\frac{1}{4},\frac{5}{4}}$$

$$(\mathfrak{k}^{\frac{7}{4}},\mathfrak{k}^{\frac{1}{4}}) \oplus (-1)(-3,-1,4) \odot (7-\mathfrak{r}^2) \odot \mathfrak{k}^{\frac{1}{4}} \odot \mathbb{E}_{\frac{7}{4},\frac{1}{4},\frac{5}{4}}(\mathfrak{k}^{\frac{7}{4}},\mathfrak{k}^{\frac{1}{4}}) \ominus (2,4,7) \odot (\mathfrak{r}^2+4) \odot \mathfrak{k}^{\frac{5}{4}} \odot$$

$$\mathbb{E}_{\frac{7}{4},\frac{1}{4},\frac{9}{4}}(\mathfrak{k}^{\frac{7}{4}},\mathfrak{k}^{\frac{1}{4}}) \oplus (-1)(\mathbb{I}_{\frac{7}{4},\frac{1}{4},\frac{7}{4}}^{1;1,1}[2 \odot \mathbb{S}^2(\mathfrak{r},\mathfrak{k}) \ominus \mathbb{S}^3(\mathfrak{r},\mathfrak{k})])(\mathfrak{k}). \tag{104}$$

(d) If a FVF $\mathbb{S}(\mathfrak{r},\mathfrak{k}), {}_{gH}^{\mathrm{CF}}\mathfrak{D}_+^{\gamma}\mathbb{S}(\mathfrak{r},\mathfrak{k})$ and ${}_{gH}^{\mathrm{CF}}\mathfrak{D}_+^{\varsigma}\mathbb{S}(\mathfrak{r},\mathfrak{k})$ are *FFCgH-differentiable* in its $\mathbb{S}^{\circledast}-\mathbb{F}^{\circledast}$, then the system (1) contains the solution which is given by

$$\mathbb{S}(\mathfrak{r},\mathfrak{k}) = (-3,-1,4) \odot (7-\mathfrak{r}^2) \odot \mathbb{E}_{\frac{7}{4},\frac{1}{4},1}(\mathfrak{k}^{\frac{7}{4}},\mathfrak{k}^{\frac{1}{4}}) \ominus (-1)(2,4,7) \odot (\mathfrak{r}^2+4) \odot \mathfrak{k} \odot \mathbb{E}_{\frac{7}{4},\frac{1}{4},\frac{9}{4}}$$

$$(\mathfrak{k}^{\frac{7}{4}},\mathfrak{k}^{\frac{1}{4}}) \oplus (-1)(2,4,7) \odot (\mathfrak{r}^2+4) \odot \mathfrak{k}^{\frac{5}{4}} \odot \mathbb{E}_{\frac{7}{4},\frac{1}{4},\frac{9}{4}}(\mathfrak{k}^{\frac{7}{4}},\mathfrak{k}^{\frac{1}{4}}) \ominus (-3,-1,4) \odot (7-\mathfrak{r}^2) \odot \mathfrak{k}^{\frac{1}{4}} \odot$$

$$\mathbb{E}_{\frac{7}{4},\frac{1}{4},\frac{5}{4}}(\mathfrak{k}^{\frac{7}{4}},\mathfrak{k}^{\frac{1}{4}}) \ominus (\mathbb{I}_{\frac{7}{4},\frac{1}{4},\frac{7}{4}}^{1;1,1}[2 \odot \mathbb{S}^2(\mathfrak{r},\mathfrak{k}) \ominus \mathbb{S}^3(\mathfrak{r},\mathfrak{k})])(\mathfrak{k}). \tag{105}$$

(e) If a FVF $\mathbb{S}(\mathfrak{r},\mathfrak{k})$, ${}^{\mathrm{CF}}_{gH}\mathfrak{D}^{\gamma}_{+}\mathbb{S}(\mathfrak{r},\mathfrak{k})$ are *FFCgH-differentiable* in its $\mathbb{F}^{*}-\mathbb{F}^{*}$ and ${}^{\mathrm{CF}}_{gH}\mathfrak{D}^{\varsigma}_{+}\mathbb{S}(\mathfrak{r},\mathfrak{k})$ is *FFCgH-differentiable* in its $\mathbb{S}^{\circledR}-\mathbb{F}^{*}$, then the system (1) contains the solution which is given by

$$\mathbb{S}(\mathfrak{r},\mathfrak{k}) = (2,4,7)\odot(\mathfrak{r}^2+4)\odot\mathfrak{k}\odot\mathbb{E}_{\frac{7}{4},\frac{1}{4},2}(\mathfrak{k}^{\frac{7}{4}},\mathfrak{k}^{\frac{1}{4}})\oplus(-1)(-3,-1,4)\odot(7-\mathfrak{r}^2)\odot\mathbb{E}_{\frac{7}{4},\frac{1}{4},1}$$

$$(\mathfrak{k}^{\frac{7}{4}},\mathfrak{k}^{\frac{1}{4}})\ominus(-3,-1,4)\odot(7-\mathfrak{r}^2)\odot\mathfrak{k}^{\frac{1}{4}}\odot\mathbb{E}_{\frac{7}{4},\frac{1}{4},\frac{5}{4}}(\mathfrak{k}^{\frac{7}{4}},\mathfrak{k}^{\frac{1}{4}})\ominus(2,4,7)\odot(\mathfrak{r}^2+4)\odot\mathfrak{k}^{\frac{5}{4}}\odot$$

$$\mathbb{E}_{\frac{7}{4},\frac{1}{4},\frac{9}{4}}(\mathfrak{k}^{\frac{7}{4}},\mathfrak{k}^{\frac{1}{4}})\oplus(-1)(\mathbb{I}^{1;1,1}_{\frac{7}{4},\frac{1}{4},\frac{7}{4}}[2\odot\mathbb{S}^2(\mathfrak{r},\mathfrak{k})\ominus\mathbb{S}^3(\mathfrak{r},\mathfrak{k})])(\mathfrak{k}). \tag{106}$$

(f) If a FVF $\mathbb{S}(\mathfrak{r},\mathfrak{k})$, ${}^{\mathrm{CF}}_{gH}\mathfrak{D}^{\varsigma}_{+}\mathbb{S}(\mathfrak{r},\mathfrak{k})$ are *FFCgH-differentiable* in its $\mathbb{F}^{*}-\mathbb{F}^{*}$ and ${}^{\mathrm{CF}}_{gH}\mathfrak{D}^{\gamma}_{+}\mathbb{S}(\mathfrak{r},\mathfrak{k})$ is *FFCgH-differentiable* in its $\mathbb{S}^{\circledR}-\mathbb{F}^{*}$, then the system (1) contains the solution which is given by

$$\mathbb{S}(\mathfrak{r},\mathfrak{k}) = (-3,-1,4)\odot(7-\mathfrak{r}^2)\odot\mathbb{E}_{\frac{7}{4},\frac{1}{4},1}(\mathfrak{k}^{\frac{7}{4}},\mathfrak{k}^{\frac{1}{4}})\oplus(2,4,7)\odot(\mathfrak{r}^2+4)\odot\mathfrak{k}\odot\mathbb{E}_{\frac{7}{4},\frac{1}{4},2}$$

$$(\mathfrak{k}^{\frac{7}{4}},\mathfrak{k}^{\frac{1}{4}})\oplus(-1)(2,4,7)\odot(\mathfrak{r}^2+4)\odot\mathfrak{k}^{\frac{5}{4}}\odot\mathbb{E}_{\frac{7}{4},\frac{1}{4},\frac{9}{4}}(\mathfrak{k}^{\frac{7}{4}},\mathfrak{k}^{\frac{1}{4}})\oplus(-1)(-3,-1,4)\odot(7-\mathfrak{r}^2)\odot\mathfrak{k}^{\frac{1}{4}}$$

$$\odot\mathbb{E}_{\frac{7}{4},\frac{1}{4},\frac{5}{4}}(\mathfrak{k}^{\frac{7}{4}},\mathfrak{k}^{\frac{1}{4}})\oplus(-1)(\mathbb{I}^{1;1,1}_{\frac{7}{4},\frac{1}{4},\frac{7}{4}}[2\odot\mathbb{S}^2(\mathfrak{r},\mathfrak{k})\ominus\mathbb{S}^3(\mathfrak{r},\mathfrak{k})])(\mathfrak{k}). \tag{107}$$

2. (a) If a FVF $\mathbb{S}(\mathfrak{r},\mathfrak{k})$, ${}^{\mathrm{CF}}_{gH}\mathfrak{D}^{\gamma}_{+}\mathbb{S}(\mathfrak{r},\mathfrak{k})$ and ${}^{\mathrm{CF}}_{gH}\mathfrak{D}^{\varsigma}_{+}\mathbb{S}(\mathfrak{r},\mathfrak{k})$ are *FFCgH-differentiable* in its $\mathbb{F}^{*}-\mathbb{F}^{*}$, then the system (1) contains the solution which is given by

$$\mathbb{S}(\mathfrak{r},\mathfrak{k}) = (-3,-1,4)\odot(7-\mathfrak{r}^2)\odot\mathbb{E}_{\frac{7}{4},\frac{1}{4},1}(0,\mathfrak{k}^{\frac{1}{4}})\oplus(2,4,7)\odot(\mathfrak{r}^2+4)\odot\mathfrak{k}\odot\mathbb{E}_{\frac{7}{4},\frac{1}{4},2}(0,\mathfrak{k}^{\frac{1}{4}})$$

$$\ominus(-3,-1,4)\odot(7-\mathfrak{r}^2)\odot\mathfrak{k}^{\frac{5}{4}}\odot\mathbb{E}_{\frac{7}{4},\frac{1}{4},\frac{5}{4}}(0,\mathfrak{k}^{\frac{1}{4}})\ominus(2,4,7)\odot(\mathfrak{r}^2+4)\odot\mathfrak{k}^{\frac{5}{4}}\odot\mathbb{E}_{\frac{7}{4},\frac{1}{4},\frac{9}{4}}(0,\mathfrak{k}^{\frac{1}{4}})$$

$$\ominus(\mathbb{I}^{1;0,1}_{\frac{7}{4},\frac{1}{4},\frac{7}{4}}[\mathbb{S}^2(\mathfrak{r},\mathfrak{k})\ominus\mathbb{S}^3(\mathfrak{r},\mathfrak{k})])(\mathfrak{k}). \tag{108}$$

(b) If a FVF $\mathbb{S}(\mathfrak{r},\mathfrak{k})$ is *FFCgH-differentiable* in its $\mathbb{F}^* - \mathbb{F}^*$ and $_{gH}^{\mathrm{CF}}\mathfrak{D}_+^{\varsigma}\mathbb{S}(\mathfrak{r},\mathfrak{k}), _{gH}^{\mathrm{CF}}\mathfrak{D}_+^{\gamma}\mathbb{S}(\mathfrak{r},\mathfrak{k})$ are *FFCgH-differentiable* in its $\mathbb{S}^{\circledR} - \mathbb{F}^*$, then the system (1) contains the solution which is given by

$$\mathbb{S}(\mathfrak{r},\mathfrak{k}) = (2,4,7) \odot (\mathfrak{r}^2+4) \odot \mathfrak{k} \odot \mathbb{E}_{7\,\frac{1}{4},\frac{1}{4},2}(0,\mathfrak{k}^{\overline{\frac{1}{4}}}) \ominus (-1)(-3,-1,4) \odot (7-\mathfrak{r}^2) \odot \mathbb{E}_{7\,\frac{1}{4},\frac{1}{4},1}(0,\mathfrak{k}^{\overline{\frac{1}{4}}})$$

$$\oplus (-1)(-3,-1,4) \odot (7-\mathfrak{r}^2) \odot \mathfrak{k}^{\overline{\frac{1}{4}}} \odot \mathbb{E}_{7\,\frac{1}{4},\frac{1}{4},\frac{5}{4}}(0,\mathfrak{k}^{\overline{\frac{1}{4}}}) \oplus (-1)(2,4,7) \odot (\mathfrak{r}^2+4) \odot \mathfrak{k}^{\overline{\frac{5}{4}}} \odot \mathbb{E}_{7\,\frac{1}{4},\frac{1}{4},\frac{9}{4}}$$

$$(0,\mathfrak{k}^{\overline{\frac{1}{4}}}) \oplus (-1)(\mathbb{I}_{7\,\frac{1}{4},\frac{1}{4},\frac{7}{4}}^{1;0,1}[\mathbb{S}^2(\mathfrak{r},\mathfrak{k}) \ominus \mathbb{S}^3(\mathfrak{r},\mathfrak{k})])(\mathfrak{k}). \tag{109}$$

(c) If a FVF $\mathbb{S}(\mathfrak{r},\mathfrak{k})$ is *FFCgH-differentiable* in its $\mathbb{S}^{\circledR} - \mathbb{F}^*$ and $_{gH}^{\mathrm{CF}}\mathfrak{D}_+^{\varsigma}\mathbb{S}(\mathfrak{r},\mathfrak{k}), _{gH}^{\mathrm{CF}}\mathfrak{D}_+^{\gamma}\mathbb{S}(\mathfrak{r},\mathfrak{k})$ are *FFCgH-differentiable* in its $\mathbb{F}^* - \mathbb{F}^*$, then the system (1) contains the solution which is given by

$$\mathbb{S}(\mathfrak{r},\mathfrak{k}) = (-3,-1,4) \odot (7-\mathfrak{r}^2) \odot \mathbb{E}_{7\,\frac{1}{4},\frac{1}{4},1}(0,\mathfrak{k}^{\overline{\frac{1}{4}}}) \oplus (2,4,7) \odot (\mathfrak{r}^2+4) \odot \mathfrak{k} \odot \mathbb{E}_{7\,\frac{1}{4},\frac{1}{4},\frac{5}{4}}(0,\mathfrak{k}^{\overline{\frac{1}{4}}})$$

$$\oplus (-1)(-3,-1,4) \odot (7-\mathfrak{r}^2) \odot \mathfrak{k}^{\overline{\frac{1}{4}}} \odot \mathbb{E}_{7\,\frac{1}{4},\frac{1}{4},\frac{5}{4}}(0,\mathfrak{k}^{\overline{\frac{1}{4}}}) \ominus (2,4,7) \odot (\mathfrak{r}^2+4) \odot \mathfrak{k}^{\overline{\frac{5}{4}}} \odot \mathbb{E}_{7\,\frac{1}{4},\frac{1}{4},\frac{9}{4}}$$

$$(0,\mathfrak{k}^{\overline{\frac{1}{4}}}) \oplus (-1)(\mathbb{I}_{7\,\frac{1}{4},\frac{1}{4},\frac{7}{4}}^{1;0,1}[\mathbb{S}^2(\mathfrak{r},\mathfrak{k}) \ominus \mathbb{S}^3(\mathfrak{r},\mathfrak{k})])(\mathfrak{k}). \tag{110}$$

(d) If a FVF $\mathbb{S}(\mathfrak{r},\mathfrak{k}), _{gH}^{\mathrm{CF}}\mathfrak{D}_+^{\gamma}\mathbb{S}(\mathfrak{r},\mathfrak{k})$ and $_{gH}^{\mathrm{CF}}\mathfrak{D}_+^{\varsigma}\mathbb{S}(\mathfrak{r},\mathfrak{k})$ are *FFCgH-differentiable* in its $\mathbb{S}^{\circledR} - \mathbb{F}^*$, then the system (1) contains the solution which is given by

$$\mathbb{S}(\mathfrak{r},\mathfrak{k}) = (-3,-1,4) \odot (7-\mathfrak{r}^2) \odot \mathbb{E}_{7\,\frac{1}{4},\frac{1}{4},1}(0,\mathfrak{k}^{\overline{\frac{1}{4}}}) \ominus (-1)(2,4,7) \odot (\mathfrak{r}^2+4) \odot \mathfrak{k} \odot \mathbb{E}_{7\,\frac{1}{4},\frac{1}{4},\frac{9}{4}}$$

$$(0,\mathfrak{k}^{\overline{\frac{1}{4}}}) \oplus (-1)(2,4,7) \odot (\mathfrak{r}^2+4) \odot \mathfrak{k}^{\overline{\frac{5}{4}}} \odot \mathbb{E}_{7\,\frac{1}{4},\frac{1}{4},\frac{9}{4}}(0,\mathfrak{k}^{\overline{\frac{1}{4}}}) \ominus (-3,-1,4) \odot (7-\mathfrak{r}^2) \odot \mathfrak{k}^{\overline{\frac{1}{4}}} \odot \mathbb{E}_{7\,\frac{1}{4},\frac{1}{4},\frac{5}{4}}$$

$$(0,\mathfrak{k}^{\overline{\frac{1}{4}}}) \ominus (\mathbb{I}_{7\,\frac{1}{4},\frac{1}{4},\frac{7}{4}}^{1;0,1}[\mathbb{S}^2(\mathfrak{r},\mathfrak{k}) \ominus \mathbb{S}^3(\mathfrak{r},\mathfrak{k})])(\mathfrak{k}). \tag{111}$$

(e) If a FVF $\mathbb{S}(\mathfrak{r},\mathfrak{k}), {}_{gH}^{\mathrm{CF}}\mathfrak{D}_+^\gamma \mathbb{S}(\mathfrak{r},\mathfrak{k})$ are *FFCgH-differentiable* in its $\mathbb{F}^* - \mathbb{F}^*$ and ${}_{gH}^{\mathrm{CF}}\mathfrak{D}_+^\varsigma \mathbb{S}(\mathfrak{r},\mathfrak{k})$ is *FFCgH-differentiable* in its $\mathbb{S}^\circledR - \mathbb{F}^*$, then the system (1) contains the solution which is given by

$$\mathbb{S}(\mathfrak{r},\mathfrak{k}) = (2,4,7) \odot (\mathfrak{r}^2 + 4) \odot \mathfrak{k} \odot \mathbb{E}_{\frac{7}{4},\frac{1}{4},2}(0,\mathfrak{k}^{\frac{1}{4}}) \oplus (-1)(-3,-1,4) \odot (7-\mathfrak{r}^2) \odot \mathbb{E}_{\frac{7}{4},\frac{1}{4},1}$$

$$(0,\mathfrak{k}^{\frac{1}{4}}) \ominus (-3,-1,4) \odot (7-\mathfrak{r}^2) \odot \mathfrak{k}^{\frac{5}{4}} \odot \mathbb{E}_{\frac{7}{4},\frac{1}{4},\frac{5}{4}}(0,\mathfrak{k}^{\frac{1}{4}}) \ominus (2,4,7) \odot (\mathfrak{r}^2 + 4) \odot \mathfrak{k}^{\frac{5}{4}} \odot \mathbb{E}_{\frac{7}{4},\frac{1}{4},\frac{9}{4}}$$

$$(0,\mathfrak{k}^{\frac{1}{4}}) \oplus (-1)(\mathbb{I}_{\frac{7}{4},\frac{1}{4},\frac{7}{4}}^{1;0,1}[\mathbb{S}^2(\mathfrak{r},\mathfrak{k}) \ominus \mathbb{S}^3(\mathfrak{r},\mathfrak{k})])(\mathfrak{k}). \tag{112}$$

(f) If a FVF $\mathbb{S}(\mathfrak{r},\mathfrak{k}), {}_{gH}^{\mathrm{CF}}\mathfrak{D}_+^\varsigma \mathbb{S}(\mathfrak{r},\mathfrak{k})$ are *FFCgH-differentiable* in its $\mathbb{F}^* - \mathbb{F}^*$ and ${}_{gH}^{\mathrm{CF}}\mathfrak{D}_+^\gamma \mathbb{S}(\mathfrak{r},\mathfrak{k})$ is *FFCgH-differentiable* in its $\mathbb{S}^\circledR - \mathbb{F}^*$, then the system (1) contains the solution which is given by

$$\mathbb{S}(\mathfrak{r},\mathfrak{k}) = (-3,-1,4) \odot (7-\mathfrak{r}^2) \odot \mathbb{E}_{\frac{7}{4},\frac{1}{4},1}(0,\mathfrak{k}^{\frac{1}{4}}) \oplus (2,4,7) \odot (\mathfrak{r}^2+4) \odot \mathfrak{k} \odot \mathbb{E}_{\frac{7}{4},\frac{1}{4},2}(0,\mathfrak{k}^{\frac{1}{4}})$$

$$\oplus (-1)(2,4,7) \odot (\mathfrak{r}^2+4) \odot \mathfrak{k}^{\frac{5}{4}} \odot \mathbb{E}_{\frac{7}{4},\frac{1}{4},\frac{9}{4}}(0,\mathfrak{k}^{\frac{1}{4}}) \oplus (-1)(-3,-1,4) \odot (7-\mathfrak{r}^2) \odot \mathfrak{k}^{\frac{1}{4}} \odot \mathbb{E}_{\frac{7}{4},\frac{1}{4},\frac{5}{4}}$$

$$(0,\mathfrak{k}^{\frac{1}{4}}) \oplus (-1)(\mathbb{I}_{\frac{7}{4},\frac{1}{4},\frac{7}{4}}^{1;0,1}[\mathbb{S}^2(\mathfrak{r},\mathfrak{k}) \ominus \mathbb{S}^3(\mathfrak{r},\mathfrak{k})])(\mathfrak{k}). \tag{113}$$

## Comparative analysis

The model of FFGFH-NDEs offers a novel and general tool for investigating nonlinear dynamical systems with memory, uncertain or imprecise data and spatial interactions. By incorporating fractional order derivatives, the model has captured the influence of long-term memory in both temporal and spatial dynamics. Most of the biological and physical situation involves uncertainties in parameters and measurements. The crisp solutions of FFH-NDEs produces a single deterministic curve that offers exact projections but with accurate conditions without any inclusion of uncertainty. In order to overcome this constraint, the FFGFH-NDEs uses triangular fuzzy-valued initial conditions and fuzzy parameters in order to deal with the lower endpoints, upper endpoints and the points in between the lower and upper endpoints. This leads to the fact that the set of solutions is extended to a family of curves instead of one curve. The upper fuzzy endpoint is the extreme excitability and the lower the future of the fuzzy endpoint is conservative responses. The excitability is also governed by graded values in fuzzy environment in between the lower and upper extreme values giving more generalized visualization for each solution of the FFGFH-NDEs. By changing the values of fuzzy parameters, one gets a novel solution of FFGFH-NDEs for each value and visualization. By fixing some parameters, one can get the crisp solution from the solutions of FFGFH-NDEs. Therefore, the crisp solutions of FFH-NDEs are regarded as the special cases of the solutions

of FFGFH-NDEs. Table 3 provides the numerical solution as fuzzy lower membership solution(Fuzzy Lower), fuzzy peak membership solution(Fuzzy Peak) and fuzzy upper membership solutions(Fuzzy Upper). Table 3 provides the fuzzy solutions of case (*a*) of Example 32 at fixed $\mathfrak{k}$. At the fixed $\mathfrak{k}$, we obtained the variety of fuzzy solutions i.e. fuzzy lower membership, fuzzy peak membership and fuzzy upper membership solutions. Furthermore, our proposed technique gives the graded values of fuzzy solutions which provides the better understanding for understanding the fuzzy solutions and also broader visualization of each fuzzy solution at each point. By changing the fuzzy parameters in each case will provide not only the above mentioned three types of fuzzy graded solution but also provide the fuzzy solutions of the above concerned fuzzy model of FFGFH-NDEs. If we analyse the above table, the other contributors only provided the only one stage of solutions such as Fan et al. [61] provided the solution of crisp model by many semi-analytical techniques such as RPSM, HPM etc. At $\mathfrak{r} = 10$, the approximate solution of FFH-NDEs is $9.79 \times 10^{-1}$ but in our case there are many fuzzy solutions such as $9.97 \times 10^{-1}$ represents the lower fuzzy membership solution, $9.98 \times 10^{-1}$ represents the peak fuzzy membership solution, $9.99 \times 10^{-1}$ represents the upper membership fuzzy solution and there exist many more fuzzy graded solutions in between lower and upper membership fuzzy solutions. The graphical representation of the above fuzzy solutions of FFH-NDEs provides a wider clarity and understanding at each point of the solutions. Thus, the model of FFGFH-NDEs offers a novel and general tool for investigating nonlinear dynamical systems with memory, uncertain or imprecise data and spatial interactions.

Table 4 provides the fuzzy solutions of case (*d*) of Example 32 at fixed $\mathfrak{k}$. At $\mathfrak{r} = 14$, the approximate solution of FFH-NDEs is $9.79 \times 10^{-1}$ but in our case there are many fuzzy solutions such as $9.97 \times 10^{-1}$ represents the lower fuzzy membership solution, $9.99 \times 10^{-1}$ represents the peak fuzzy membership solution, $9.99 \times 10^{-1}$ represents the upper membership fuzzy solution. Table 4 shows that the crisp solution is a particular case of fuzzy solutions within the fuzzy framework, but the fuzzy formulation extends it by quantifying uncertainty and demonstrating improved approximation quality. Similarly, the fuzzy solutions of the remaining cases provide a novel understanding and visualization of FFGFH-NDEs.

**Table 3. Fuzzy solutions of case (a) of Example 32 at fixed $\mathfrak{k} = 0.1$.**

| $\mathfrak{r}$ | Crisp Approx [61] | Fuzzy Lower | Fuzzy Peak | Fuzzy Upper |
|---|---|---|---|---|
| −12 | $1.02 \times 10^{-4}$ | $1.01 \times 10^{-4}$ | $1.04 \times 10^{-3}$ | $1.07 \times 10^{-4}$ |
| −10 | $1.09 \times 10^{-4}$ | $1.05 \times 10^{-4}$ | $1.09 \times 10^{-3}$ | $1.13 \times 10^{-4}$ |
| −8 | $4.46 \times 10^{-3}$ | $4.31 \times 10^{-3}$ | $4.47 \times 10^{-3}$ | $4.62 \times 10^{-3}$ |
| −6 | $1.76 \times 10^{-2}$ | $1.76 \times 10^{-2}$ | $1.85 \times 10^{-2}$ | $1.86 \times 10^{-2}$ |
| −4 | $7.01 \times 10^{-2}$ | $6.97 \times 10^{-2}$ | $7.05 \times 10^{-2}$ | $7.14 \times 10^{-2}$ |
| 4 | $9.56 \times 10^{-1}$ | $9.86 \times 10^{-1}$ | $9.89 \times 10^{-1}$ | $9.96 \times 10^{-1}$ |
| 6 | $9.69 \times 10^{-1}$ | $9.96 \times 10^{-1}$ | $9.97 \times 10^{-1}$ | $9.99 \times 10^{-1}$ |
| 10 | $9.79 \times 10^{-1}$ | $9.97 \times 10^{-1}$ | $9.98 \times 10^{-1}$ | $9.99 \times 10^{-1}$ |
| 12 | $9.81 \times 10^{-1}$ | $9.98 \times 10^{-1}$ | $9.99 \times 10^{-1}$ | $9.99 \times 10^{-1}$ |

**Table 4. Fuzzy solutions of case (d) of Example 32 at fixed $\mathfrak{k} = 0.5$.**

| $\mathfrak{r}$ | Crisp Approx [61] | Fuzzy Lower | Fuzzy Peak | Fuzzy Upper |
|---|---|---|---|---|
| −14 | $2.18 \times 10^{-4}$ | $2.05 \times 10^{-4}$ | $2.25 \times 10^{-4}$ | $2.32 \times 10^{-4}$ |
| −12 | $2.09 \times 10^{-4}$ | $2.05 \times 10^{-4}$ | $2.14 \times 10^{-4}$ | $2.16 \times 10^{-4}$ |
| −10 | $1.06 \times 10^{-3}$ | $1.02 \times 10^{-3}$ | $1.09 \times 10^{-3}$ | $1.13 \times 10^{-3}$ |
| −8 | $4.46 \times 10^{-3}$ | $4.34 \times 10^{-3}$ | $4.47 \times 10^{-3}$ | $4.52 \times 10^{-3}$ |
| −6 | $1.74 \times 10^{-2}$ | $1.76 \times 10^{-2}$ | $1.82 \times 10^{-2}$ | $1.84 \times 10^{-2}$ |
| −4 | $7.02 \times 10^{-2}$ | $6.97 \times 10^{-2}$ | $7.05 \times 10^{-2}$ | $7.11 \times 10^{-2}$ |
| 4 | $7.56 \times 10^{-2}$ | $7.32 \times 10^{-2}$ | $7.62 \times 10^{-2}$ | $7.71 \times 10^{-1}$ |
| 6 | $8.69 \times 10^{-1}$ | $8.56 \times 10^{-1}$ | $8.73 \times 10^{-1}$ | $8.90 \times 10^{-1}$ |
| 10 | $8.78 \times 10^{-1}$ | $8.69 \times 10^{-1}$ | $8.93 \times 10^{-1}$ | $8.96 \times 10^{-1}$ |
| 12 | $9.76 \times 10^{-1}$ | $9.71 \times 10^{-1}$ | $9.92 \times 10^{-1}$ | $9.97 \times 10^{-1}$ |
| 14 | $9.79 \times 10^{-1}$ | $9.97 \times 10^{-1}$ | $9.99 \times 10^{-1}$ | $9.99 \times 10^{-1}$ |

Therefore , the FFGFH-NDEs offer a novel and general tool of investigating nonlinear dynamical systems with memory, uncertainty and spatial interactions.

## 5 Application

Fractional calculus and fuzzy systems have revolutionized circuit theory, particularly in the modeling and analysis of complex, nonlinear and uncertain systems. Traditional models based on integer-order derivatives and precise parameters are often insufficient for modern circuits because of memory effects and uncertainty of the parameters such as resistance, capacitance and inductance that may change with the change of temperature and environmental factors.

The FFGFH-NDEs are beneficial in modeling circuits with memory effects and uncertainty of parameters such as Memristors. This section describes its importance, formulation and real life application of FFGFH-NDEs in digital circuit theory. Both of the frameworks explain nonlinear dynamical systems with memory and nonlocality. The nonlinear dynamical behavior and nonlocality make both the frameworks naturally compatible. In the FFGFH-NDEs model, excitation variable $\mathbb{S}(\mathfrak{r}, \mathfrak{f})$ represents the membrane potential and the recovery variable takes into consideration slower inhibitory dynamics. The inclusion of fuzzy fractional derivatives offers a memory effect, in which the current state is determined by the whole history in the past, and the fuzzy environment includes uncertainty in both initial conditions and system parameters. The voltage serves as the excitation around the memristors in the corresponding memristor networks and the current passing through the memristor shows the recovery. The nonlinearity of the current-voltage characteristic of the memristor is inherent to the cubic nonlinearity of the FFGFH-NDEs and the hysteresis and memory history-dependent resistance of memristors is naturally analogous to the fractional memory kernel of the FFH-NDEs equations. Triangular fuzzy numbers through the fuzzy fractional setting are effectively used to represent variability in device fabrication and values of operational noise; conservative and extreme behavior are simultaneously represented within a single mathematical framework. We establish the one-one correspondence between the functioning of FFGFH-NDEs and memristors networks as follows:

- Fuzzy transmembrane excitation function $\mathbb{S}(\mathfrak{r}, \mathfrak{f}) \Leftrightarrow$ Voltage function in the memristors networks.
- The recovery variable in FFGFH-NDEs $\Leftrightarrow$ Current flowing in memristors.
- The fractional order derivative of FFGFH-NDEs $\Leftrightarrow$ Memory effects in memristors.
- Fuzzy parameters in FFGFH-NDEs $\Leftrightarrow$ Uncertainty or variability of devices in memristors.

Assume that $\mathbb{S}(\mathfrak{r}, \mathfrak{f})$ acts as a voltage function in the context of fuzzy environment, fuzzy fractional order derivative $\left[{}_{gH}^{CF}\mathfrak{D}_+^{\varsigma}\mathbb{S}\right](\mathfrak{r}, \mathfrak{f})$ determines the memory effects of voltage in memristor, $\left[{}_{gH}^{CF}\mathfrak{D}_+^{\gamma}\mathbb{S}\right](\mathfrak{r}, \mathfrak{f})$ represents the diffusion of the voltage signals, the non- linear function $\mathbb{S}(\mathfrak{r}, \mathfrak{f})\left[(1 - \mathbb{S}(\mathfrak{r}, \mathfrak{f}))(\mathbb{S}(\mathfrak{r}, \mathfrak{f}) - \psi)\right]$ describes the changes in state with time, fuzzy threshold function $\psi$ acts as a control function with the graded values with in the range [0,1] and the fuzzy parameters along with the initial conditions as an triangular FVFs helps to investigates how uncertainty propagates through fractional derivatives and non-linear dynamics. The schematic representation and flowchart of FFGFH-NDEs and memristors networks is given in the Figs 7 and 8. Neuromorphic computing, brain-inspired electronics and adaptive signal processing are areas that digital memristor networks can be used. Their dynamics, however, are inherently nonlinear, history dependent and possibly affected by uncertainty related to variation of devices, fabrication requirements and noise in the environment. The FFGFH-NDEs cubic term is analogous to the nonlinear current-voltage characteristic in memristors. The FFGFH-NDEs are the best tool to replicate excitability, threshold switching and oscillatory dynamics observed in digital arrays of memristors. Fractional derivatives have long-term memory effects, like memristor resistance does on the full list of the past history of applied voltage/current. The memory effects such as long-term and short-term memory effects are controlled by CFFgH-derivatives by adjusting the values of fractional orders involved in the FFGFH-NDEs. The memory effects are history dependent which causes the uncertainty in the proposed study. The parameter described by triangular fuzziness in the FFHNDEs enables the generation of solution bands by the model which represent best-case, worst-case

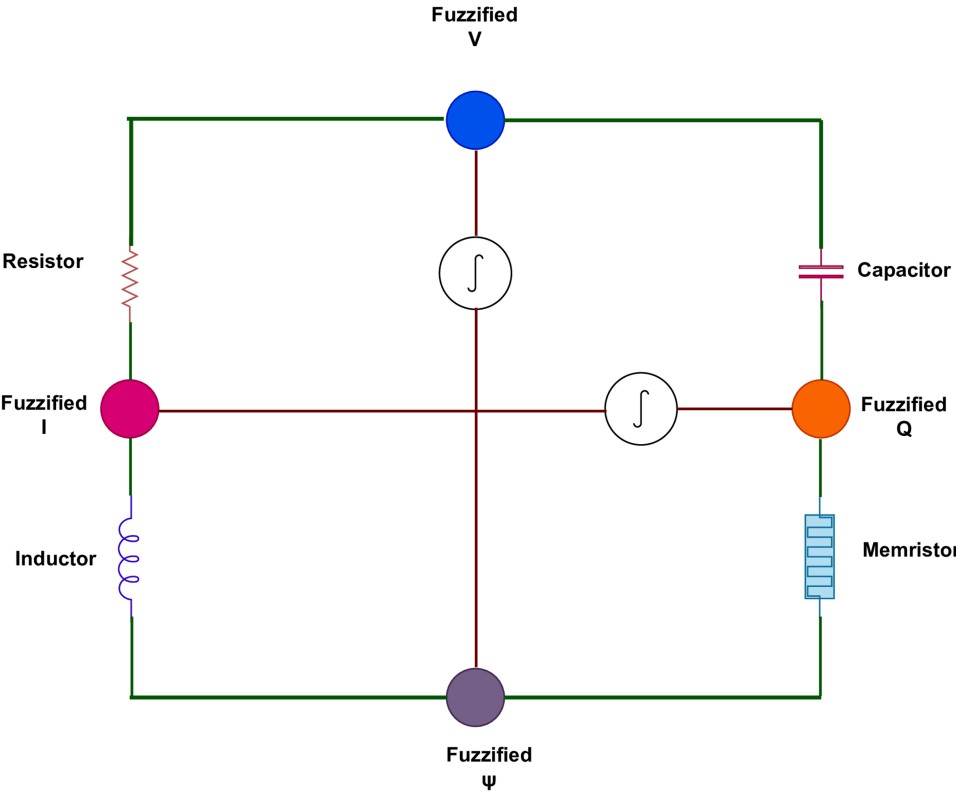

**Fig 7**. Schematic diagram of FFGFH-NDEs as Memristor networks.

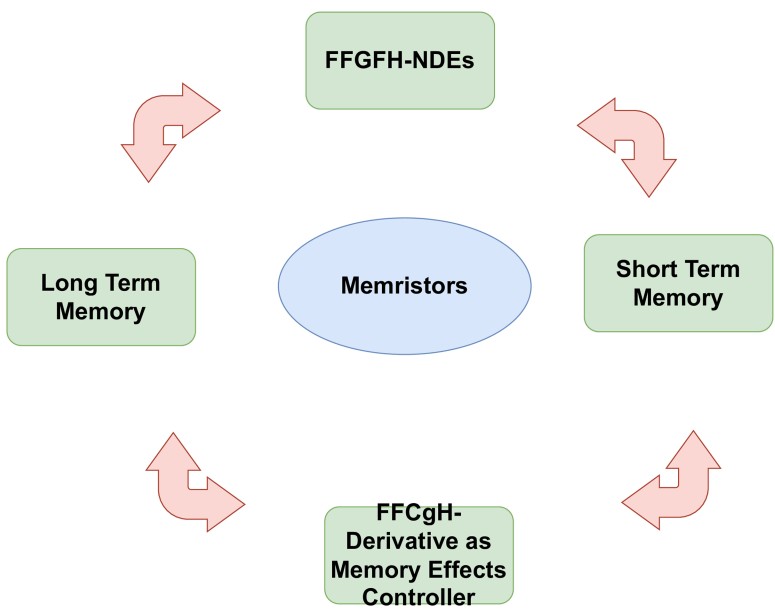

**Fig 8**. Flowchart of FFGFH-NDEs as Memristors networks.

and most-likely behaviors of devices. This allows robust control design, as the fuzzy solution envelope ensures improved performance even at states of uncertainty of the device. Suppose that $\mathbb{S}(\mathfrak{r},\mathfrak{k})[(1-\mathbb{S}(\mathfrak{r},\mathfrak{k}))(\mathbb{S}(\mathfrak{r},\mathfrak{k})-\psi)]=\mathfrak{f}(\mathcal{W}(\mathfrak{k}),\mathcal{I}(\mathfrak{k}))$, then the special case of FFGFH-NDEs in the digital memristors networks is given by:

$$\begin{cases} \left[{}^{\mathrm{CF}}_{gH}\mathfrak{D}^{\varsigma}_{+}\mathbb{S}\right](\mathfrak{r},\mathfrak{k}) \ominus \left[{}^{\mathrm{CF}}_{gH}\mathfrak{D}^{\gamma}_{+}\mathbb{S}\right](\mathfrak{r},\mathfrak{k}) \oplus \mathfrak{f}(\mathcal{W}(\mathfrak{k}),\mathcal{I}(\mathfrak{k})) = 0, \\[2mm] \mathbb{S}(\mathfrak{r},0) = (\mathbb{S}_1,\mathbb{S}_2,\mathbb{S}_3) \odot \tilde{\ell}(\mathfrak{r}), \\[2mm] \mathbb{S}'(\mathfrak{r},0) = (\mathbb{S}_1,\mathbb{S}_2,\mathbb{S}_3) \odot \tilde{j}(\mathfrak{r}), \end{cases} \tag{114}$$

where $\mathbb{S}(\mathfrak{r},0)$ and $\mathbb{S}'(\mathfrak{r},0)$ are the triangular fuzzy initial conditions. Consider the FFGFH-NDEs (114) for $\gamma = \dfrac{4.5}{3}; \varsigma = \dfrac{5.5}{3}$ along with the initial conditions $\mathbb{S}(\mathfrak{r},0)=(-5,-3,1)\odot(3-\mathfrak{r})$ and $\mathbb{S}'(\mathfrak{r},0)=(-7,-4,5)\odot(1+\mathfrak{r})$. Then from Theorem 23, the FFGFH-NDEs (114) contain the following forms of solutions:

(a) If a FVF $\mathbb{S}(\mathfrak{r},\mathfrak{k}), {}^{\mathrm{CF}}_{gH}\mathfrak{D}^{\gamma}_{+}\mathbb{S}(\mathfrak{r},\mathfrak{k})$ and ${}^{\mathrm{CF}}_{gH}\mathfrak{D}^{\varsigma}_{+}\mathbb{S}(\mathfrak{r},\mathfrak{k})$ are *FFCgH-differentiable* in its $\mathbb{F}^{\circledast}-\mathbb{F}^{\circledast}$, then the Eq (114) contains the solution which is given as

$$\mathbb{S}(\mathfrak{r},\mathfrak{k}) = (-5,-3,1)\odot(3-\mathfrak{r})\odot \mathbb{E}_{\frac{5.5}{3},\frac{1}{3},1}(\psi\mathfrak{k}^{\frac{5.5}{3}},\mathfrak{k}^{\frac{1}{3}}) \oplus (-7,-4,5)\odot(1+\mathfrak{r})\odot\mathfrak{k}\odot\mathbb{E}_{\frac{5.5}{3},\frac{1}{3},2}$$

$$(\psi\mathfrak{k}^{\frac{5.5}{3}},\mathfrak{k}^{\frac{1}{3}})\ominus(-5,-3,1)\odot(3-\mathfrak{r})\odot\mathbb{E}_{\frac{5.5}{3},\frac{1}{3},\frac{4}{3}}(\psi\mathfrak{k}^{\frac{5.5}{3}},\mathfrak{k}^{\frac{1}{3}})\ominus(-7,-4,5)\odot(1+\mathfrak{r})\odot\mathfrak{k}^{\frac{4}{3}}\odot\mathbb{E}_{\frac{5.5}{3},\frac{1}{3},\frac{7}{3}}$$

$$(\psi\mathfrak{k}^{\frac{5.5}{3}},\mathfrak{k}^{\frac{1}{3}})\ominus(\mathbb{I}^{1;\psi,1}_{\frac{5.5}{3},\frac{1}{3},\frac{5.5}{3}}[(1+\psi)\odot\mathbb{S}^2(\mathfrak{r},\mathfrak{k})\ominus\mathbb{S}^3(\mathfrak{r},\mathfrak{k})])(\mathfrak{k}). \tag{115}$$

(b) If a FVF $\mathbb{S}(\mathfrak{r},\mathfrak{k})$ is *FFCgH-differentiable* in its $\mathbb{F}^{\circledast}-\mathbb{F}^{\circledast}$ and ${}^{\mathrm{CF}}_{gH}\mathfrak{D}^{\varsigma}_{+}\mathbb{S}(\mathfrak{r},\mathfrak{k}), {}^{\mathrm{CF}}_{gH}\mathfrak{D}^{\gamma}_{+}\mathbb{S}(\mathfrak{r},\mathfrak{k})$ are *FFCgH-differentiable* in its $\mathbb{S}^{\circledcirc}-\mathbb{F}^{\circledast}$, then the Eq (114) contains the solution which is given as

$$\mathbb{S}(\mathfrak{r},\mathfrak{k}) = (-7,-4,5)\odot(1+\mathfrak{r})\odot\mathfrak{k}\odot\mathbb{E}_{\frac{5.5}{3},\frac{1}{3},2}(\psi\mathfrak{k}^{\frac{5.5}{3}},\mathfrak{k}^{\frac{1}{3}})\ominus(-1)(-5,-3,1)\odot(3-\mathfrak{r})\odot\mathbb{E}_{\frac{5.5}{3},\frac{1}{3},1}$$

$$(\psi\mathfrak{k}^{\frac{5.5}{3}},\mathfrak{k}^{\frac{1}{3}})\oplus(-1)(-5,-3,1)\odot(3-\mathfrak{r})\odot\mathfrak{k}^{\frac{1}{3}}\odot\mathbb{E}_{\frac{5.5}{3},\frac{1}{3},\frac{4}{3}}(\psi\mathfrak{k}^{\frac{5.5}{3}},\mathfrak{k}^{\frac{1}{3}})\oplus(-1)(-7,-4,5)\odot(1+\mathfrak{r})$$

$$\odot\mathfrak{k}^{\frac{4}{3}}\odot\mathbb{E}_{\frac{5.5}{3},\frac{1}{3},\frac{7}{3}}(\psi\mathfrak{k}^{\frac{5.5}{3}},\mathfrak{k}^{\frac{1}{3}})\oplus(-1)(\mathbb{I}^{1;\psi,1}_{\frac{5.5}{3},\frac{1}{3},\frac{5.5}{3}}[(1+\psi)\odot\mathbb{S}^2(\mathfrak{r},\mathfrak{k})\ominus\mathbb{S}^3(\mathfrak{r},\mathfrak{k})])(\mathfrak{k}). \tag{116}$$

(c) If a FVF $\mathbb{S}(\mathfrak{r},\mathfrak{k})$ is *FFCgH-differentiable* in its $\mathbb{S}^{\circledast} - \mathbb{F}^{\divideontimes}$ and ${}^{\mathrm{CF}}_{gH}\mathfrak{D}^{\varsigma}_{+}\mathbb{S}(\mathfrak{r},\mathfrak{k}), {}^{\mathrm{CF}}_{gH}\mathfrak{D}^{\gamma}_{+}\mathbb{S}(\mathfrak{r},\mathfrak{k})$ are *FFCgH-differentiable* in its $\mathbb{F}^{\divideontimes} - \mathbb{F}^{\divideontimes}$, then the Eq (114) contains the solution which is given as

$$\mathbb{S}(\mathfrak{r},\mathfrak{k}) = (-5,-3,1) \odot (3-\mathfrak{r}) \odot \mathbb{E}_{\frac{5.5}{3},\frac{1}{3},1} (\psi\mathfrak{k}^{\frac{5.5}{3}}, \mathfrak{k}^{\frac{1}{3}}) \oplus (-7,-4,5) \odot (1+\mathfrak{r}) \odot \mathfrak{k} \odot \mathbb{E}_{\frac{5.5}{3},\frac{1}{3},\frac{4}{3}}$$

$$(\psi\mathfrak{k}^{\frac{5.5}{3}}, \mathfrak{k}^{\frac{1}{3}}) \oplus (-1)(-5,-3,1) \odot (3-\mathfrak{r}) \odot \mathfrak{k}^{\frac{1}{3}} \odot \mathbb{E}_{\frac{5.5}{3},\frac{1}{3},\frac{4}{3}} (\psi\mathfrak{k}^{\frac{5.5}{3}}, \mathfrak{k}^{\frac{1}{3}}) \ominus (-7,-4,5) \odot (1+\mathfrak{r}) \odot \mathfrak{k}^{\frac{4}{3}}$$

$$\odot \mathbb{E}_{\frac{5.5}{3},\frac{1}{3},\frac{7}{3}} (\psi\mathfrak{k}^{\frac{5.5}{3}}, \mathfrak{k}^{\frac{1}{3}}) \oplus (-1)(\mathbb{I}^{1;\psi,1}_{\frac{5.5}{3},\frac{1}{3},\frac{5.5}{3}} [(1+\psi) \odot \mathbb{S}^{2}(\mathfrak{r},\mathfrak{k}) \ominus \mathbb{S}^{3}(\mathfrak{r},\mathfrak{k})])(\mathfrak{k}). \tag{117}$$

(d) If a FVF $\mathbb{S}(\mathfrak{r},\mathfrak{k}), {}^{\mathrm{CF}}_{gH}\mathfrak{D}^{\gamma}_{+}\mathbb{S}(\mathfrak{r},\mathfrak{k})$ and ${}^{\mathrm{CF}}_{gH}\mathfrak{D}^{\varsigma}_{+}\mathbb{S}(\mathfrak{r},\mathfrak{k})$ are *FFCgH-differentiable* in its $\mathbb{S}^{\circledast} - \mathbb{F}^{\divideontimes}$, then the Eq (114) contains the solution which is given as

$$\mathbb{S}(\mathfrak{r},\mathfrak{k}) = (-5,-3,1) \odot (3-\mathfrak{r}) \odot \mathbb{E}_{\frac{5.5}{3},\frac{1}{3},1} (\psi\mathfrak{k}^{\frac{5.5}{3}}, \mathfrak{k}^{\frac{1}{3}}) \ominus (-1)(-7,-4,5) \odot (1+\mathfrak{r}) \odot \mathfrak{k} \odot \mathbb{E}_{\frac{5.5}{3},\frac{1}{3},\frac{7}{3}}$$

$$(\psi\mathfrak{k}^{\frac{5.5}{3}}, \mathfrak{k}^{\frac{1}{3}}) \oplus (-1)(-7,-4,5) \odot (1+\mathfrak{r}) \odot \mathfrak{k}^{\frac{4}{3}} \odot \mathbb{E}_{\frac{5.5}{3},\frac{1}{3},\frac{7}{3}} (\psi\mathfrak{k}^{\frac{5.5}{3}}, \mathfrak{k}^{\frac{1}{3}}) \ominus (-5,-3,1) \odot (3-\mathfrak{r}) \odot \mathfrak{k}^{\frac{1}{3}}$$

$$\odot \mathbb{E}_{\frac{5.5}{3},\frac{1}{3},\frac{4}{3}} (\psi\mathfrak{k}^{\frac{5.5}{3}}, \mathfrak{k}^{\frac{1}{3}}) \ominus (\mathbb{I}^{1;\psi,1}_{\frac{5.5}{3},\frac{1}{3},\frac{5.5}{3}} [(1+\psi) \odot \mathbb{S}^{2}(\mathfrak{r},\mathfrak{k}) \ominus \mathbb{S}^{3}(\mathfrak{r},\mathfrak{k})])(\mathfrak{k}). \tag{118}$$

(e) If a FVF $\mathbb{S}(\mathfrak{r},\mathfrak{k}), {}^{\mathrm{CF}}_{gH}\mathfrak{D}^{\gamma}_{+}\mathbb{S}(\mathfrak{r},\mathfrak{k})$ are *FFCgH-differentiable* in its $\mathbb{F}^{\divideontimes} - \mathbb{F}^{\divideontimes}$ and ${}^{\mathrm{CF}}_{gH}\mathfrak{D}^{\varsigma}_{+}\mathbb{S}(\mathfrak{r},\mathfrak{k})$ is *FFCgH-differentiable* in its $\mathbb{S}^{\circledast} - \mathbb{F}^{\divideontimes}$, then the Eq (114) contains the solution which is given as

$$\mathbb{S}(\mathfrak{r},\mathfrak{k}) = (-7,-4,5) \odot (1+\mathfrak{r}) \odot \mathfrak{k} \odot \mathbb{E}_{\frac{5.5}{3},\frac{1}{3},2} (\psi\mathfrak{k}^{\frac{5.5}{3}}, \mathfrak{k}^{\frac{1}{3}}) \oplus (-1)(-5,-3,1) \odot (3-\mathfrak{r}) \odot \mathbb{E}_{\frac{5.5}{3},\frac{1}{3},1}$$

$$(\psi\mathfrak{k}^{\frac{5.5}{3}}, \mathfrak{k}^{\frac{1}{3}}) \ominus (-5,-3,1) \odot (3-\mathfrak{r}) \odot \mathfrak{k}^{\frac{1}{3}} \odot \mathbb{E}_{\frac{5.5}{3},\frac{1}{3},\frac{4}{3}} (\psi\mathfrak{k}^{\frac{5.5}{3}}, \mathfrak{k}^{\frac{1}{3}}) \ominus (-7,-4,5) \odot (1+\mathfrak{r}) \odot \mathfrak{k}^{\frac{4}{3}} \odot$$

$$\mathbb{E}_{\frac{5.5}{3},\frac{1}{3},\frac{7}{3}} (\psi\mathfrak{k}^{\frac{5.5}{3}}, \mathfrak{k}^{\frac{1}{3}}) \oplus (-1)(\mathbb{I}^{1;\psi,1}_{\frac{5.5}{3},\frac{1}{3},\frac{5.5}{3}} [(1+\psi) \odot \mathbb{S}^{2}(\mathfrak{r},\mathfrak{k}) \ominus \mathbb{S}^{3}(\mathfrak{r},\mathfrak{k})])(\mathfrak{k}). \tag{119}$$

(f) If a FVF $\mathbb{S}(\mathfrak{r},\mathfrak{k})$, $_{gH}^{\mathrm{CF}}\mathfrak{D}_+^\varsigma\mathbb{S}(\mathfrak{r},\mathfrak{k})$ are *FFCgH*-differentiable in its $\mathbb{F}^* - \mathbb{F}^*$ and $_{gH}^{\mathrm{CF}}\mathfrak{D}_+^\gamma\mathbb{S}(\mathfrak{r},\mathfrak{k})$ is *FFCgH*-differentiable in its $\mathbb{S}^\circledR - \mathbb{F}^*$, then the Eq (114) contains the solution which is given as

$$\mathbb{S}(\mathfrak{r},\mathfrak{k}) = (-5,-3,1) \odot (3-\mathfrak{r}) \odot \mathbb{E}_{\frac{5.5}{3},\frac{1}{3},1}(\psi\mathfrak{k}^{\frac{5.5}{3}},\mathfrak{k}^{\frac{1}{3}}) \oplus (-7,-4,5) \odot (1+\mathfrak{r}) \odot \mathfrak{k} \odot \mathbb{E}_{\frac{5.5}{3},\frac{1}{3},2}$$

$$(\psi\mathfrak{k}^{\frac{5.5}{3}},\mathfrak{k}^{\frac{1}{3}}) \oplus (-1)(-7,-4,5) \odot (1+\mathfrak{r}) \odot \mathfrak{k}^{\frac{4}{3}} \odot \mathbb{E}_{\frac{5.5}{3},\frac{1}{3},\frac{7}{3}}(\psi\mathfrak{k}^{\frac{5.5}{3}},\mathfrak{k}^{\frac{1}{3}}) \oplus (-1)(-5,-3,1) \odot (3-\mathfrak{r})\odot$$

$$\mathfrak{k}^{\frac{1}{3}} \odot \mathbb{E}_{\frac{5.5}{3},\frac{1}{3},\frac{4}{3}}(\psi\mathfrak{k}^{\frac{5.5}{3}},\mathfrak{k}^{\frac{1}{3}}) \oplus (-1)(\mathbb{I}_{\frac{5.5}{3},\frac{1}{3},\frac{5.5}{3}}^{1;\psi,1}[(1+\psi) \odot \mathbb{S}^2(\mathfrak{r},\mathfrak{k}) \ominus \mathbb{S}^3(\mathfrak{r},\mathfrak{k})])(\mathfrak{k}). \tag{120}$$

The solutions of the Eq (114) for $\psi = 1$ and $\psi = 0$ can be determined on the similar way by using the Theorems 25 and 27 respectively for any of the case either $1 < \varsigma \le 2, 0 < \gamma \le 1$ or $1 < \varsigma \le 2, 1 < \gamma \le 2$. The generalized FFGFH-NDE demonstrates versatility in modeling complex dynamic behaviors and is particularly effective for systems operating within fuzzy environments. Its ability to incorporate the uncertainty intrinsic to fuzzy systems enables a more accurate depiction of real-world phenomena. The memristor model is a specific application of fractional calculus in circuit theory, focusing on a single component with memory. The FFGFH-NDE model generalizes this concept to include spatial coupling and uncertainty, making it more suitable for complex circuits like advanced memristor networks. It provides a richer framework to study circuits with distributed, nonlinear, and uncertain dynamics. By deriving the analytical solution for FFGFH-NDE, this study enhances the understanding of how such phenomena respond under uncertain conditions. This analytical solution offers a precise and straightforward mathematical framework, enhancing understanding of the system's dynamics and enabling deeper analysis. The graphical depiction of the analytical solution for FFGFH-NDE across varying parameter values of $\varsigma$ and $\gamma$ provided that either $1 < \varsigma \le 2, 0 < \gamma \le 1$ or $1 < \varsigma \le 2, 1 < \gamma \le 2$ introduces a visual perspective to the study. The three-dimensional graphical analysis of FFGFH-NDEs (114) is given by the Figs 9 to 14, showing the behavior for different values of fractional orders $\varsigma$ and $\gamma$. A three-dimensional display in Figs 9 and 10 shows how the fuzzy solutions of the case (*a*) and (*b*) of system (114) change throughout its entire domain while encompassing both the fractional orders. The Figs 11 to 14 give the three-dimensional display of cases (*c*), (*d*), (*e*), (*f*) respectively for fixed values of fractional orders as mentioned in the cases. Fuzzified fractional models serve to represent system uncertainties and physical tolerances according to the research making them suitable for digital circuit theory and biological modeling. The system (114) can be further generalized to many more advanced areas, especially in fuzzy control systems as well as in biological signal processing, because it deals with three fundamental attributes of real-world systems known as memory effects, uncertainty effects, and nonlinear excitability.

## 6 Conclusion

The FFGFH-NDE model offers a novel and general means of investigating nonlinear dynamical systems with memory, uncertainty and spatial interactions. By incorporating fractional order derivatives, the model has captured the influence of long-term memory in both temporal and spatial dynamics. The integration of fuzzy logic into the model allows the model to take into account uncertainties in parameters and makes this model highly applicable to real world science and engineering applications. The FFGFH-NDEs is a well-known and generalized model that plays a significant role in biological systems, including complex synchronization in brain networks, cardiac dynamics, propagation of signals through nerve

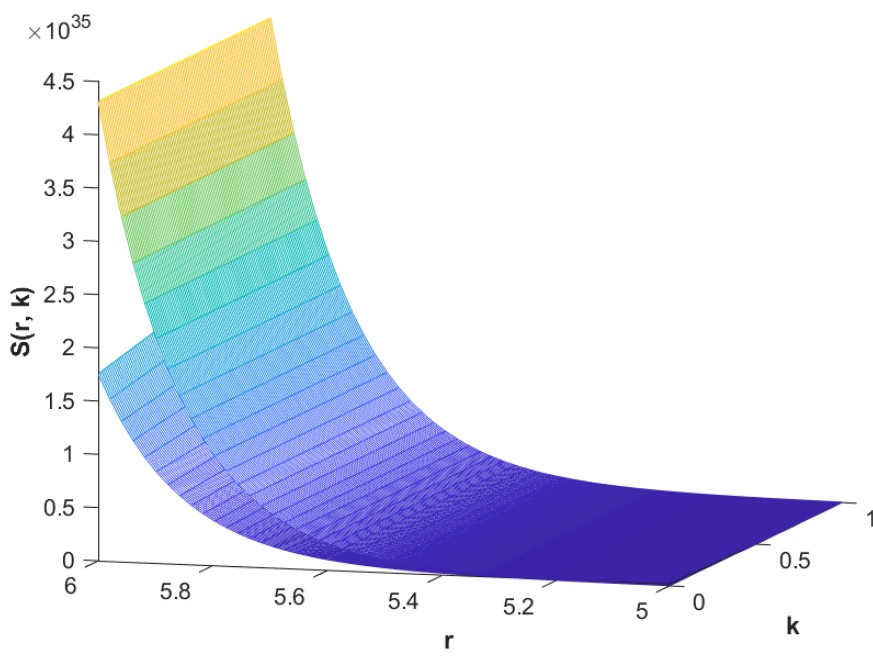

**Fig 9. Graphical representation of case (a) for $\varsigma = \frac{5.5}{3}; \gamma = \frac{4.5}{3}$ and $\psi = 0.9$.**

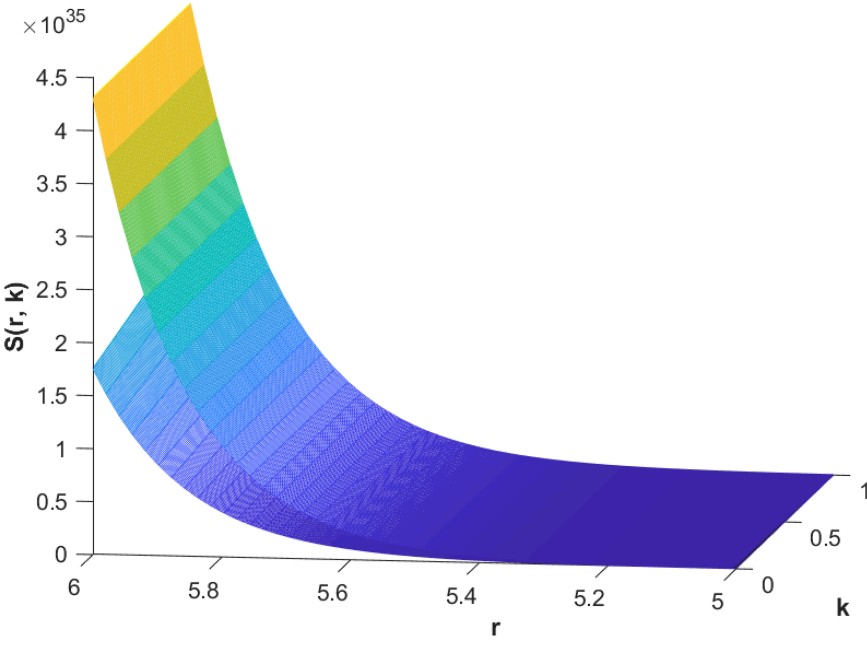

**Fig 10. Graphical representation of case (b) for $\varsigma = \frac{5.5}{3}; \gamma = \frac{4.5}{3}$ and $\psi = 0.9$.**

impulses, and digital circuit theory. An effective and efficient technique is required to solve FFGFH-NDEs analytically. This have presented the analytical fuzzy solutions of FFGFH-NDEs using various cases of the fuzzy fractional Caputo generalized Hukuhara differentiability. The solutions have been formulated and expressed as bivariate and trivariate MLF

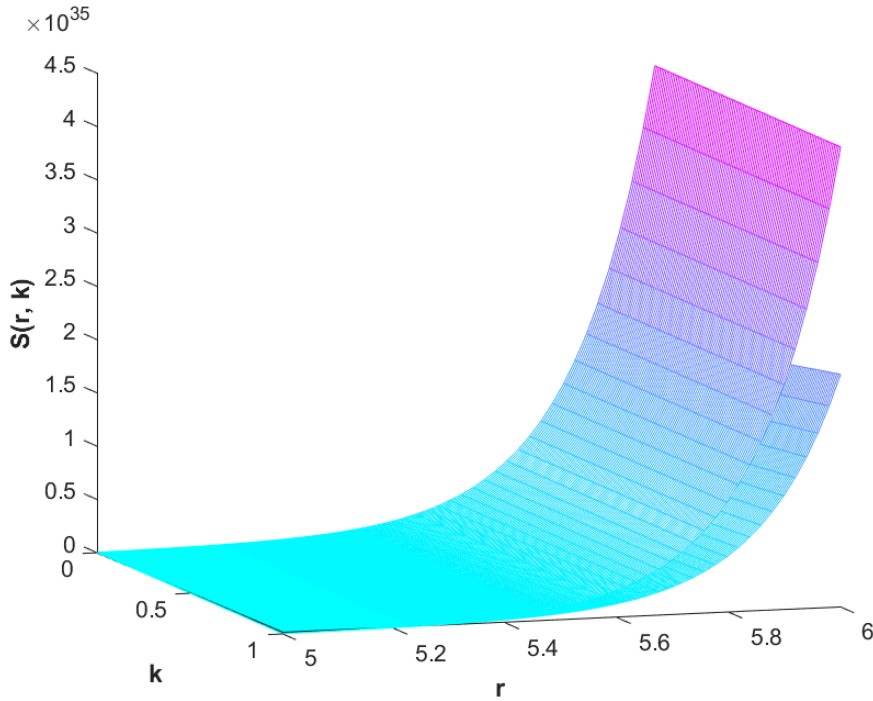

**Fig 11**. Graphical representation of case (*c*) for $\varsigma = \dfrac{5.5}{3}; \gamma = \dfrac{4.5}{3}$ and $\psi = 0.8$.

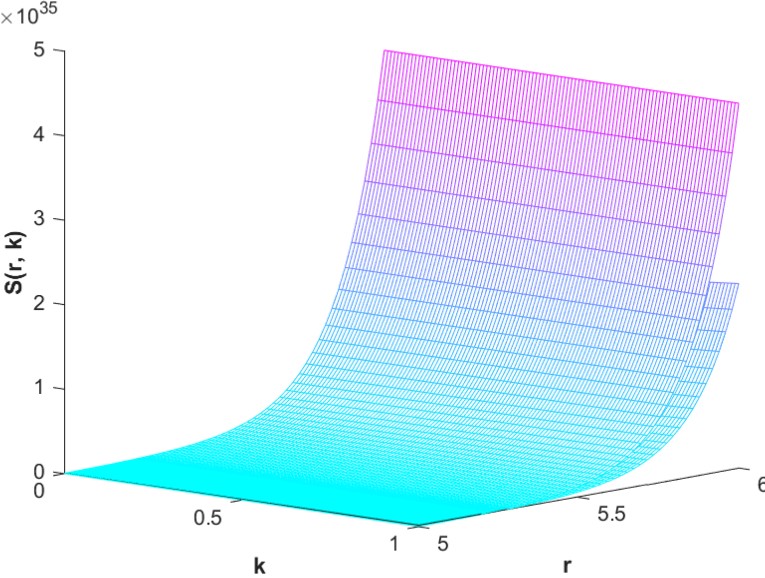

**Fig 12**. Graphical representation of case (*d*) for $\varsigma = \dfrac{5.5}{3}; \gamma = \dfrac{4.5}{3}$ and $\psi = 0.7$.

using Laplace transformation technique. To draw attention to the innovation of this work, we have establish the connection between FFGFH-NDEs and digital memristor networks using one-one correspondence and the functioning behavior of both the models. The incorporation of fuzzy and fractional dynamics has contributed to improving the model's accuracy

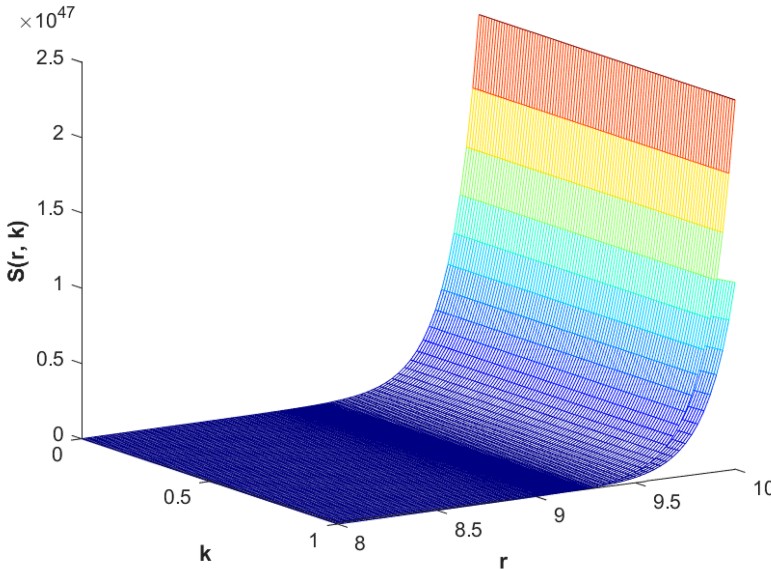

**Fig 13**. Graphical representation of case (e) for $\varsigma = \dfrac{5.5}{3}; \gamma = \dfrac{4.5}{3}$ and $\psi = 0.8$.

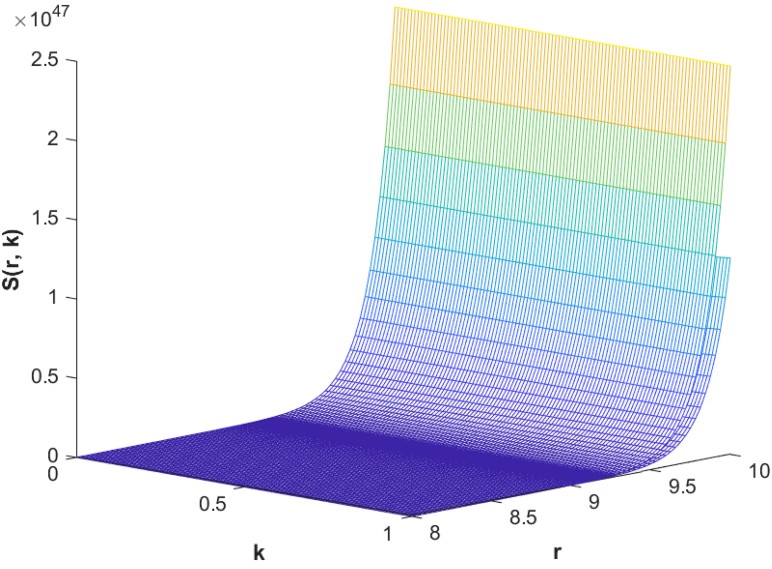

**Fig 14**. Graphical representation of case (f) for $\varsigma = \dfrac{5.5}{3}; \gamma = \dfrac{4.5}{3}$ and $\psi = 0.7$.

and allowed for new pathways for the analysis of systems exhibiting complex and nonlinear behaviors. The graphical representation of the fuzzy solutions of FFGFH-NDEs under various types of *FFCgH*-differentiability is presented to show the novelty of the proposed work. Researchers will use our methodology to solve systems of fuzzy fractional differential equations in the Bi-Polar, Pythagorean, Spherical, m-Polar and Pythagorean m-Polar environments.

## Author contributions

**Conceptualization:** Muhammad Yousuf.

**Formal analysis:** Uzma Ahmad, Ghulam Muhammad, Hamed Alsulami.

**Funding acquisition:** Hamed Alsulami.

**Methodology:** Muhammad Yousuf.

**Supervision:** Uzma Ahmad.

**Visualization:** Ghulam Muhammad.

**Writing – original draft:** Muhammad Yousuf.

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
