## [Decision Letter · Decision Letter 0]

25 Aug 2025

PONE-D-25-28173Generalized FitzHugh-Nagumo equations with Caputo gH-differentiability: A novel fuzzy fractional approach to digital Memristor networksPLOS ONE

Dear Dr.  Ghulam,

Thank you for submitting your manuscript to PLOS ONE. After careful consideration, we feel that it has merit but does not fully meet PLOS ONE’s publication criteria as it currently stands. Therefore, we invite you to submit a revised version of the manuscript that addresses the points raised during the review process.

We look forward to receiving your revised manuscript.

Kind regards,

Muntazir Hussain

Academic Editor

PLOS ONE

Additional Editor Comments:

Suggestions for Improvement

Explicitly explain the significance of applying this model to digital memristor networks—how does the fuzzy fractional modeling enhance control of such systems?

Provide a systematic schematic or flowchart to summarize the mathematical solution approach.

Include a clearer explanation of how the results relate to real-world digital memristors.

Reviewers' comments:

Reviewer's Responses to Questions

**Comments to the Author**

1. Is the manuscript technically sound, and do the data support the conclusions?

Reviewer #1: Yes

Reviewer #2: Yes

Reviewer #3: Yes

Reviewer #4: Yes

Reviewer #5: Yes

2. Has the statistical analysis been performed appropriately and rigorously?

Reviewer #1: Yes

Reviewer #2: No

Reviewer #3: N/A

Reviewer #4: No

Reviewer #5: Yes

3. Have the authors made all data underlying the findings in their manuscript fully available?

Reviewer #1: Yes

Reviewer #2: No

Reviewer #3: Yes

Reviewer #4: Yes

Reviewer #5: Yes

4. Is the manuscript presented in an intelligible fashion and written in standard English?

Reviewer #1: Yes

Reviewer #2: No

Reviewer #3: No

Reviewer #4: Yes

Reviewer #5: Yes

5. Review Comments to the Author

Reviewer #1: 1. Explain clearly fuzzy fractional generalized FitzHugh-Nagumo differential equations ?

2. How do fuzzy fractional generalized FitzHugh-Nagumo differential equations enhance the modeling of neurodynamical systems by incorporating uncertainty, memory effects, and nonlinear behavior?

Reviewer #2: Dear Authors, thank you for submitting manuscript with interesting topic and with mathematical derivations appear very good. However, I have few comments that need to be addressed for improvement of this manuscript.

(1) This manuscript lacks the validation of analytical solutions. Author must provide the evidence that solution are accurate and relevant to real-world memristor network. I suggest

(a) Compare your a analytical solutions with numerial solutions.

(b) Provide a more detailed discussion of the parameters used in your memristor network application.

(2) Figures could be improved. Add axis labels and legends to make more easier to understand.

(3) The English language needs significant improvement. There are numerous grammatical errors, inconsistencies in notation.

(4) Improve the flow of the manuscript by adding more introductory and concluding sentences to each section.

I hope addressing these concerns will improve the quality of manuscript for publication.

Reviewer #3: This manuscript presents a fuzzy fractional modeling approach to generalized FitzHugh–Nagumo (FHN) equations incorporating Caputo gH-differentiability, with application to digital memristor networks. The study integrates nonlinear neuroscience modeling with fractional fuzzy calculus and emerging electronic devices, which is an interesting and multidisciplinary topic. The mathematical formulation is solid, but some aspects require clarification or refinement for technical completeness.

1. The physical motivation for applying the Caputo gH-differentiable fractional framework to the FHN model should be explained more clearly, especially its specific advantage over other fractional definitions in the context of memristor network dynamics.

2. The description of the generalized FHN system would benefit from explicitly listing all assumptions, including parameter constraints, initial conditions, and boundary conditions.

3. The derivation of the fuzzy fractional model lacks intermediate mathematical steps in some areas—adding these would improve transparency and reproducibility.

4. The link between the fuzzy fractional formulation and the memristor network model should be strengthened by explaining the mapping of FHN variables to memristor parameters.

5. Numerical implementation details (e.g., discretization method, time step size, stability considerations) are too brief—these should be expanded for clarity.

6. Figures showing simulation results should include axis labels with units where applicable, and legends should clearly identify different parameter cases.

7. A brief comparative discussion with other fractional-order modeling techniques (e.g., Atangana–Baleanu, Caputo–Fabrizio derivatives) could help highlight the novelty of the chosen method.

8. The conclusion should elaborate more on the potential practical implications in neuromorphic computing and whether the proposed approach is computationally feasible for large-scale memristor arrays.

9. For enhancing the introduction section with the new publications, old references may be replaced with new ones such as:

Artificial neural network validation of MHD natural bioconvection in a square enclosure: entropic analysis and optimization

Reviewer #4: The manuscript demonstrates significant potential and originality but requires substantial improvements in:

1. Clearer articulation of novelty.

2. Comparative and statistical validation of results.

3. Stronger discussion and conclusion.

4. Updated and more consistent references.

Reviewer #5: Review Report

The paper addresses the fuzzy fractional generalized FitzHugh-Nagumo differential equations (FFGFH-NDEs), which are crucial in biological systems, neuroscience, cardiac dynamics, and digital circuit theory. The study is well-aligned with the interdisciplinary scope of applied mathematics, fuzzy analysis, and computational neuroscience.

• Provide numerical simulations alongside the analytical solutions to validate results.

• Discuss biological interpretations in more depth, linking mathematics to real-world brain or cardiac systems.

• Add a comparative analysis with existing solution techniques (e.g., Adomian decomposition, homotopy perturbation, or PINNs).

• Enhance figures and graphical results with clearer visual representation.

6. PLOS authors have the option to publish the peer review history of their article (what does this mean?). If published, this will include your full peer review and any attached files.

Reviewer #1: No

Reviewer #2: No

Reviewer #3: No

Reviewer #4: No

Reviewer #5: No

---

## [Author Response · Author response to Decision Letter 1]

22 Sep 2025

\noindent August 25, 2025\\

Journal: PLOS One\\

Manuscript ID: Ms. Ref. No.: PONE-D-25-28173\\

Title: Generalized FitzHugh-Nagumo equations with Caputo gH-differentiability: A novel fuzzy fractional approach to digital Memristor networks\\\\

Dear Emily Chenette,\\\\

We sincerely appreciate the opportunity to revise our manuscript for possible publication in PLOS One and would like to thank you and the reviewers for their guidance and comments in your letter dated August 25, 2025. We have carefully incorporated their suggestions and made the necessary changes, including fixing grammatical and typing errors and improving the writing in certain areas. We are grateful for the valuable input provided by the reviewers and agree with all the comments made. We hope that the revised version of our paper meets their expectations and is suitable for publication in your esteemed journal.\\\\

 Thank you for your time and consideration.\\\\

Best Regards,\\

Dr. G. Muhammad\\

(Corresponding Author)

\newpage

\begin{center}

{\bf\large  Response to the Comments of Reviewer 1}

\end{center}

1. Explain clearly fuzzy fractional generalized FitzHugh-Nagumo differential equations ?

\textcolor{blue} {\bf Our Response.}

We appreciate this constructive comment. We have revised and clearly explained the fuzzy fractional generalized FitzHugh-Nagumo differential equations in section 3 according to your suggestions.\\\\

2. How do fuzzy FFHNDEs enhance the modeling of neurodynamical systems by incorporating uncertainty, memory effects, and nonlinear behavior?

\textcolor{blue} {\bf Our Response.}

We appreciate this constructive comment. We have explained that how fuzzy fractional generalized FitzHugh-Nagumo differential equations enhance the modeling of neurodynamical systems by incorporating uncertainty, memory effects, and nonlinear behavior in introduction and section 5.\\

\begin{center}

{\bf\large  Response to the Comments of Reviewer 2}

\end{center}

1. This manuscript lacks the validation of analytical solutions. Author must provide the evidence that solution are accurate and relevant to real-world memristor network. I suggest (a)..., (b)...

\textcolor{blue} {\bf Our Response.}

We appreciate your constructive comment. We have provided the detailed comparison of the proposed generalized fuzzy solutions and provided the detailed discussion in order to validate our results in section 4.\\

2. Figures could be improved. Add axis labels and legends to make more easier to understand.

\textcolor{blue} {\bf Our Response.}

Thank you so much for your valuable comment. We have improved the figures by showing the axes clearly.\\

3. The English language needs significant improvement. There are numerous grammatical errors, inconsistencies in notation.

\textcolor{blue} {\bf Our Response.}

Thank you so much for your valuable comment. We have corrected the grammatical errors and improved the manuscript by adding concluding sentences to each section.

\begin{center}

{\bf\large  Response to the Comments of Reviewer 3}

\end{center}

1. The physical motivation for applying the Caputo gH-differentiable fractional framework to the FHN model should be explained more clearly, especially its specific advantage over other fractional definitions in the context of memristor network dynamics.\\

\textcolor{blue} {\bf Our Response.}

Thank you so much for your valuable comment. We have provided the physical motivation for applying the Caputo gH-differentiable in the introduction section and explained the proposed model in section 3 and the real-world benefits of the propsed model as memristors networks in section 5.\\

2,3. The derivation of the fuzzy fractional model lacks intermediate mathematical steps in some areasâ€”adding these would improve transparency and reproducibility.\\

\textcolor{blue} {\bf Our Response.}

Thank you so much for your valuable comment. We have revised the derivation of the fuzzy fractional model and organized the fuzzy solutions in a arranged way in section 3.\\

4. The link between the fuzzy fractional formulation and the memristor network model should be strengthened by explaining the mapping of FHN variables to memristor parameters.\\

\textcolor{blue} {\bf Our Response.}

Thank you so much for your valuable comment. We have provided the one-one correspondence of proposed model as an application of fuzzy memristor networks in in section 5.\\

5,6,7. Numerical implementation details (e.g., discretization method, time step size, stability considerations) are too brief these should be expanded for clarity. A brief comparative discussion with other fractional-order modeling techniques (e.g., Atangana-Baleanu, Caputo-Fabrizio derivatives) could help highlight the novelty of the chosen method.

\textcolor{blue} {\bf Our Response.}

Thank you so much for your valuable comment. We have provided the numerical solutions at fixed fuzzy parameters and variables in order to determine the novelty and accuracy of the solutions of proposed model. We have provided the comparative discussion of fuzzy solutions with the other existing techniques to show the novelty of the fuzzy solutions.\\

8. The conclusion should elaborate more on the potential practical implications in neuromorphic computing and whether the proposed approach is computationally feasible for large-scale memristor arrays.\\

\textcolor{blue} {\bf Our Response.}

We have revised the conclusion section and explained the results in a broader way. \\

9. For enhancing the introduction section with the new publications, old references may be replaced with new ones such as:

Artificial neural network validation of MHD natural bioconvection in a square enclosure: entropic analysis and optimization\\

\textcolor{blue} {\bf Our Response.}

We have revised the introduction section and added the new pubilcations related to the concerned work according to your suggestions.

\begin{center}

{\bf\large  Response to the Comments of Reviewer 4}

\end{center}

The manuscript demonstrates significant potential and originality but requires substantial improvements in:1. Clearer articulation of novelty.2. Comparative and statistical validation of results.....

\textcolor{blue} {\bf Our Response.}

Thank you very much for your constructive commets on this manuscript. We have revised the article and provided clear understanding of novelty,

comparative analysis, stronger discussion, and updated references related to proposed model.

\begin{center}

{\bf\large  Response to the Comments of Reviewer 5}

\end{center}

The paper addresses the fuzzy fractional generalized FitzHugh-Nagumo differential equations (FFGFH-NDEs), which are crucial in biological systems...

\textcolor{blue} {\bf Our Response.}

Thank you very much for your constructive comments on this manuscript. We have provided the real-world application of the proposed model in memristors networks, added the numerical results of fuzzy solutions for fixed fuzzy parameters and variables, comparative analysis of fuzzy solutions with the exixting solutions and the graphical representations in the clear way according to the valuable suggestions.\\

Thank you very much!!

---

## [Decision Letter · Decision Letter 1]

19 Nov 2025

PONE-D-25-28173R1Generalized FitzHugh-Nagumo equations with Caputo gH-differentiability: A novel fuzzy fractional approach to digital Memristor networksPLOS ONE

Dear Dr. Muhammad,

Thank you for submitting your manuscript to PLOS ONE. After careful consideration, we feel that it has merit but does not fully meet PLOS ONE’s publication criteria as it currently stands. Therefore, we invite you to submit a revised version of the manuscript that addresses the minor points raised during the review process.

We look forward to receiving your revised manuscript.

Kind regards,

Muntazir Hussain

Academic Editor

PLOS ONE

Journal Requirements:

Additional Editor Comments:

The manuscript presents a study with good theoretical and applied contributions. All major concerns have been addressed. Please consider minor refinements.

Reviewers' comments:

Reviewer's Responses to Questions

**Comments to the Author**

1. If the authors have adequately addressed your comments raised in a previous round of review and you feel that this manuscript is now acceptable for publication, you may indicate that here to bypass the “Comments to the Author” section, enter your conflict of interest statement in the “Confidential to Editor” section, and submit your "Accept" recommendation.

Reviewer #2: All comments have been addressed

Reviewer #3: (No Response)

Reviewer #5: All comments have been addressed

2. Is the manuscript technically sound, and do the data support the conclusions?

Reviewer #2: Yes

Reviewer #3: (No Response)

Reviewer #5: Yes

3. Has the statistical analysis been performed appropriately and rigorously?

Reviewer #2: Yes

Reviewer #3: (No Response)

Reviewer #5: I Don't Know

4. Have the authors made all data underlying the findings in their manuscript fully available?

Reviewer #2: Yes

Reviewer #3: (No Response)

Reviewer #5: Yes

5. Is the manuscript presented in an intelligible fashion and written in standard English?

Reviewer #2: Yes

Reviewer #3: (No Response)

Reviewer #5: Yes

6. Review Comments to the Author

Reviewer #2: The manuscript presents a novel and well-executed study with strong theoretical and applied contributions. The suggested improvements are minor but aimed at enhancing clarity and usability.

1) Summary Table of Differentiability Cases:Include a concise table summarizing all differentiability types and corresponding solution forms.

2) Schematic Diagram:Add a visual diagram illustrating the mapping between FFGFH-NDEs components and memristor network elements.

3) Notation Table:Include a table of symbols and notations used throughout the manuscript for clarity.

4) Broader Implications:Expand the discussion on how the proposed model could be extended to other domains such as fuzzy control systems or biological signal processing.

Reviewer #3: Authors have done the required amendments and article is ready for publication.

Reviewer #5: Accepted in the current form. All comments are addressed with full Concentration and care. paper can be accepted now

7. PLOS authors have the option to publish the peer review history of their article (what does this mean?). If published, this will include your full peer review and any attached files.

Reviewer #2: No

Reviewer #3: No

Reviewer #5: **Yes:** Assad Ayub

---

## [Author Response · Author response to Decision Letter 2]

26 Nov 2025

\noindent November 19, 2025\\

Journal: PLOS One\\

Manuscript ID: Ms. Ref. No.: PONE-D-25-28173\\

Title: Generalized FitzHugh-Nagumo equations with Caputo gH-differentiability: A novel fuzzy fractional approach to digital Memristor networks\\\\

Dear Emily Chenette,\\\\

We sincerely appreciate the opportunity to revise our manuscript for possible publication in PLOS One and would like to thank you and the reviewers for their guidance and comments in your letter dated November 19, 2025. We have carefully incorporated their suggestions and made the necessary changes, including fixing grammatical and typing errors and improving the writing in certain areas. We are grateful for the valuable input provided by the reviewers and agree with all the comments made. We hope that the revised version of our paper meets their expectations and is suitable for publication in your esteemed journal.\\\\

 Thank you for your time and consideration.\\\\

Best Regards,\\

Dr. G. Muhammad\\

(Corresponding Author)

\newpage

\begin{center}

{\bf\large  Response to the Comments of Reviewer 2}

\end{center}

\begin{enumerate}

\item Summary Table of Differentiability Cases: Include a concise table summarizing all differentiability types and corresponding solution forms.\\

\textcolor{blue} {\bf Our Response.}

We appreciate this valuable comment. We have presented the summary table summarizing all differentiability types and corresponding solution forms.\\

\item Schematic Diagram: Add a visual diagram illustrating the mapping between FFGFH-NDEs components and memristor network elements.\\

\textcolor{blue} {\bf Our Response.}

We appreciate this worthy comment. We have presented the schematic diagrams illustrating the mapping between FFGFH-NDEs components and memristor network.\\

\item Notation Table: Include a table of symbols and notations used throughout the manuscript for clarity.\\

\textcolor{blue} {\bf Our Response.}

We appreciate your comment. We have presented the notation table of symbols and notation used throughout the manuscript.\\

\item Broader Implications: Expand the discussion on how the proposed model could be extended to other domains such as fuzzy control systems or biological signal processing.\\\\

\textcolor{blue} {\bf Our Response.}

Thank you so much for your constructive comment. We have presented the discussion on the extension of proposed model for suggested domains. We have also revised and checked the correctness of all the references.\\

\end{enumerate}

---

## [Editor Report · Decision Letter 2]

15 Dec 2025

Generalized FitzHugh-Nagumo equations with Caputo gH-differentiability:~ A novel fuzzy fractional approach to digital Memristor networks

PONE-D-25-28173R2

Dear Dr. Ghulam,

We’re pleased to inform you that your manuscript has been judged scientifically suitable for publication and will be formally accepted for publication once it meets all outstanding technical requirements.

Kind regards,

Muntazir Hussain

Academic Editor

PLOS One
---

## [Editor Report · Acceptance letter]

PONE-D-25-28173R2

PLOS One

Dear Dr. Muhammad,

I'm pleased to inform you that your manuscript has been deemed suitable for publication in PLOS One. Congratulations! Your manuscript is now being handed over to our production team.

Kind regards,

on behalf of

Dr. Muntazir Hussain

Academic Editor

PLOS One